# On the Surprising Effectiveness of Large Learning Rates under Standard Width Scaling

**Moritz Haas**[1]    **Sebastian Bordt**[1]    **Ulrike von Luxburg**[1]    **Leena Chennuru Vankadara**[2]

[1]University of Tübingen, Tübingen AI Center
`{mo.haas,sebastian.bordt,ulrike.luxburg}@uni-tuebingen.de`

[2]Gatsby Computational Neuroscience Unit, University College London
`l.vankadara@ucl.ac.uk`

## Abstract

Scaling limits, such as infinite-width limits, serve as promising theoretical tools to study large-scale models. However, it is widely believed that existing infinite-width theory does not faithfully explain the behavior of practical networks, especially those trained in *standard parameterization* (SP) meaning He initialization with a global learning rate. For instance, existing theory for SP predicts instability at large learning rates and vanishing feature learning at stable ones. In practice, however, optimal learning rates decay slower than theoretically predicted and networks exhibit both stable training and non-trivial feature learning, even at very large widths. Here, we show that this discrepancy is not fully explained by finite-width phenomena.

Instead, we find a resolution through a finer-grained analysis of the regime previously considered unstable and therefore uninteresting. In particular, we show that, under the cross-entropy (CE) loss, the unstable regime comprises two distinct sub-regimes: a catastrophically unstable regime and a more benign controlled divergence regime, where logits diverge but gradients and activations remain stable. Moreover, under large learning rates at the edge of the controlled divergence regime, there exists a well-defined infinite width limit where features continue to evolve in all the hidden layers. In experiments across optimizers, architectures, and data modalities, we validate that neural networks operate in this controlled divergence regime under CE loss but not under MSE loss. Our empirical evidence suggests that width-scaling considerations are surprisingly useful for predicting empirically maximal stable learning rate exponents which provide useful guidance on optimal learning rate exponents. Finally, our analysis clarifies the effectiveness and limitations of recently proposed layerwise learning rate scalings for standard initialization.

 **Experiment Code**   |    **Refined Coordinate Check Package**

## 1 Introduction

Scaling has become the dominant paradigm in building ever more capable vision and language models (Brown et al., 2020, Dosovitskiy et al., 2021, Kaplan et al., 2020, Hoffmann et al., 2022, Grattafiori et al., 2024). Infinite-width limits have served as a crucial theoretical tool for understanding large models, providing valuable insights into their optimization and generalization behaviour (Jacot et al., 2018, Du et al., 2018, Allen-Zhu et al., 2019, Arora et al., 2019, Mei et al., 2018, Rotskoff and Vanden-Eijnden, 2022, Sirignano and Spiliopoulos, 2020, Lai et al., 2023). Nevertheless, it is now widely believed that existing infinite-width theory, particularly in the kernel regime, does not

serve as a faithful proxy for practical neural networks, as it fails to capture fundamental aspects of their training (Sohl-Dickstein et al., 2020, Lee et al., 2020, Vyas et al., 2022, Wenger et al., 2023).

This disconnect is especially prominent for the dominant training practice, *standard parameterization* (SP): He initialization (He et al., 2015) with a single global learning rate (OLMo Team et al., 2024). For example, infinite-width theory predicts that under SP, network dynamics should become unstable with learning rates scaling larger than $\mathcal{O}(1/n)$ (where $n$ is network width), and that feature learning vanishes with $\mathcal{O}(1/n)$ learning rates, causing the models to enter a kernel regime (Sohl-Dickstein et al., 2020, Yang and Hu, 2021). Empirically, however, networks trained in SP exhibit stable feature learning and excellent generalization performance, often with optimal learning rates decaying much slower than theoretically predicted (commonly around $\Omega(1/\sqrt{n})$). This is depicted in Figure 1, where we see that the optimal learning rates (solid lines) for different models trained in SP decay much slower than the theoretically predicted maximal stable scaling law (dashed gray lines). These observations represent a fundamental puzzle and motivate two crucial open questions:

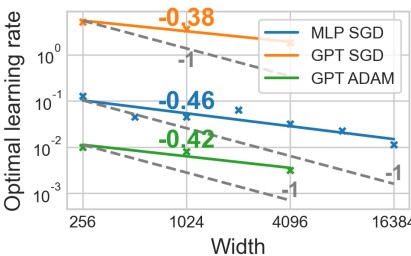

Figure 1: **Optimal learning rate exponents exceed the theoretically predicted stability threshold.** For MLPs on MNIST and GPT on language data, optimal learning rates in SP decay slower than the theoretically predicted maximal stable $\eta_n = \mathcal{O}(n^{-1})$ in gray.

*Why does SP remain stable and effective at large learning rates, despite the theoretical predictions? And does there exist an infinite-width limit that corresponds more closely with the behaviour of practical finite-width networks?*

In this work, we investigate these questions and find a resolution through a finer grained analysis of the regime previously dismissed as unstable. Consequently, we provide the first infinite-width proxy that corresponds more closely to practical, finite-width networks. Our main contributions are:

(a) **We investigate and rule out finite-width effects as the key explanation.** Plausible explanations to explain these discrepancies include finite-width effects, either accumulated over large depth or longer training times. We show these explanations are insufficient, as the gap is surprisingly more pronounced in shallow (2-layer) MLPs in a single pass (Figure F.15). Furthermore, we investigate other known finite-width dynamics, like the catapult regime (Lewkowycz et al., 2020) or the edge of stability (Cohen et al., 2021, 2022) in simplified linear models, and show that these mechanisms alone also cannot explain the stability of large learning rates in SP.

(b) **We validate infinite-width alignment predictions.** Contrary to what was hypothesized in previous work (Everett et al., 2024), we show that the infinite-width alignment predictions between weights and incoming activations indeed hold at moderate width when measured with sufficiently refined coordinate checks (RCC). This is a crucial finding, since it confirms the theoretical prediction that logits do diverge at sufficient width in SP under empirically optimal learning rates.

(c) **The resolution.** Instead, we find a resolution to these discrepancies through a fine-grained analysis of the regime previously considered unstable and therefore uninteresting. In particular, we show that, under the CE loss, the unstable regime comprises two distinct sub-regimes: a catastrophically unstable regime and a more benign controlled divergence regime, where logits diverge but gradients and activations remain stable. Moreover, at the edge of the controlled divergence regime, which corresponds to scaling the learning rate as $n^{-1/2}$ for deep MLPs under SP, there exists a well-defined infinite width limit where features continue to evolve in all hidden layers, which could partially explain the practical success of SP. To the best of our knowledge, this provides the first practical infinite-width limit in the feature-learning regime for SP.

(d) **Empirical validation.** We show that our width-scaling considerations provide surprisingly accurate predictions of maximal stable learning rate exponents, which often dominate optimal learning rates, particularly in Transformers (Vaswani et al., 2017). However, while output-layer divergence under CE loss remains benign with respect to the stability threshold, it occasionally influences the optimal learning rate choice. As one important example, previously observed learning rate transfer under layerwise learning rates breaks on all considered image datasets. Avoiding logit divergence while recovering feature learning with $\mu$P, on the other hand, opens up more loss functions such as MSE loss as competitive alternatives.

Taken together, our results deepen the theoretical understanding of why SP remains effective at large scales, provide thorough empirical validation for critical assumptions in infinite-width theory, and offer practical insights into stable hyperparameter transfer for scaling neural networks.

## 2  Background: Width-scaling arguments from Tensor Program theory

Before exploring plausible explanations for the empirical width-scaling properties of neural networks, we first define used notation and distill all necessary width-scaling arguments from Tensor Program (TP) theory (Yang and Hu, 2021, Yang and Littwin, 2023). We provide a more detailed introduction to TP scaling arguments in Appendix C.1, and a detailed account of related work in Appendix A.

**Setting and Notation.** We define an $(L+1)$-layer MLP of width $n$ iteratively via

$$h^1(\xi) := W^1\xi, \qquad x^l(\xi) := \phi(h^l(\xi)), \qquad h^{l+1}(\xi) := W^{l+1}x^l(\xi), \qquad f(\xi) := W^{L+1}x^L(\xi),$$

for inputs $\xi \in \mathbb{R}^{d_{in}}$ with trainable weight matrices $W^1 \in \mathbb{R}^{n \times d_{in}}$, $W^l \in \mathbb{R}^{n \times n}$ for $l \in [2, L]$, and $W^{L+1} \in \mathbb{R}^{d_{out} \times n}$. We call $h^l$ preactivations, $x^l$ activations, and $f(\xi)$ output logits. Training the MLP with Stochastic Gradient Descent (SGD) with global learning rate $\eta > 0$ under loss function $\mathcal{L} : \mathbb{R}^{d_{out}} \times \mathbb{R}^{d_{out}} \to \mathbb{R}$ with labelled training point $(\xi_t, y_t) \in \mathbb{R}^{d_{in}} \times \mathbb{R}^{d_{out}}$ is defined as $W_{t+1}^l = W_t^l - \eta\nabla_{W^l}\mathcal{L}(f_t(\xi_t), y_t)$. We denote updates accumulated over all time steps by $\Delta h_t^l = h_t^l - h_0^l$ and the change in a single update step by $\delta h_t^l = h_t^l - h_{t-1}^l$. The fan-notation has the purpose of unifying all weight matrices and simply means $W \in \mathbb{R}^{\texttt{fan\_out} \times \texttt{fan\_in}}$. In this paper, we define *standard parameterization (SP)* to mean He initialization $(W_0^l)_{ij} \sim N(0, c_\phi/\texttt{fan\_in}(W_0^l))$ trained with SGD or Adam with a single possibly width-dependent learning rate $\eta_n = \eta \cdot n^\alpha$, $\alpha \in \mathbb{R}$, for all trainable weights $\{W_t^l\}_{l \in [L+1]}$. This models the typical practice, in which a global learning rate is tuned at each model scale. We denote the softmax function by $\sigma(f)_i = \exp(f_i) \cdot (\sum_{j \in [d_{out}]} \exp(f_j))^{-1}$. In this paper, by CE loss, we refer to the concatenation $\mathcal{L} \circ \sigma$ of the cross-entropy loss function $\mathcal{L}(f, y) = -y \cdot \log(f)$ and the softmax $\sigma$, as is the dominant practice implemented in `torch.CrossEntropyLoss`. For naturally measuring the average scaling of entries in vectors $x \in \mathbb{R}^d$, we use the root-mean-squared norm $\|x\|_{RMS} := d^{-1/2} \cdot \|x\|_2$ as the standard vector norm. For matrices $W$, we write $\|W\|_F$ for the Frobenius norm and measure entry-wise scaling with the RMS norm $\|W\|_{RMS} := \left(\frac{1}{\texttt{fan\_in} \cdot \texttt{fan\_out}}\right)^{1/2}\|W\|_F$. The operator norm w.r.t. the RMS-norm is defined as $\|W\|_{op} := \|W\|_{RMS \to RMS} := \sup_{x \in \mathbb{R}^{\texttt{fan\_in}(W)}}(\|Wx\|_{RMS}/\|x\|_{RMS})$. We use Bachmann-Landau notation $\mathcal{O}, \Theta, \Omega$ that purely tracks dependence on width $n$ and omits all other dependencies.

**Effective and Propagating Updates.** When training neural networks, weights $W_t^l$ of layer $l$ evolve from their initialization $W_0^l$ through updates $\Delta W_t^l$, such that $W_t^l = W_0^l + \Delta W_t^l$. Although we directly control the scaling of these initial weights and updates, we are ultimately interested in their impact on subsequent activations in the network. For standard architectures, including convolutional networks and Transformers, weights typically act linearly on incoming activations. Thus, for weights $W_t^l$ and incoming activations $x_t^{l-1}$, the change in the next layer's pre-activations $\Delta h_t^l$ can be decomposed into two distinct contributions: the *effective updates* arising directly from the change in weights $\Delta W_t^l$ of the current layer, and the *propagating updates*, arising indirectly from activation changes $\Delta x_t^{l-1}$ in preceding layers:

$$\Delta h_t^l = \underbrace{(\Delta W_t^l)x_t^{l-1}}_{\text{Effective Updates}} + \underbrace{W_0^l(\Delta x_t^{l-1})}_{\text{Propagating Updates}}. \tag{RCC}$$

We say a layer admits *maximal stable feature learning* if both the effective updates and propagating updates remain width-independent as network width $n \to \infty$, that is $\|(\Delta W_t^l)x_t\|_{RMS} = \Theta(1)$ and $\|W_0^l(\Delta x_t^{l-1})\|_{RMS} = \Theta(1)$. This definition of feature learning as non-vanishing effective updates formulates a necessary but not sufficient condition for learning well-generalizing features at large width, common in related literature (Yang and Hu, 2021, Vyas et al., 2024, Bordelon et al., 2025).

**Identifying the correct scaling exponents.** In the spirit of Everett et al. (2024), we use $p_l$ and $q_l$ to denote the width-scaling exponents of the *alignment ratios* of the pairs $(\Delta W_t^l, x_t^{l-1})$ and $(W_0^l, \Delta x_t^{l-1})$ respectively, that is,

$$\frac{\|\Delta W_t^l x_t^{l-1}\|_{RMS}}{\|\Delta W_t^l\|_{RMS} \cdot \|x_t^{l-1}\|_{RMS}} = \Theta(n^{p_l}), \quad \frac{\|W_0^l\Delta x_t^{l-1}\|_{RMS}}{\|W_0^l\|_{RMS} \cdot \|\Delta x_t^{l-1}\|_{RMS}} = \Theta(n^{q_l}). \tag{$\alpha$-rms}$$

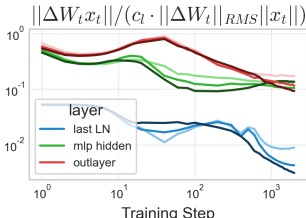 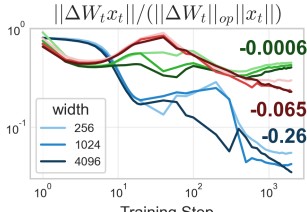 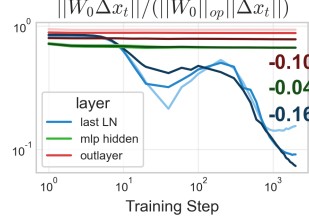

Figure 2: **Alignment has minimal width-dependence.** Alignment ratio between accumulated weight updates $\Delta W_t$ and incoming activations $x_t$ in RMS norm (left) and operator norm (center) as well as between initial weights $W_0$ and activation updates $\Delta x_t$ in operator norm (right) for the last layernorm layer, the first MLP layer in Transformer block 2 and the readout layer. RMS norm may be confounded by accumulated rank over the course of training (e.g. compare $(\Delta W_t, x_t)$ values for last LN). While operator norm alignment tends to decay over the course of training, it does not display strong width-dependence, even after 2000 batches (see annotated width-dependent exponents).

A key insight from Yang and Hu (2021) and Yang and Littwin (2023) is that during training, correlations can emerge in certain layers between the two quantities in each pair in (RCC), causing them to become *aligned* in the infinite-width limit and thereby inducing $p_l = 1$ and $q_l = 1$ due to a law of large numbers effect. If, instead, these quantities were uncorrelated, their product would exhibit smaller scaling exponents ($p_l = 1/2$ and $q_l = 1/2$) due to a central limit effect. In particular, infinite-width theory predicts the exponents $p_{1:L+1} = 1$, $q_{1:L} = 1/2$, and $q_{L+1} = 1$. The alignment exponents $p_l, q_l$ are a consequence of gradient-based training and do not depend on the specific parameterization used (e.g., SP, NTP, or $\mu$P).

By adjusting the initialization variance, which controls the scale of initial weights $W_0$, and the learning rate, which governs the magnitude of updates $\Delta W_t$, we can ensure that both contributions in (RCC) remain width-independent as the network width $n$ grows. The corresponding choice of hyperparameter scaling defines the *Maximal Update Parameterization* ($\mu$P). As we will discuss in Section 4, under the theoretically predicted alignment exponents, SP with $\mathcal{O}(1/n)$ learning rates leads to vanishing activation updates in all layers $\|\Delta x_t\|_{RMS} = o(1)$ and choosing the learning rate $\omega(1/n)$ leads to logit divergence in the infinite-width limit.

## 3 Finite-width distortions and long-training dynamics alone do not explain the stability of large learning rates in SP

Discrepancies between finite- vs infinite-width networks is often attributed to finite-width effects. Plausible explanations include dynamics induced by **(1) large depth or (2) longer training time** which may induce accumulation of finite-width effects over multiple steps (e.g., when $T > n$). Here, we show these explanations are insufficient. In Figure F.15, we find this discrepancy is surprisingly more pronounced in shallow (2-layer) MLPs in a single pass under SP. This finding—that the issue persists even without large depth or multi-epoch accumulation—motivates investigating other mechanisms. In this section, we explore two such explanations.

### 3.1 Update alignment between weights and activations is barely width-dependent

Everett et al. (2024) highlight that at finite width and over extended training times, it is a priori unclear whether the pairs $(\Delta W_t^l, x_t^{l-1})$ and $(W_0^{L+1}, \Delta x_t^L)$ remain strongly correlated or whether their alignment exponents $(p_{1:L+1}, q_{L+1})$ should rather be thought of as dynamically changing over the course of training. If the alignment exponents instead transition towards the central-limit regime and in particular if $p_{1:L+1} = 1/2$, this could explain the observed $\sqrt{n}$ gap between theoretically predicted and empirically observed optimal learning rate scalings.

Since our objective is measuring with which width-scaling $\Delta W_t^l$ propagates incoming activations $x_t^{l-1}$ forward, but rank accumulation can decouple the RMS norm $\|\Delta W_t^l\|_{RMS}$ from this objective, we measure alignment with operator norms (Yang et al., 2023a),

$$\alpha_{A,x} = \frac{\|Ax\|_{\text{RMS}}}{\|A\|_{\text{RMS}\to\text{RMS}} \cdot \|x\|_{\text{RMS}}}. \tag{$\alpha$-op}$$

Specifically, if the alignment exponents $p_{1:L+1} = 1$, $q_{1:L} = \frac{1}{2}$, $q_{L+1} = 1$ hold, both contributions in (RCC) must propagate signals forward maximally as a function of width with alignment metrics $\alpha_{\Delta W_t^l, x_t^{l-1}}$ and $\alpha_{W_0^l, \Delta x_t^{l-1}}$ of order $\Theta(1)$ unifying all layers. In Appendix C.2, we explain our alignment measurement considerations in more detail.

In Figure 2, we plot the alignment metrics at varying widths over the course of Transformer training with AdamW in SP. The figure shows that while alignment can decrease over the course of training, it exhibits minimal dependence on network width. Even after accumulating approximately 2000 batches of training, the width-scaling exponents are much closer to 0 than to $-0.5$, indicating that infinite-width alignment predictions hold reasonably well. Hence a lack of alignment alone cannot explain the large optimal learning rate exponents observed in practice.

### 3.2 Does a catapult mechanism in the first update steps stabilize large learning rates in SP?

As another plausible explanation, initial divergence under large learning rates may be stabilized over the course of training at finite width. Unlike at infinite width, where there only exist a divergent regime and a lazy regime without feature learning, an intermediate catapult regime was identified by Lewkowycz et al. (2020), for SGD training with MSE loss in Neural Tangent Parameterization (NTP) at finite width. They provide theory for 2-layer linear networks. Under small learning rates $\eta \leq 2/\lambda_0$, where $\lambda_0$ denotes the largest eigenvalue of the Hessian at initialization, the network monotonically converges to a minimum. Under large learning rates $\eta > 4/\lambda_0$, training diverges. But in an *edge of stability* regime (Cohen et al., 2021, 2022) of intermediate learning rates, the loss increases in the first $\mathcal{O}(\log(n))$ update steps while the sharpness $\lambda_t$ decreases. Once the sharpness lies below the edge of stability $2/\eta$, the loss decreases and the final learned function may generalize better as the solution lies in a basin with lower sharpness. But existing work does not study width-scaling with SP. May similar initial training dynamics be at play here?

In Appendix C.4 we analyse the 2-layer linear network model from Lewkowycz et al. (2020) in NTP, SP and $\mu$P trained with SGD under MSE loss, and provide loss and sharpness increase characterizations in Proposition C.18. In $\mu$P, the update equations of the learned function $f_t$ and the sharpness $\lambda_t$ are fully width-independent, which allows width-independent learning rates. In NTP, at least the conditions for loss and sharpness reduction are approximately width-independent. In SP, on the other hand, sharpness increases $\lambda_{t+1} \geq \lambda_t$ iff $\lambda_t \geq \frac{4}{n\eta_n}(1 + \frac{y}{f_t - y})$, requiring $\eta_n = \mathcal{O}(n^{-1})$ to avoid sharpness (as well as loss) divergence in the first update steps. The simulations shown in Figure C.2 validate the maximal stable learning rate scaling $\eta = \mathcal{O}(n^{-1})$. Hence catapult dynamics alone do not suffice for explaining large learning rate stability in SP.

## 4 Cross-entropy loss enables stable feature learning under large learning rates in standard parameterization

First, let us briefly recall why infinite-width theory predicts divergence under SGD training in SP with learning rates $\eta_n = \eta \cdot n^{-\alpha}$ for $\alpha < 1$.

Recall that the alignment exponents in ($\alpha$-rms) satisfy $p_{1:L+1} = 1$. In particular, for the output layer, we have $\|\Delta W_t^{L+1} x_t^L\|_{RMS} = \Theta(n \cdot \|\Delta W_t^{L+1}\|_{RMS} \cdot \|x_t^L\|_{RMS})$. For SGD, the weight update of the last layer after 1 update step is given by $\Delta W^{L+1} = -\eta \cdot n^{-\alpha} \cdot \chi_0 \cdot (x_0^L)^T$, where $\chi_0 := \partial_f \mathcal{L}(f_0(\xi_0), y_0)$. Under SP, at initialization, both $\|x_0^L\|_{RMS} = \Theta(1)$ and $\|\chi_0\|_{RMS} = \Theta(1)$. This implies logit divergence after 1 step of SGD with learning rates $\eta_n = \omega(1/n)$:

$$\|x^L\|_{RMS} = \Theta(1), \ \|\Delta W^{L+1}\|_{RMS} = \Theta(n^{-\alpha}), \implies \|\Delta W^{L+1} x^L\|_{RMS} = \Theta(n^{1-\alpha})$$

So, *why do larger learning rates remain stable and even effective, despite logit divergence?*

Here, we demonstrate that a simple yet fundamental aspect of training, the choice of loss function, resolves the large learning rate puzzle, and enables a well-defined and practical infinite-width limit that allows feature learning under SP. The key insight is that, under cross-entropy (CE) loss, the logits $f$ never directly appear in the training dynamics; instead, the effective output function is $\sigma(f)$. Unlike the destabilizing logit blowup encountered under mean squared error (MSE) loss, under CE loss, logit growth has a harmless effect on training stability. Therefore, CE loss introduces an intermediate *controlled divergence* regime that is absent for the MSE loss (Figure 3).

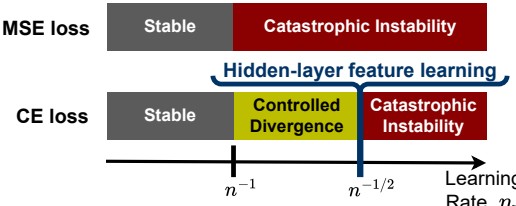

Figure 3: **Learning rate regimes for SGD in SP.** Under MSE loss, training a deep MLP either remains stable ($\alpha \geq 1$) or logits and hidden-layer activations diverge ($\alpha < 1$) in the infinite-width limit. Under CE loss, a *controlled divergence* regime $\alpha \in [1/2, 1)$ emerges where logits diverge, but training does not diverge. At $\alpha = 1/2$, hidden layers learn features width-independently.

**Definition 1** (**Learning regimes**). Fix $t \in \mathbb{N}$. We say that training lies in the *stable regime* iff the activations $\|x_t^l\|_{RMS}$ of all layers remain $\Theta(1)$ and the logits $\|f_t\|_{RMS}$ remain $\mathcal{O}(1)$ after $t$ update steps. We say that training is *catastrophically unstable* iff the logits and activations in at least one layer diverge after $t$ steps, that is $\|f_t\|_{RMS} \to \infty$ and $\exists\, l \in [L]$ such that $\|x_t^l\|_{RMS} \to \infty$. If training neither lies in the stable nor in the catastrophic regime, it lies in the *controlled divergence* regime. ◀

Proposition 2 shows that there exists a non-vanishing controlled divergence regime under the CE loss.

**Proposition 2.** *(Asymptotic regimes in SP, informal)* *For fixed $L \geq 2$, $t \geq 1$, $\eta > 0$, $\alpha \in \mathbb{R}$, consider training a $(L+1)$-layer MLP of width $n$ in SP with SGD and global learning rate $\eta_n = \eta \cdot n^{-\alpha}$ for $t$ steps. Then the logits $f_t$, loss-logit derivatives $\chi_t := \partial_f \mathcal{L}(f_t(\xi_t), y_t)$, loss-weight gradients $\nabla_t^l := \nabla_{W^l} \mathcal{L}(f_t(\xi_t), y_t)$ and activations $x_t^l$, $l \in [L]$, after training scale as follows as $n \to \infty$.*

***Under cross-entropy (CE) loss***, *three qualitatively distinct regimes arise:*

    (a) ***Stable regime*** *($\alpha \geq 1$): Logits, gradients and activations remain stable, that is*
$$\|f_t\|_{RMS} = \mathcal{O}(1),\ \|\chi_t\|_{RMS} = \mathcal{O}(1),\ \|\nabla_t^l\|_{RMS} = \mathcal{O}(n^{-1/2})\ \text{and}\ \|x_t^l\|_{RMS} = \Theta(1)\ \text{for all } l \in [L].$$

    (b) ***Controlled divergence*** *($\frac{1}{2} \leq \alpha < 1$): Logits diverge $\|f_t\|_{RMS} = \Theta(n^{1-\alpha})$, but gradients and activations remain stable, that is $\|x_t^l\|_{RMS} = \Theta(1)$, $\|\chi_t\|_{RMS} = \mathcal{O}(1)$ and $\|\nabla_t^l\|_{RMS} = \mathcal{O}(n^{-1/2})$ for all $l \in [L]$.*

    (c) ***Catastrophic instability*** *($\alpha < \frac{1}{2}$): Logits, activations and weight gradients diverge, that is $\|f_t\|_{RMS} \to \infty$, $\|x_t^l\|_{RMS} \to \infty$ and $\|\nabla_t^l\|_{RMS} \to \infty$, $l \in [2, L]$.*

***Under mean-squared error (MSE) loss***, *a stable regime as in (a) above arises if $\alpha \geq 1$. If $\alpha < 1$, training is catastrophically unstable as in (c) above and, in addition, $\|\chi_t\|_{RMS} \to \infty$.*

**Remark 3** (**Maximal-stable learning rate**). In SP, different layer types learn at different rates. For SGD, the output layer logits remain stable iff $\eta_n = \mathcal{O}(n^{-1})$. The input layer, biases, Layernorm gains and embedding layers all behave input-like and remain stable iff $\eta_n = \mathcal{O}(1)$. All other trainable weights in typical CNNs and Transformers behave hidden-like and remain stable iff $\eta_n = O(n^{-1/2})$.

If we define the maximal stable learning rate (max-stable LR) scaling of a network as the edge to the catastrophic regime, and the max-stable LR scaling of each layer as that above which the layer's output diverges (Definition C.11), then Proposition 2 shows that the max-stable LR of a network can exceed that of individual layers. Since CE loss does not require output layer stability, the proposition naturally extends to 2-layer MLPs, which do not contain a hidden layer, so that their controlled divergence regime under CE loss extends to $0 \leq \alpha < 1$ (Remark C.14). ◀

The formal statement together with a proof can be found in Appendix C.3. For an intuitive understanding of this result, note that the only effect that the choice of loss function $\mathcal{L}(f, y)$ has on the final learned function is through the loss-logit gradients $\chi_t := \partial_f \mathcal{L}(f_t(\xi_t), y_t)$ over the course of training. Under MSE loss, the loss gradients are given by the residuals $\chi_t = f_t(\xi_t) - y_t$. But CE loss induces loss gradients $\chi_t = \sigma(f_t(\xi_t)) - y_t$. Crucially, it is the correct choice of loss function to effectively view $\sigma(f)$ as the output of the network instead of the unnormalized logits $f$. If one were to use $\text{MSE}(\sigma(f), y)$ as a loss function instead, additional derivative terms can induce vanishing gradients under exploding network output and not increase the optimal learning rate exponent (Appendix F.4). Under CE loss, the effective network output $\sigma(f)$ at most converges to one-hot predictions when the logits diverge, and with increasing width training points are sharply memorized after a single update step. At large learning rates $\eta_n = \Theta(n^{-1/2})$, training points are not just memorized in last-layer weights, but feature learning is recovered in the infinite-width limit:

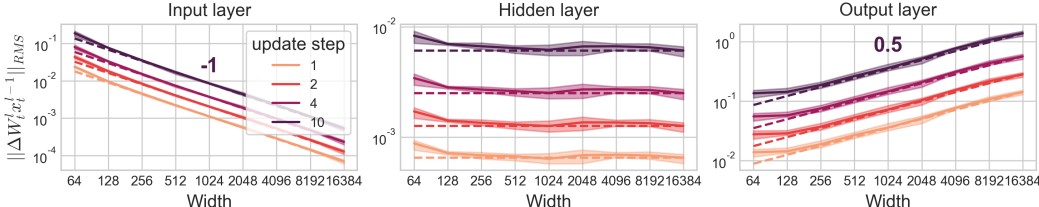

Figure 4: **Hidden-layer feature learning albeit logit divergence in SP under large learning rates.** Effective $l$-th layer update scalings $\|\Delta W_t x_t\|_{RMS}$ of MLPs trained with SGD in SP with $\eta_n = 0.0001 \cdot (n/256)^{-1/2}$ on CIFAR-10 under CE loss. Our TP scaling predictions are accurate: Hidden layers learn features width-independently, and input layers have vanishing feature learning. *The update scaling exponents can already be accurately estimated at small width $n \leq 512$.*

**Proposition 4** (**Under CE loss, SP with large learning rates learns features at large width, informal**). *Consider the setting of Proposition 2 of training a (L+1)-layer MLP with SGD in SP with global learning rate $\eta_n = \eta \cdot n^{-\alpha}$, $\alpha \in \mathbb{R}$, in the infinite-width limit $n \to \infty$.*

  (a) *Under both MSE and CE loss in the stable regime ($\alpha \geq 1$), feature learning vanishes in all layers $l \in [L]$, that is $\|\Delta x_t^l\|_{RMS} = \mathcal{O}(n^{-1/2})$.*
  (b) *Under CE loss in the controlled divergence regime ($\frac{1}{2} \leq \alpha < 1$), input layer feature learning vanishes at rate $\|\Delta x_t^1\|_{RMS} = \Theta\left(n^{-1/2-\alpha}\right)$, and hidden layers $l \in [2, L]$ learn features at rate $\|\Delta x_t^l\|_{RMS} = \Theta\left(n^{1/2-\alpha}\right)$. In particular, when $\alpha = 1/2$, the weight updates of all hidden layers induce width-independent activation updates, that is $\|\Delta x_t^l\|_{RMS} = \Theta\left(1\right)$.*

To the best of our knowledge, this provides the first infinite-width limit of SP in the practical feature learning regime. Figure 4 empirically validates that the predicted width-scaling exponents that induce maximally stable feature learning despite logit blowup under $\eta_n = \eta \cdot n^{-1/2}$ are already accurate at moderate width 512. Appendix E.4 shows that effective update predictions also hold accurately in Transformers trained with Adam. In the next section, we discuss the implications that training stability despite logit blowup has on learning rate scaling exponents in practice.

## 5 Consequences of training stability under logit divergence

In this section we perform extensive experiments to empirically evaluate the implications of the stability and feature learning predictions of our infinite-width theory from the previous section.

**Experimental details.** We train MLPs of varying depth up to width 16384 with plain SGD and Adam on CIFAR-10, MNIST and a generated multi-index model (reported in Appendix F). We also train Pythia-GPTs with warmup and cosine learning rate decay on the DCLM-Baseline dataset (Li et al., 2024) up to width 4096 or 1.4B parameters using both Adam with decoupled weight decay (Loshchilov and Hutter, 2019) and SGD (reported in Appendix F.2). If not stated otherwise, we consider SP with a global learning rate. In this paper, we train for a single epoch to prevent overfitting effects. We leave a systematic study of the multi-epoch setting to future work. All details can be found in Appendix D. Open-source code to reproduce our experiments is publicly available.

**Refined coordinate checks.** Our results demonstrate that disentangling the width dependence of effective updates and propagating updates in each trainable weight in refined coordinate checks (RCC) can serve as an useful diagnostic tool for understanding and correcting layerwise update signal propagation, for improving training stability and performance at scale. Our implementation is easy to adapt and publicly available.

### 5.1 Infinite-width theory is a useful predictor of empirical optimal learning rate exponents

While, in general, the optimal learning rate depends on the architecture and dataset (see Appendix F.1), it often saturates at the accurately predicted maximal stable learning rate in deep non-linear networks. We hypothesize that maximal stable feature learning in all layers induces optimal performance at large width. However, since different layer types require different maximal stable learning rate exponents, the single global learning rate under SP is subject to opposing forces for recovering feature learning under the constraint of training stability. We now evaluate several instantiations of this hypothesis.

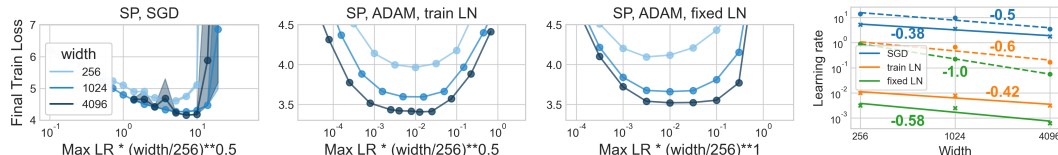

Figure 5: **Approximate learning rate transfer for GPT in SP.** *Left to center-right:* Width-scaled learning rate versus training loss for GPT trained with SGD, Adam with trainable Layernorm parameters and Adam without trainable Layernorm parameters. *Right:* Corresponding optimal (solid) and maximal stable (dashed) learning rate exponents. For SGD, hidden-layer stability $\eta_n = \mathcal{O}(n^{-1/2})$ clearly dominates the maximal stable as well as optimal learning rate scaling. For Adam without Layernorm parameters, hidden-layer stability induces a stability threshold $\eta_n = \mathcal{O}(n^{-1})$. Trainable Layernorm parameters further stabilize large learning rates and induce larger optimal learning rate scaling $\eta_n \approx \Theta(n^{-1/2})$ toward preserving input-layer feature learning at scale.

**MLPs and Transformers with SGD.** Figures 5 and 6 show that the empirical maximal learning rate exponents under CE loss closely follow $\alpha = 1/2$ for both MLPs on vision data and for GPT on language data. The x-axes scale the learning rate with the closest clean exponent from $\{0, 0.5, 1\}$ to show that approximate empirical transfer is often enforced by the stability threshold $\mathcal{O}(n^{-1/2})$. While the theory only predicts the maximal stable exponent, Proposition 4 suggests that the optimal learning rate may follow the maximal stable exponent $\alpha = 1/2$ since it is the only choice under which feature learning is preserved at large width in all hidden layers. When optimal learning rate exponents exceed the maximal stable exponents, as for CIFAR-10, optimal learning rates saturate at the maximal stable learning rate at realistic width $\geq 16384$ on all considered datasets (see Figure F.11). The maximal stable learning rate under MSE loss also consistently scales as its infinite-width prediction $\mathcal{O}(n^{-1})$ and optimal learning rates closely follow this exponent, as under smaller exponents $\alpha > 1$, not even logits are updated $\|\Delta f_t\|_{RMS} \to 0$. However, this approximate learning rate transfer under MSE loss is not useful, since the loss also converges fast and does not monotonically improve with scale. Overall, this shows that existing infinite-width theory was indeed predictive of the maximal stable learning rate exponents under MSE loss, but that CE loss induces qualitatively more favorable behaviour that is only captured by a finer-grained analysis.

**MLPs with Adam.** Adam approximately normalizes the gradient and therefore further stabilizes training against misscaled gradients beyond the effect of CE loss. $W^l$ is effectively updated if the learning rate scaling counteracts the scaling accumulated in the inner product between normalized weight gradients and incoming activations. This leads to the ideal ($\mu$P) learning rates $\eta(W^l) = \eta/\texttt{fan\_in}(W^l)$. Thus Adam in SP with $\eta_n = \Theta(n^{-1})$ induces width-independent updates, except for vanishing input layer feature learning and logit divergence through $W_0^{L+1}\Delta x_t^L$. For deep MLPs on image datasets, Figure F.25 shows optimal learning rate exponents close to $\eta_n = \mathcal{O}(n^{-1})$ for both CE and MSE loss, suggesting a controlled divergence regime also arises from a stabilized backward pass and that stable hidden-layer feature learning dominates the optimal learning rate scaling.

**Transformer training with AdamW.** In Transformers with trainable Layernorm parameters, which scale input-like, training is stabilized, and the exponent is increased toward input layer feature learning. Without trainable Layernorm parameters, in contrast, only the embedding layer scales input-like so that training becomes approximately width-independent under $\eta_n = \Theta(n^{-1})$. Figure 5 shows that the max-stable and optimal learning rate exponents shrink from $-1/2$ toward $-1$ if we remove the trainable layer-norm parameters. This suggests that trainable scale parameters in normalization layers play an essential role in maintaining high learning rates in Transformers, which could explain why they are almost unanimously used in modern architectures (OLMo Team et al., 2024, Grattafiori et al., 2024, Gemma Team et al., 2024). Moreover, input layer learning vanishes at scale in SP, which may explain techniques like removing weight decay in the embedding layer (OLMo Team et al., 2024). Logit divergence under large learning rates may be a reason for regularizing techniques like the z-loss (Chowdhery et al., 2023, Wortsman et al., 2024, OLMo Team et al., 2024).

**Taken together, our empirical evidence suggests that infinite-width theory may serve as a helpful proxy for understanding practical neural networks at finite width.** Since training divergence imposes a hard constraint on the optimal learning rate and activation divergence in multiple layers becomes harder to stabilize, width-scaling predictions seem to hold even more accurately on deep and sensitive architectures such as Transformers.

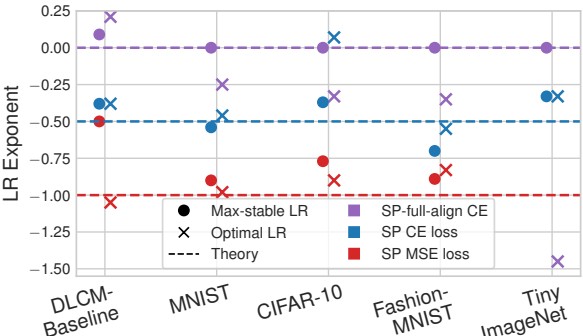

Figure 6: **LR exponents at the edge of controlled divergence.** Closest clean exponent of max-stable LR follows theoretical prediction $-1$ for SP under MSE loss, $-0.5$ for SP under CE loss and $0$ for SP-full-align under CE loss. Max-stable LR often dominates optimal LR at sufficient width in SP. Corresponding learning rate curves are provided in Appendix F.2 for GPT and in Appendix F.3 for MLPs.

## 5.2 A novel understanding of standard initialization with layerwise learning rates

Everett et al. (2024) perform extensive Transformer experiments, and recommend training with Adam in SP with $\mu$P learning rates (SP-full-align) as the overall best performing parameterization in terms of validation loss, learning rate transfer and learning rate sensitivity. This parameterization only differs from $\mu$P through the larger last-layer He initialization $W_0^{L+1} \sim N(0, n^{-1})$. While the authors attribute the success of SP-full-align to a lack of alignment between $W_0^{L+1}$ and $\Delta x_t^L$, they only measure the joint alignment between $W_t$ and $x_t$ for each layer, which confounds the individual alignment exponents of $(\Delta W_t, x_t)$ and $(W_0, \Delta x_t)$ from (RCC). We provide a detailed explanation in Appendix C.2. Our empirical alignment reevaluation in Figure 2 and Appendix E.4 does not support the hypothesized lack of alignment. Thus, at sufficient width, logits diverge through $W_0^{L+1}\Delta x_t^L$ as soon as feature learning does not vanish. Instead our theoretical results in Section 4 show that logit divergence does not harm training stability under CE loss. Just like SP with $\eta_n = \Theta(n^{-1/2})$, SP-full-align with $\eta_n = \Theta(1)$ lies at the feature learning edge of the controlled divergence regime.

**Learning rate transfer of SP-full-align breaks on image datasets.** Due to width-independent alignment between $W_0^{L+1}$ and $\Delta x_t^L$, logits diverge with width in SP-full-align at sufficient width. We validate this claim for CIFAR-10 at moderate width in Figure F.32. This introduces width-dependent training dynamics. Consequently our single-pass experiments in Figure 6 and Appendix F.7 consistently show decaying optimal learning rates in SP-full-align for both SGD and Adam on common image datasets and generated multi-index data. We also observe that the maximal stable learning rate remains remarkably width-independent as our theory would predict. This constitutes our only experiment in which the maximal stable learning rate scaling is suboptimal in deep nonlinear networks. We leave fully understanding the driving mechanism to future work.

**Large output dimension may explain learning rate transfer of SP-full-align on language data.** On language data, Everett et al. (2024) report learning rate transfer of SP-full-align. To understand this difference to the vision settings above, we measure the individual contributions to the only source of width dependence $W_0^{L+1}\Delta x_t^L$ in this parameterization.

By empirically verifying the predicted width-independent incoming updates $\|\Delta x_t^L\|_{RMS} = \Theta(1)$ and alignment ratio $\alpha_{W_0^{L+1}\Delta x_t^L} = \Theta(1)$ from ($\alpha$-op) (Figures E.17 and E.18), the scaling of the propagating updates $\|W_0^{L+1}\Delta x_t^L\|_{RMS}$ is determined by the initial operator norm

$$\|W_0^{L+1}\|_{RMS \to RMS} \approx d_{\text{out}}^{-1/2}(d_{\text{out}}^{1/2} + n^{1/2}) = 1 + (n/d_{\text{out}})^{1/2},$$

under standard initialization (Vershynin, 2010). Now, in the large width regime $n \gg d_{\text{out}}$ the second term dominates, but in the regime $d_{\text{out}} \gg n$ the first term dominates and induces an approximately width-independent effect on the logits (verified in Figure E.17). From this perspective, standard initialization is the approximately correct initialization in the regime $d_{\text{out}} \gg n$, but transfer should eventually break at sufficient width $n \approx d_{\text{out}}$ as logits start to diverge.

**Initialization for transfer at all scales.** To ensure that updates remain non-asymptotically scale-preserving, the above arguments motivate choosing the last-layer initialization variance $\sigma_{L+1} = \left(\frac{\text{fan\_in}}{\sqrt{\text{fan\_out}}} + \sqrt{\text{fan\_in}}\right)^{-1}$, which transitions from SP to $\mu$P with increasing width.

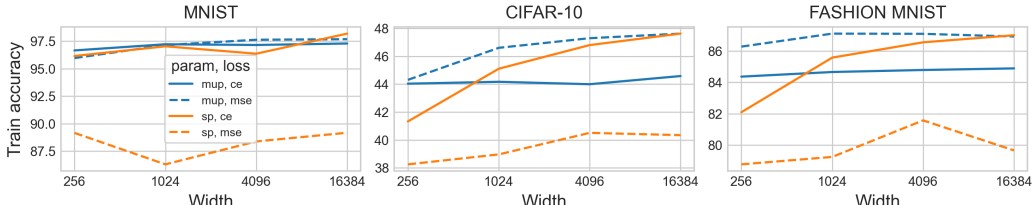

Figure 7: $\mu$**P enables other loss functions to be competitive.** Optimal training accuracy of 8-layer MLPs trained with SGD on MNIST (left), CIFAR-10 (center) and FASHION MNIST (right). In SP, feature learning is lost under MSE loss, inducing highly suboptimal performance. The layer-balanced learning in $\mu$P allows loss functions beyond CE loss to perform well.

### 5.3 A scaling-theoretic view on the practical success of CE loss in deep learning

Many success stories in deep learning, from computer vision to natural language processing, use the cross-entropy loss. We propose a scaling-theoretic explanation for this practical dominance. Our results show that networks trained under CE loss allow stable optimization at significantly larger learning rates in SP than under MSE loss, which recovers feature learning at large widths and consequently improves generalization. To empirically investigate this hypothesis, we compare the performance of CE and MSE losses under both SP and $\mu$P. Since $\mu$P admits asymptotically stable dynamics, both losses exhibit similar limiting behaviours. Thus we predict that CE loss only significantly outperforms MSE loss in SP, but not in $\mu$P. Figure 7 confirms this prediction, which suggests that MSE loss may deserve renewed consideration as a practical choice under stable parameterizations like $\mu$P, especially given its theoretical simplicity and widespread use in theoretical analyses.

## 6 Discussion and future work

On the theoretical side, we have provided the first infinite-width proxy model for finite neural networks, as they are initialized and trained in practice. On the practical side, we have seen that infinite-width feature learning and instability predictions are surprisingly predictive indicators for empirical width-scaling exponents, in particular for deep Transformers.

**Understanding of the controlled divergence regime.** As practical neural networks operate at the edge of the controlled divergence regime, better understanding parameterizations beyond the stable regime is paramount. We believe that many conclusions in previous work about the properties of wide networks in SP or neural tangent parameterization (Lee et al., 2020, Wenger et al., 2023) change when studying the optimal learning rate scaling at the edge of the controlled divergence regime. Since the NTK diverges in SP with $\eta_n = \Theta(n^{-1/2})$, studying this limit is subtle. However, investigating the rescaled NTK might still be a useful tool in better understanding this limit. While width dependence is undesirable from a transfer perspective, fast memorization under logit blowup may improve learning speed. How is generalization affected? Logit blowup may partially explain overconfidence in neural networks in SP, and suggests that wide networks in $\mu$P may be more calibrated.

**Numerical considerations.** In this paper, we consider the regime of sufficient numerical precision. From a numerical perspective, signals that diverge fast can leave floating point range at moderate widths. Hence implementations that ensure minimal accumulation of width-dependent factors in SP akin to Blake et al. (2025) could stabilize large-scale model training in practice.

**Understanding optimal learning rate exponents.** The exact conditions that induce hyperparameter transfer are still poorly understood. Without full width-independence, the optimal learning rate scaling cannot be predicted with certainty, and rigorous statements about optimal LR exponents likely require strong architectural and distributional assumptions, akin to neural scaling laws (Hoffmann et al., 2022, Bachmann et al., 2023) (see Appendix F.1 for more details). Both vanishing feature learning in input-like layers and logit divergence can induce strong finite-width effects, so that we would still recommend $\mu$P learning rates over SP from a width-scaling perspective. Similar to CE loss, normalization layers correct scaling in the forward pass. In combination with Adam which stabilizes the backward pass (Figure F.25), such stabilizing components can correct most misscaled signals. Deeply understanding their interplay and effect on optimal learning rates remains an important direction for future work.

## Acknowledgments and Disclosure of Funding

This work has been supported by the German Research Foundation through the Cluster of Excellence "Machine Learning - New Perspectives for Science" (EXC 2064/1 number 390727645). The authors thank the International Max Planck Research School for Intelligent Systems (IMPRS-IS) for supporting Moritz Haas. Leena Chennuru Vankadara is supported by the Gatsby Charitable Foundation.

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

# Appendices

**Appendix Contents.**

# A Detailed Related Work

Here we provide a more detailed account of related work than what is possible in the main body of the paper.

Strongly related, Atanasov et al. (2025) study the 2D plane of learning rate and feature learning strength, meaning the output multiplier, in $\mu$P and find a related pseudo-catapult regime in CE but not in MSE loss at fixed width. They study a simplified one-parameter model and provide several insightful empirical evaluations in MLPs and CNNs. Their results highlight the importance of tuning the 'feature learning strength' multiplier in the output layer, which can also be interpreted as the softmax temperature (Agarwala et al., 2023). As SP and NTP are width-dependent parameterizations, the feature learning strength is implicitly altered when scaling width.

**Neural networks in the infinite-width limit.** Past work has extensively analysed the *Neural Tangent Parameterization (NTP)* (Jacot et al., 2018) due to its tractability. But due to lacking feature learning in the infinite-width limit, finite networks in NTP behave qualitatively differently and hence NTP is not the ideal model for understanding finite neural networks. Finite-width deviations already accumulate after a few steps of training (Wenger et al., 2023), in particular under CE loss (Yu et al., 2025). Much feature learning has been empirically shown to improve generalization over the infinite-width kernel regime Chizat et al. (2019), Lee et al. (2020), Agarwala et al. (2023). Considerable effort has been invested in finding a descriptive infinite-width model for SP. Sohl-Dickstein et al. (2020) note that the NTK diverges under large learning rates $\eta_n = \omega(n^{-1})$ in SP, which motivates them to consider a different parameterization which preserves a finite NTK in the infinite-width limit, but consequently does not correspond to SP anymore. Golikov (2020) studies a class of 'dynamically stable' parameterizations, allowing large learning rates under a variant of SP, they call 'sym-default' parameterization, which again is not equivalent SP. Another popular width-dependent parameterization is the Maximal Update Parameterization ($\mu$P). It achieves a width-independent effect of updates in all trainable weights on the output function. Its infinite-width limit has been observed to closely track finite networks in $\mu$P well over long periods of training in, for example, feature learning strength, the learned function, or gradient and Hessian statistics (Vyas et al., 2024, Noci et al., 2024b). As an important practical consequence, it allows to tune small proxy models and train the large model only once with the optimal HPs (Yang et al., 2022). $\mu$P was derived using *Tensor Programs (TP)* framework (Yang, 2019, Yang and Hu, 2021, Yang and Littwin, 2023) that, in theory, allows to exactly track the learning dynamics of many popular architectures like MLPs, ResNets and Transformers trained with SGD or Adam in arbitrary parameterizations in the infinite-width limit. Haas et al. (2024) derive a $\mu$P-like parameterization for sharpness aware minimization algorithms achieving transfer of the optimal learning rate and perturbation radius jointly by showing that perturbations should be scaled like updates in $\mu$P. Vankadara et al. (2024) derive an initialization and learning rate scaling rule that achieves width-independent training dynamics for the state-space model Mamba, which shows that the spectral condition on the weights and weight updates in every layer for achieving $\mu$P provided by Yang et al. (2023a) does not apply to arbitrary architectures. At sufficient numerical precision, the mean-field parameterization (Mei et al., 2018, Chizat and Bach, 2018) is equivalent to $\mu$P. While it was initially restricted to shallow neural networks, the dynamical mean-field theory (DMFT) by Bordelon and Pehlevan (2022) generalizes it to more complex architectures, including Transformers (Bordelon et al., 2024a). Although still expensive, the approximate solvers from DMFT are more computationally feasible than iteratively solving the exact TP limit equations. Chizat et al. (2024) studies deep linear networks in $\mu$P and shows convergence of gradient flow to a minimum $l_2$-norm solution.

**Other neural network scaling limits.** Beyond width scaling, depth scaling $L \to \infty$ has been studied in detail. For ResNets, Yang et al. (2023b), Hayou and Yang (2023), Bordelon et al. (2024b) show that $L^{-1/2}$-scaling of shallow residual blocks induces depth-independence and this limit commutes with width scaling, implying that depth can be scaled independent of width. Using approximative DMFT theory, Bordelon et al. (2024a) suggest that $L^{-1}$-depth scaling may be necessary to preserve feature learning in attention blocks although they consider a pure depth limit. Dey et al. (2025) confirm $L^{-1}$-block scaling to be the 'correct' scaling by providing additional desiderata and empirical evidence on Transformers. Bordelon et al. (2024a) also show that the infinite within-head dimension limit effectively leads to a single-head Transformer, an the infinite number of heads limit concentrates by aggregating over the coordinate distribution at fixed within-head size, closer to how scaling is typically performed in practice (Brown et al., 2020). Noci et al. (2024a) study a joint width and depth limit close to initialization for Transformers with the goal of preventing rank collapse. Long

training time is much less understood. Bordelon and Pehlevan (2025) study the training dynamics of deep and wide linear networks trained on structureless Gaussian data. Chizat and Netrapalli (2024) considers the angle between activations and gradients to give scaling rules for hyperparameters toward automatic HP scaling. They correct output layer scaling of MLPs in $\mu$P depth-dependently, only for SGD.

**Scaling laws.** Robust compute-optimal scaling laws in LLMs were reported by Kaplan et al. (2020), Hoffmann et al. (2022). Paquette et al. (2024) provide theory on random feature models trained with one-pass SGD and identify 4 phases and 3 subphases depending on properties of the data and the target. Bjorck et al. (2025) observe no transfer across token horizons but a predictable scaling law with exponent $-0.32$ on LLama. McCandlish et al. (2018) suggests that the optimal learning rate scales as $a/(1 + b/batchsize)$ with setting-dependent constants $a, b$. Hence for sufficiently large batch size the optimal learning rate is roughly constant, which is in line with the empirical observations by Shallue et al. (2018), Yang et al. (2022). Ren et al. (2025) study SGD training of 2-layer MLPs on isotropic Gaussian data under MSELoss and find that different teacher neurons are abruptly learned at different timescales leading to a smooth scaling law in the cumulative objective. Further work toward assessing the compute-optimal and data-optimal Pareto frontiers under realistic assumptions remains an important and challenging task for future work.

**Finite width training dynamics.** Understanding finite-width training dynamics complements infinite-width theory very well, as the former line of work operates at fixed width, while the latter ask what changes with increasing width. From a practical perspective, scaling networks with $\mu$P appears to preserve the properties from base width (Vyas et al., 2024, Noci et al., 2022). Deep understanding of neural network training dynamics is still limited to 2-layer nonlinear MLP (Ren et al., 2025, Zhang et al., 2025) or (deep) linear MLP (Kunin et al., 2024, Tsigler et al., 2025) toy models under strong distributional assumptions. Lee et al. (2020) find that large learning rates cause differences between finite and infinite-width networks.

Kunin et al. (2024) explain for 2-layer networks that varying layerwise initialization variance and learning rate scaling induces differing learning regimes: fast feature learning in balanced parameterizations (desirable for linear nets), faster learning of earlier layers in upstream parameterizations with small parameter movement (desirable in nonlinear networks, as it reduces time to grokking and sample complexity of hierarchical data structures), faster learning of later layers in downstream initializations (that is initial lazy fitting followed by slow feature learning). Abbe et al. (2023) show that, opposed to lazy networks, feature learning networks can learn low rank spikes in hidden layer weights/kernels to help with sparse tasks. Qiao et al. (2024) show that large learning rates induce sparse linear spline fits in univariate gradient descent training by showing that all stable minima are flat, non-interpolating and produce small first order total variation, hence avoid overfitting and learn functions with bounded first order total variation

**Edge of stability.** Large learning rates have broadly been observed to induce optimal generalization. While the empirical connection between optimal and max-stable learning rates is well-documented, the mechanisms remain poorly understood, independent of model scale. Our findings offer a novel perspective on the advantages of large learning rates for facilitating feature learning at large model scales. Previously suggested explanations include improved generalization through reduced sharpness (Andriushchenko et al., 2023a), a shift in the learning order of patterns (Li et al., 2019), enhanced SGD noise (Keskar et al., 2017), and implicit bias towards sparsity (Andriushchenko et al., 2023b).

Lewkowycz et al. (2020) observe that under large learning rates at the edge of stability, $2/\lambda_0 < \eta < c_{arc}/\lambda_0$ (where $c_{arc} = 12$ for ReLU nets) an initial blowup at training time at least $\log(n)$ induces a bias towards flatter minima. Cohen et al. (2021) find loss spikes during training, but that training self-stabilizes through sharpness reduction. Damian et al. (2023) and (Cai et al., 2024) develop some understanding of the mechanisms behind EOS dynamics. For Adam, the preconditioner matrix provides an additional mechanism by which stabilization can occur (Cohen et al., 2022, Gilmer et al., 2022).

**Warmup.** Warmup allows stability under larger learning rates via slow sharpness reduction (Kalra and Barkeshli, 2024). Kalra and Barkeshli (2024) also shows that warmup allows using larger learning rates than otherwise stable by constantly operating at the edge of stability. Warmup does not improve performance but stabilizes training; by allowing training with larger learning rates, these often induce improved performance. Large catapults harm Adam by persisting in its memory. Hence Adam's optimal learning rate is further away from the failure boundary than for SGD, and Adam benefits

more from longer warmup. Above the optimal learning rate, Adam has a regime of training failure, where early catapults persist in the second moment and prevent learning. Warmup also widens the regime of near-optimal learning rate choices. Liu et al. (2020) find that particularly Adam needs warmup due to large initial variance.

**Effective learning rates.** Kosson et al. (2024) study the effect of weight decay on rotation in weight vectors, which influences the effective learning rate. Also see references therein for literature on effective learning rates, which is related to the alignment discussion in this paper.

**Stability of Transformer training.** More empirically, a plethora of works study the training stability of large-scale Transformers with respect to warmup, weight decay (D'Angelo et al., 2024), batch size (You et al., 2020), the optimizer (Kosson et al., 2024), the position of normalization layers (Xiong et al., 2020) and their interplay with the parameterization and numerical considerations (Wortsman et al., 2024, Blake et al., 2025, Everett et al., 2024). Wortsman et al. (2024) find that qk-Layernorm stabilizes Transformer training beyond the stabilizing effect from using $\mu$P. Xiong et al. (2020) propose pre-LN for enhanced training stability requiring less warmup. He et al. (2024) observe that outlier features (=extremely activated coordinates in activations) emerge quickly in Transformer training with AdamW and that rank-collapse under strong correlations between inputs is correlated with more outlier features. Non-diagonal preconditioning like SOAP and Shampoo resolves the issue.

Srećković et al. (2025) find that SGD with momentum almost performs en par with Adam in LLMs under small batch size, so that understanding small-batch SGD (as we do) is practically relevant.

Most relevant to our work, Everett et al. (2024) perform extensive and insightful experiments for Nan-oDO decoder-only Transformers (Liu et al., 2024) in SP, $\mu$P, NTP and mean field parameterizations with corrected layerwise learning rate scalings, questioning the infinite-width alignment predictions between weights and incoming activations at finite width over the course of long training. They recommend SP with ADAM in conjunction with $\mu$P-learning rate scaling (they call SP-full-align) as the best-performing empirical parameterization in terms of generalization, learning rate transfer and learning rate sensitivity.

# B Take-aways for practitioners

## B.1 A practitioner-oriented introduction to width scaling

In this paper, the term *parameterization* refers to width-dependent scaling of initialization and learning rate of each trainable weight tensor. Studying parameterizations then means applying a scaling rule for layerwise initialization variances and learning rates and understanding how relevant quantities such as update scaling in activations and logits evolves, and where instabilities may arise at large widths. At some fixed base width, all parameterizations can be considered equivalent, if we allow tuning constant multipliers.

For properly comparing the performance of parameterizations, constant weight and initialization multipliers should be tuned at some fixed base width (Yang et al., 2022). This adjusts the layerwise activation and gradient size at finite width for all parameterizations at once. The parameterization then prescribes the rule, by which the layerwise initialization and updates are rescaled when changing the width in relation to that base width `width/base_width`. Alternatively, multipliers can be tuned at larger width for each parameterization separately, but this quickly becomes computationally infeasbile. The extensive LLM experiments in Everett et al. (2024) suggest that the advantage of large last-layer initialization may just be an artifact of the community extensively tuning performance in SP, and after also tuning all layerwise multipliers for $\mu$P, the performance difference vanishes.

While SP performs better than naive theory would predict, and can learn hidden-layer features width-independently under CE loss, feature learning still vanishes in input-like layers like embedding or Layernorm layers under both SGD and Adam. Still only $\mu$P learning rate scaling effectively updates all layers. For AdamW without using width-dependent weight multipliers, layer-balancing $\mu$P learning rates are simply given by the learning rate scaling $\eta(W) = \eta/\texttt{fan\_in}(W)$. Here, all biases as well as normalization layer weights should be understood as weights to the one-dimensional input 1, hence $\texttt{fan\_in} = 1$. For recovering width-independent weight decay, weight decay requires the inverse scaling $\texttt{wd} \cdot \texttt{fan\_in}(W)$.

TP-like width scaling arguments are very useful for identifying sources of divergence or shrinkage with scale, and architecture components such as normalization layers and training algorithms such

as Adam correct most *but not all* divergent or vanishing scalings in the forward and backward pass, respectively. Of particular importance for evaluating the width-dependent signal propagation is the refined coordinate check (RCC) for disentangling effective updates in the current layer from updates propagating forward through the network. Ideally, all $W_0 \Delta x_t^L$ and $\Delta W_t x_t$ should remain width-independent, which is only guaranteed in $\mu$P at sufficient width.

## B.2 Practical implications of our results

**Constraining the search space for the optimal learning rate.** Our experiments strongly support the prediction from Proposition 2 that the maximal stable learning rate scales as $\eta_n = \eta \cdot n^{-1/2}$ when training deep non-linear networks with SGD in SP under CE loss. This result provides a concrete constraint on the learning rate search space, significantly reducing the range practitioners must consider. Rather than exhaustive or heuristic searches, practitioners may narrow their search around the theoretically justified scaling, leading to substantial computational savings and more efficient hyperparameter tuning. Furthermore, we observe that, particularly in deep nonlinear architectures, the optimal learning rate often saturates at this maximal stable scaling (see e.g. Figures 5 and F.10, Figures F.4 and F.5, Appendix F.3). Thus, employing the correct scaling often effectively enables approximate transfer of optimal learning rates, even in SP.

**Potential performance gains at scale using alternative loss functions with $\mu$P.** Correctly attributing the observed performance gap between MSE and CE loss in SP to scaling considerations reveals a concrete recommendation: Practitioners could leverage stable parameterizations like $\mu$P for image datasets or SP-full-align for language datasets to effectively utilize loss functions beyond cross-entropy at large scales. In Figure 7, we consistently observe that CE loss greatly outperforms MSE under SP, whereas MSE consistently outperforms CE loss under $\mu$P, with differences becoming particularly pronounced at large widths. These results highlight the potential for substantial performance gains from further investigating the interplay between the loss function and parameterization. They also suggest that it is worth exploring the use of further loss functions in conjunction with $\mu$P.

**Identifying the correct scaling mechanisms enables principled improvements of scaling practice.**

Identifying the correct causal mechanism for training stability under large learning rates enables principled and informed exploration of the extended search space of practically relevant potentially best-performing parameterizations for efficiently finding improvements in model scaling practice: The controlled divergence regime had previously been neglected, even though common scaling practice as well as the best-performing parameterization SP-full-align from Everett et al. (2024) exactly operate in this regime. We now detail two concrete implications together with exciting avenues for future work.

**Identifying the correct mechanism for training instability at large scales.** Training instability due to logit divergence at large scales is a widely-established empirical phenomenon under SP which has motivated popular interventions like the z-loss (Chowdhery et al., 2023, Wortsman et al., 2024). However, only understanding the underlying causal mechanism and providing the exact width-dependent scaling exponents enables to design principled interventions. Both our theoretical and empirical results suggest that practically relevant large-scale neural networks naturally operate at the boundary of the controlled divergence regime, allowing persistent hidden-layer feature learning despite logit divergence. Consequently scaling up networks in SP inherently causes logit divergence, leading to numerical instability. As a principled intervention, stable parameterizations like $\mu$P should not suffer from such systematic divergence and potentially eliminate the need for ad-hoc interventions like the z-loss.

**Logit divergence induces overconfidence.** A similar argument applies to uncertainty calibration. Our theory also explains why we should expect predictions to be increasingly overconfident with increasing model scale and suggests that this may be partially mitigated by considering stable parameterizations like $\mu$P. However there may be inherent trade-offs between miss-calibration and faster memorization due to logit divergence. Their effect on performance and designing the ideal intervention deserves a more thorough investigation beyond the scope of the current paper.

**Understanding the transfer of SP-full-align motivates a novel last-layer initialization.** Small last-layer initialization recovers width-independent and hence predictable scaling dynamics under sufficient precision in the regime $n \gg d_{\text{out}}$, whereas standard last-layer initialization induces logit blowup at sufficient width, which is not necessarily harmful for generalization but reduces

predictability as scaling is not fully width-independent. Standard initialization with $\mu$P learning rates (SP-full-align) can induce 'practical transfer' and empirically update all weights effectively without logit blowup at moderate scales $n \ll d_{\text{out}}$ in the regime where the width is much smaller than the output dimension, as is relevant for NLP settings, but likely exhibits unexpected changed behaviour at sufficient scale, when logits start to diverge due to the last-layer term $W_0^{L+1} \Delta x_t^L$. This can be read off from differing dominating terms in $\|W_0^{L+1}\|_{RMS \rightarrow RMS}$, assuming width-independent alignment $\alpha_{W_0^{L+1} \Delta x_t^L} = \Theta(1)$ (as verified in Figure E.18). For uniform non-asymptotic transfer for both $n \gg d_{\text{out}}$ and $n \ll d_{\text{out}}$, the same argument motivates a last-layer initialization $\sigma_{L+1} = (\frac{\texttt{fan\_in}}{\sqrt{\texttt{fan\_out}}} + \sqrt{\texttt{fan\_in}})^{-1}$, that transitions from SP initialization in the regime $n \ll d_{\text{out}}$ to $\mu$P initialization in the regime $n \gg d_{\text{out}}$, similar to the one proposed in Yang et al. (2023a). Verifying improved transfer under this initialization is left to future work, since it requires scaling to widths beyond our computational constraints.

**Faithful evaluation of the accuracy of infinite-width theory.** While in principle the refined coordinate check (RCC) was previously known (Yang and Hu, 2021, Appendix H), we find that the predicted update exponents are surprisingly accurate in this decomposition already at moderate width and over the course of training (Figures 2 and 4), even in parameterizations with width-dependent dynamics like SP. Consequently, optimal and maximal stable learning rate exponents in SP also follow the predicted width-dependent exponent in realistic settings (see e.g. Figures 5 and F.10, Figures F.4 and F.5). To lower the hurdle for practitioners to incorporate the (RCC) in their workflow as a diagnostic tool, we have made our fine-grained and easily adaptable implementation of the RCC using LitGPT publicly available at https://github.com/tml-tuebingen/torch-module-monitor. Understanding and correcting miss-scaled update signals through each weight matrix has impactful consequences on performance and trainability at large scale.

## C   Theoretical considerations

### C.1   Distilled TP scaling arguments

Here we aim to provide a more detailed, comprehensive introduction to the essential width-scaling arguments inspired by Tensor Program (TP) theory.

**Effective Updates.** When training neural networks, we have control over the initial scaling $W_0$ and update scaling $\Delta W_t$ of trainable weights $W_t = W_0 + \Delta W_t$, but we are ultimately interested in their effect on the activations in the following layers. In standard architectures (including convolutional networks and Transformers), weights typically act linearly on the incoming activations. For such weights $W_t$ and incoming activations $x_t$, we can decompose the next layer's (pre-)activation updates $\Delta h_t$ into effective updates of $W_t$ and activation updates $\Delta x_t$ propagating forward from previous layers. Evaluating the contributions of both terms separately yields a *refined coordinate check*,

$$\Delta h_t = (\Delta W_t)x_t + W_0(\Delta x_t). \tag{RCC}$$

Note that updates of previous layers can propagate forward through the term $W_0 \Delta x_t$ even when the current layer's effect on the output vanishes $\Delta W_t x_t \rightarrow 0$ as width $n \rightarrow \infty$. Hence, we say that the weight $W_t$ is *effectively updated* only if $\Delta W_t x_t$ contributes non-vanishingly. Plotting the width-dependent scaling of $\|(\Delta W_t)x_t\|_{RMS}$ and $\|W_0(\Delta x_t)\|_{RMS}$ as a *refined coordinate check*, has been very useful for us to gain insights into the network internal signal propagation. The usefulness of (RCC) for effective update scalings is illustrated in Figure C.1. While the activations and activation updates in a Layernorm layer evolve width-independently when training GPT with Adam in SP and global learning rate scaled as $\eta_n = \eta \cdot n^{-1}$, the refined coordinate check reveals that the effective updates in the current (input-like) layer and the activation update scaling instead stems from effective updates propagating forward from previous (hidden-like) layers.

By choosing layerwise initialization variances and learning rates according to the Maximal Update Parameterization ($\mu$P), both terms in (RCC) become width-independent in all layers in each update step. Consequently, width-scaling becomes predictable, stable and feature learning is preserved even at large width. Starting from SP, $\mu$P can be realized with smaller last-layer initialization $\|W_0^{L+1}\|_{RMS} = O(n^{-1})$, larger input layer learning rate $\eta_{W^1} = \eta \cdot n$ and smaller last-layer learning rate $\eta_{W^{L+1}} = \eta \cdot n^{-1}$ for SGD.

**Predicting scaling exponents.** While the TP framework formally requires writing out all forward and backward pass computations performed during training and provides the exact infinite-width limit objects of output logits and activation coordinate distributions, we simplify its implications on width-scaling exponents for practical purposes as follows. A linear transformation either maps fixed to width-scaling dimension (*input-like*), width-scaling to width-scaling (*hidden-like*) or width-scaling to fixed dimension (*output-like*). Here, all bias vectors and normalization layer weights can be understood as input-like weights to the one-dimensional input 1. Any sum of length $n \to \infty$ that occurs in individual terms in (RCC) either accumulates a factor $n^{1/2}$ under sufficient independence of $0$-mean summands (CLT-like behaviour) or a factor $n$ when the summands are correlated or have non-zero mean (LLN-like behaviour). Crucially, not any sum may be evaluated with this heuristic but only weight and activation (update) pairs as in (RCC) (see Yang and Hu (2021, Appendix H)). If, for example, we considered the confounded term $(W_0 + \Delta W_t)x_0$, the initial part $W_0 x_0$ clearly scales CLT-like but $\Delta W_t x_0$ scales LLN-like; evaluating the scaling of their sum might result in wrong scaling predictions.

At sufficient width, all width-scaling inner products $(\Delta W_t, x_t)$ from (RCC), however, are expected to behave LLN-like, that is $\|\Delta W_t x_t\|_{RMS} = \Theta(n \cdot \|\Delta W_t\|_{RMS} \cdot \|x_t\|_{RMS})$.

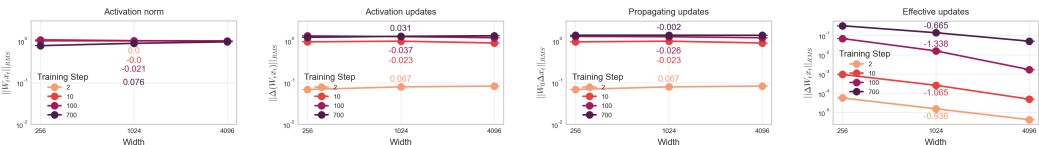

Figure C.1: **(Tracking effective updates requires refined coordinate check)** Activation norm $\|W_t x_t\|_{RMS}$, activation update norm $\|\Delta(W_t x_t)\|_{RMS}$, propagating update norm $\|W_0 \Delta x_t\|_{RMS}$ and effective update norm $\|\Delta W_t x_t\|_{RMS}$ for the last normalization layer in GPT trained with Adam and learning rate scaling $\eta_n = 0.01 \cdot n^{-1}$ for width-independent hidden layer feature learning. While activations and activation updates appear width-independent due to propagating updates, our refined coordinate check (RCC) reveals that Layernorm weight updates have vanishing effect in SP. Over time, effective updates accumulate effective rank, but do not lose alignment with width (Figure 2).

**Concrete examples.** Complementing the more generic introduction to TP scaling arguments above, we now provide more concrete examples for illustrating how weight updates affect activations in subsequent forward passes. Consider the Tensor Program for training MLPs with SGD from Yang and Hu (2021, Appendix H). We restate the relevant update scalings when using large learning rates in SP that induce output blowup. Since divergence is not allowed in the TP framework, it does not formally cover the unstable case, but we can still heuristically write down the scaling predictions, assuming that correlations still induce LLN-like exponents and independence still induces CLT-like exponents, as we have measured to hold empirically. The crucial insight is that training with cross-entropy loss effectively means that we are considering $f_t(\xi) = \sigma(W_t^{L+1} x_t^L(\xi))$ as the output function and the loss derivative also becomes $\chi_t := \frac{\partial \mathcal{L}_t}{\partial f} = \sigma(W_t^{L+1} x_t^L) - y_t$. Hence, from a stability point of view, we can allow $\tilde{f}_t := W^{L+1} x^L \to \infty$, which results in a saturated softmax. Under one-hot labels $y \in \{0,1\}^C$ with $\sum_c y_c = 1$, this means fast memorization of training points $(x_i, y_i)$. For width-independent hidden-layer feature learning, we may still require activations to have width-independent coordinate-scaling, but let the output function be arbitrary, since the softmax renormalizes.

**Definition C.1 (Activation stability).** A parameterization is *activation stable* iff $\|x_t^l\|_{RMS} = \Theta(1)$ for all times $t \geq 0$ and all layers $l \in [L]$. ◀

We now show heuristically that MLPs trained with SGD in SP are activation stable and feature learning under global learning rate scaling $\eta = \Theta(n^{-1/2})$.

**Backward pass.** Here we denote the entry-wise scaling in width-scaling vectors as $v = \Theta(n^c)$, meaning $\|v\|_{RMS} = \Theta(n^c)$. Assuming $\|\phi'(h_t^l)\|_{RMS} = \Theta(1)$ as for ReLU (otherwise we would get vanishing gradients), the entries of the following width-scaling vectors scale as

$$\frac{\partial f}{\partial x_t^L} = \qquad W_t^{L+1} = W_0^{L+1} - \Delta W_t^{L+1} = O(n^{-1/2}), \qquad \frac{\partial f}{\partial h_t^l} = \frac{\partial f}{\partial x_t^l} \odot \phi'(h_t^l) = \Theta(\frac{\partial f}{\partial x_t^l}),$$

$$\frac{\partial f}{\partial x_t^{l-1}} = (W_0^l)^\top \frac{\partial f}{\partial h_t^l} - \eta\theta_{W^l}\sum_{s=0}^{t-1}\chi_s \frac{(\frac{\partial f}{\partial h_s^l})^\top \frac{\partial f}{\partial h_s^l}}{n}x_s^{l-1} = \Theta(\max(\frac{\partial f}{\partial h_t^l}, \eta(\frac{\partial f}{\partial h_s^l})^2 x_s^{l-1}) = \Theta(n^{-1/2}).$$

Note that any larger learning rate scaling would induce exploding gradients. For example, $\eta = \Theta(1)$ induces $\delta W_1^{L+1} = \Theta(1)$, so $\frac{\partial f}{\partial x_1^L} = \Theta(1)$ and $\frac{\partial f}{\partial x_1^{L-k}} = \Theta(n\frac{\partial f}{\partial x_1^{L-k+1}}) = \Theta(n^{2k-1})$ for $k \geq 1$. This results in exploding activations in the next forward pass, and even larger gradients in the following backward pass.

We therefore continue with $\eta = \Theta(n^{-1/2})$, and get the activation updates

$$\delta h_t^1 = -\eta\chi_{t-1}\frac{\partial f}{\partial h_{t-1}^1}(\xi_{t-1})^\top \xi = \Theta(n^{-1/2}\cdot 1\cdot n^{-1/2}\cdot 1) = \Theta(n^{-1}),$$

$$\delta h_t^l = W_{t-1}^l \delta x_t^{l-1} + \delta W_t^l x_t^{l-1}$$

$$= \theta_x\left(W_0^l \delta x_t^{l-1} - \eta\theta_{W^l}\sum_s \chi_{s-1}\frac{\partial f}{\partial h_{s-1}^l}\underbrace{(x_{t-1}^{l-1})^\top \delta x_t^{l-1}}_{O(n)}\right)$$
$$\underbrace{}_{\Theta(n)^{-1/2}}$$

$$-\eta\chi_{t-1}\theta_W\underbrace{\frac{\partial f}{\partial h_{t-1}^l}}_{\Theta(n^{-1/2})}\underbrace{(x_{t-1}^{l-1})^\top x_t^{l-1}}_{\Theta(n)} = \Theta(1),$$

The output updates are

$$\delta\tilde{f}_t(\xi) = \delta W_t^{L+1}x_t^L(\xi) + (W_0^{L+1}+\Delta W_{t-1}^{L+1})\delta x_t^L(\xi)$$

$$= -\eta\chi_{t-1}\underbrace{x_{t-1}^L x_t^L(\xi)}_{\Theta(n)} + \underbrace{W_0^{L+1}\delta x_t^L(\xi)}_{\Theta(1)} + \underbrace{\Delta W_{t-1}^{L+1}\delta x_t^L(\xi)}_{\Theta(n^{1/2})} = \Theta(n^{1/2}),$$

$$\delta f_t(\xi) = \sigma(\tilde{f}_{t-1}+\delta\tilde{f}_t) - \sigma(\tilde{f}_{t-1}) = \Theta(1).$$

**2 layer networks.** Observe that in 2 layer nets, there are no hidden layers, so that a larger learning rate can be chosen. Let $\eta = \Theta(n^c)$. Then in the first step, $\delta h_1^1 = \Theta(\eta\frac{\partial f}{\partial h_0^1}) = \Theta(n^c n^{-1/2})$. But note that the gradient scaling may grow after the first step, $\frac{\partial f}{\partial x_1^L} = W_1^{L+1} = \Theta(n^c)$, so that $\delta h_2^1 = \Theta(n^c n^c)$. Hence activation stability requires $\eta = O(1)$, which results in feature learning after 2 steps $\delta x_2^1 = \Theta(1)$. Then $\tilde{f}_t = \Theta(\eta(x_{t-1}^L)^\top x_t^L(\xi)) = \Theta(n)$.

**Random feature models.** In random feature models, we only train the last layer and keep all other weights fixed $W_t^l = W_0^l$ for all $l \leq L$. There, by definition, we do not get feature learning and the backward pass does not matter. The only gradient that matters is the last-layer gradient which has fixed scaling $\Theta(\chi_{t-1}x_{t-1}^L) = \Theta(1)$ at all times $t \geq 0$. The function update becomes $\delta W_t^{L+1}x^L(\xi) = -\eta\chi_{t-1}(x^L(\xi_{t-1}))^\top x^L(\chi) = \Theta(n^c n)$, where the inner product between activations converges to the NNGP kernel in the infinite-width limit. Hence large learning rates $\eta = \omega(n^{-1})$ result in immediate extreme memorization of the training points $f_t(\xi_{t-1}) \to \text{one-hot}(y_{t-1})$ as $n \to \infty$, and $\eta_n = \Theta(n^{-1})$ results in fully width-independent dynamics.

**Adam.** Adam with small enough $\varepsilon$ normalizes the gradients in each layer before updating the weights. Since the gradients $\nabla_{W^l}\mathcal{L} = \chi\frac{\partial f}{\partial h^l}(x^{l-1})^\top$ are generally correlated with the incoming activations $x^{l-1}$, their inner product accumulates $\Theta(\texttt{fan\_in})$. Non-vanishing correlation persists when only recovering the signs of the gradient. Hence for a width-independent effect on the output of the current layer, the learning rate should always be chosen as $\eta(W) = \frac{\eta}{\texttt{fan\_in}(W)}$. Since both hidden and output layers have $\texttt{fan\_in} = n$, activation stability requires a global learning rate $\eta = O(n^{-1})$, which results in effective hidden and output layer learning, but vanishing input layer updates. Networks recover input layer feature learning under $\eta = \Theta(1)$, where $\tilde{f}_t = \Theta(n)$. In random feature models, $\eta$ just determines the extremeness of memorization of the training labels, where $\eta = \Theta(n^{-1})$ induces width-independence and $\eta = \omega(n^{-1})$ increasing memorization.

## C.2 Measuring Alignment

Everett et al. (2024, Fig. 2) provides RMS-alignment exponents between weights $W_t$ and incoming activations $x_t$. But only measuring the alignment between $\Delta W_t$ and $x_t$ as well as $W_0$ and $\Delta x_t$ from (RCC) separately allows to evaluate the width-scaling predictions from Yang and Hu (2021).

For example hidden layers in $\mu$P scale as $(W_0^l)_{ij} = \Theta(n^{-1/2})$ at initialization, as 0-mean independence induces CLT-like scaling

$$W_0^l x_0^{l-1} = \Theta(n^{1/2} \cdot \|W_0^l\|_{RMS} \cdot \|x_0^{l-1}\|_{RMS}).$$

But updates are correlated with incoming activations, so that

$$\Delta W_t x_t = \Theta(n \cdot \|\Delta W_t\|_{RMS} \cdot \|x_t\|_{RMS}),$$

which necessitates $\|\Delta W_t\|_{RMS} = \Theta(n^{-1})$. This implies that the entry size of $W_t = W_0 + \Delta W_t$ is dominated by the initialization and confounds $\|W_t\|_{RMS}$ for accurately measuring the alignment exponent of the layer's updates $\Delta W_t$. For correct width-scaling of the layer's learning rate, the influence of $W_0$ is irrelevant so that the joint alignment between $W_t$ and $x_t$ does not reveal the alignment exponent that is relevant for correct learning rate scaling.

Additionally, replacing the RMS-norm $\|A\|_{RMS}$ by the operator norm $\|A\|_{RMS \to RMS}$ provides a more natural measure of alignment (Bernstein and Newhouse, 2024), since the RMS-norm is confounded by accumulated rank whereas under maximal alignment for the operator norm it holds that $\|\Delta W_t x_t\|_{RMS} = \|\Delta W_t\|_{RMS \to RMS}\|x_t\|_{RMS}$, and the left-hand side is smaller under less alignment. Under perfect alignment we expect the ratio $\frac{\|\Delta W_t x_t\|_{RMS}}{\|\Delta W_t\|_{RMS \to RMS}\|x_t\|_{RMS}}$ to remain width-independent. We are not interested in constant prefactors, but only width-dependent scaling.

## C.3 Formal statements and proofs of Propositions 1 and 2

Before providing the full formal statements of Proposition 2 and Proposition 4, we formally introduce all definitions and assumptions.

### C.3.1 Definitions

In this section, we collect all definitions that do not appear in the main text. We adopt all definitions from Yang and Hu (2021), up to minor modifications. If not stated otherwise, limits are taken with respect to width $n \to \infty$.

**Definition C.2 (Big-O Notation).** Given a sequence of scalar random variables $c = \{c_n \in \mathbb{R}\}_{n=1}^{\infty}$, we write $c = \Theta(n^{-a})$ if there exist constants $A, B \geq 0$ such that for almost every instantiation of $c = \{c_n \in \mathbb{R}\}_{n=1}^{\infty}$, for $n$ large enough, $An^{-a} \leq |c_n| \leq Bn^{-a}$. Given a sequence of random vectors $x = \{x_n \in \mathbb{R}^n\}_{n=1}^{\infty}$, we say $x$ has coordinates of size $\Theta(n^{-a})$ and write $x = \Theta(n^{-a})$ to mean the scalar random variable sequence $\left\{\sqrt{\|x_n\|^2 / n}\right\}_n$ is $\Theta(n^{-a})$. For the definition of $c = O(n^{-a})$ and $c = \Omega(n^{-a})$, adapt the above definition of $c = \Theta(n^{-a})$ by replacing $An^{-a} \leq |c_n| \leq Bn^{-a}$ with $|c_n| \leq Bn^{-a}$ and $An^{-a} \leq |c_n|$, respectively. We write $x_n = o(n^{-a})$ if $n^a \cdot \sqrt{\|x_n\|^2 / n} \to 0$ almost surely. ◀

**Definition C.3 (SGD update rule).** Given a $(L+1)$-layer MLP with layerwise initialization variances $\{\sigma_l\}_{l \in [L+1]}$ and (potentially) layerwise learning rates $\{\eta_{W^l}\}_{l \in [L+1]}$, we define the *SGD update rule* as follows:

(a) Initialize weights iid as $(W_0^l)_{ij} \sim \mathcal{N}(0, \sigma_l^2)$.
(b) Update the weights via

$$W_{t+1}^l = W_t^l - \eta_{W^l} \cdot \nabla_{W^l} \mathcal{L}(f_t(\xi_t), y_t).$$

◀

**Definition C.4 (Parameterization).** We define a *width-scaling parameterization* as a collection of exponents $\{b_l\}_{l \in [L+1]} \cup \{c_l\}_{l \in [L+1]}$ that determine layerwise initialization variances $\sigma_l^2 = C_l \cdot n^{-b_l}$ and layerwise learning rates $\eta_l = \eta \cdot n^{-c_l}$, with width-independent constants $C_l, \eta > 0$ for all $l \in [L+1]$. ◀

| | | | Weight-multiplier version | | | Weight-multiplier-free version | | |
|---|---|---|---|---|---|---|---|---|
| | | | Input-like | Hidden-like | Output-like | Input-like | Hidden-like | Output-like |
| SP | $\alpha_l \cdot W^l$, | $\alpha_l \propto$ | | | | 1 | 1 | 1 |
| | $\mathcal{N}(0,\sigma_l^2)$, | $\sigma_l \propto$ | | - | | 1 | $n^{-1/2}$ | $n^{-1/2}$ |
| | $\eta_l \cdot \nabla_{W^l}\mathcal{L}$, | $\eta_l \propto$ | | | | $n^{-c}$ | $n^{-c}$ | $n^{-c}$ |
| NTP | $\alpha_l \cdot W^l$, | $\alpha_l \propto$ | 1 | $n^{-1/2}$ | $n^{-1/2}$ | 1 | 1 | 1 |
| | $\mathcal{N}(0,\sigma_l^2)$, | $\sigma_l \propto$ | 1 | 1 | 1 | 1 | $n^{-1/2}$ | $n^{-1/2}$ |
| | $\eta_l \cdot \nabla_{W^l}\mathcal{L}$, | $\eta_l \propto$ | 1 | 1 | 1 | 1 | $n^{-1}$ | $n^{-1}$ |
| $\mu$P | $\alpha_l \cdot W^l$, | $\alpha_l \propto$ | $n^{1/2}$ | 1 | $n^{-1/2}$ | 1 | 1 | 1 |
| | $\mathcal{N}(0,\sigma_l^2)$, | $\sigma_l \propto$ | $n^{-1/2}$ | $n^{-1/2}$ | $n^{-1/2}$ | 1 | $n^{-1/2}$ | $n^{-1}$ |
| | $\eta_l \cdot \nabla_{W^l}\mathcal{L}$, | $\eta_l \propto$ | 1 | 1 | 1 | $n$ | 1 | $n^{-1}$ |

Table C.1: (**Common** $abc$**-parameterizations**) Here, we collect standard parameterization (SP), neural tangent parameterization (NTP) and the maximal update parameterization ($\mu$P) for SGD in their multiplier version which purely adapts the architecture and allows width-independent global learning rates (*left*) and in their weight multiplier-free version (*right*). Parameterizations differ in their layerwise choice of width-dependent weight multipliers $\alpha_l$, initialization variances $\sigma_l$ and learning rates $\eta_l$. Weight multiplier-free representatives of an $abc$-equivalence class purely adapt the optimization algorithm highlighting the fact that parameterizations effectively only induce layerwise learning rates. Knowing that $\mu$P correctly scales the updates in all layers, observe that the input- and hidden-layer learning rates in NTP induce vanishing updates. The same holds in SP when choosing $c \geq 1$ as is necessary for avoiding logit blowup in the infinite-width limit.

In Table C.1, we summarize the three most common parameterizations SP, NTP and $\mu$P (equivalent to the mean-field parameterization).

**Definition C.5** (**Training routine**). A *training routine* is a combination of base learning rate $\eta \geq 0$, training sequence $\{(\xi_t, y_t)\}_{t \in \mathbb{N}}$ and a continuously differentiable loss function $\mathcal{L}(f(\xi), y)$ using the SGD update rule. ◀

**Definition C.6** (**Stability**). We say a parametrization of a $(L+1)$-layer MLP is *stable* if

1. For every nonzero input $\xi \in \mathbb{R}^{d_{in}} \backslash \{0\}$,

$$h_0^l, x_0^l = \Theta_\xi(1), \ \forall l \in [L], \quad \text{and} \quad \mathbb{E}f_0(\xi)^2 = O_\xi(1),$$

   where the expectation is taken over the random initialization.
2. For any training routine, any time $t \in \mathbb{N}$, $l \in [L]$, $\xi \in \mathbb{R}^{d_{in}}$, we have

$$h_t^l(\xi) - h_0^l(\xi), x_t^l(\xi) - x_0^l(\xi) = O_*(1), \quad \text{and} \quad f_t(\xi) = O_*(1),$$

   where the hidden constant in $O_*$ can depend on the training routine, $t$, $\xi$, $l$ and the initial function $f_0$.

◀

**Definition C.7** (**Nontriviality**). We say a parametrization is *trivial* if for every training routine, $f_t(\xi) - f_0(\xi) \to 0$ almost surely for $n \to \infty$, for every time $t > 0$ and input $\xi \in \mathbb{R}^{d_{in}}$. Otherwise the parametrization is *nontrivial*. ◀

**Definition C.8** (**Feature learning**). We say a parametrization *admits feature learning in the l-th layer* if there exists a training routine, a time $t > 0$ and input $\xi$ such that $x_t^l(\xi) - x_0^l(\xi) = \Omega_*(1)$, where the constant may depend on the training routine, the time $t$, the input $\xi$ and the initial function $f_0$ but not on the width $n$. ◀

**Definition C.9** ($\sigma$-**gelu**). Define $\sigma$-gelu to be the function $x \mapsto \frac{x}{2}\left(1 + \text{erf}\left(\sigma^{-1}x\right)\right) + \sigma \frac{e^{-\sigma^{-2}x^2}}{2\sqrt{\pi}}$. ◀

In order to apply the Tensor Program Master Theorem, all Nonlin and Moment operations in the NE$\otimes$OR$\top$ program (Yang and Littwin, 2023), which do not only contain parameters as inputs, are required to be pseudo-Lipschitz in all of their arguments. For training with SGD, this is fulfilled as soon as $\phi'$ is pseudo-Lipschitz. $\sigma$-gelu fulfills this assumption.

**Definition C.10** (**Pseudo-Lipschitz**). A function $f : \mathbb{R}^k \to \mathbb{R}$ is called *pseudo-Lipschitz of degree $d$* if there exists a $C > 0$ such that $|f(x) - f(y)| \leq C\|x - y\|(1 + \sum_{i=1}^{k} |x_i|^d + |y_i|^d)$. We say $f$ is *pseudo-Lipschitz* if it is so for any degree $d$. ◀

Finally, we differentiate relevant notions of maximal stable learning rates.

**Definition C.11** (**Maximal stable learning rates**). Fix $t \in \mathbb{N}$ and consider any layerwise learning rate scaling rule $\eta_l = \eta_n \cdot n^{-c_l}$ for $l \in [L+1]$ with prescribed width-scaling exponents $\{c_l\}_{l \in [L+1]}$ and tunable global learning rate $\eta_n = \Theta(n^{-\alpha})$ as we scale width $n \to \infty$.

(a) **Theoretical max-stable learning rate of a network.** We define the *theoretical maximal stable learning rate exponent $\bar{\alpha}$ of the network* as the boundary to the catastrophic regime, that is there exists a training routine and $\xi \in \mathbb{R}^{d_{in}}$ such that for all $\alpha < \bar{\alpha}$ training with $\eta_n = \eta \cdot n^{-\alpha}$ implies $\|f_t\|_{RMS} \to \infty$ and $\|x_t^l\|_{RMS} \to \infty$ a.s. after $t$ steps for all $l \in [L+1]$.

(b) **Theoretical max-stable learning rate of a layer.** We define the *theoretical maximal stable learning rate exponent $\bar{\alpha}_l$ w.r.t. layer $l$* as the global learning rate scaling above which the layer's effective updates diverge, that is, given stable inputs $\|x\|_{RMS} = \Theta_\xi(1)$ with maximal alignment $\alpha_{\Delta W_t^l, x} = \Theta_\xi(1)$, there exists a training routine such that for all $\alpha < \bar{\alpha}_l$ training with a global learning rate $\eta_n = \eta \cdot n^{-\alpha}$ implies $\|\Delta W_t^l x\|_{RMS} \to \infty$ a.s.

(c) **Empirical max-stable learning rate.** We softly define the *empirical maximal stable learning rate exponent $\bar{\alpha}$* as the global learning rate exponent above which training diverges and the network's predictions degrade to random guessing.

◀

### C.3.2 Full formal statements of Propositions 1 and 2

**Assumptions.** For all of the results in this section, we assume that the used activation function is $\sigma$-gelu for $\sigma > 0$ sufficiently small. For small enough $\sigma > 0$, $\sigma$-gelu (Definition C.9) approximates ReLU arbitrarily well. We assume constant training time $t \geq 1$ as width $n \to \infty$. We assume batch size 1 for clarity, but our results can be extended without further complications to arbitrary fixed batch size.

**Proposition C.12.** *(Asymptotic regimes in SP) For fixed $L \geq 2$, $t \geq 1$, $\eta > 0$, $\alpha \in \mathbb{R}$, consider training a $(L+1)$-layer MLP of width $n$ in SP with SGD and global learning rate $\eta_n = \eta \cdot n^{-\alpha}$ for $t$ steps. Then the logits $f_t$, training loss $\mathcal{L}(f_t(\xi_t), y_t)$, loss-logit derivatives $\chi_t := \partial_f \mathcal{L}(f_t(\xi_t), y_t)$, loss-weight gradients $\nabla_t^l := \nabla_{W^l} \mathcal{L}(f_t(\xi_t), y_t)$ and activations $x_t^l$, $l \in [L]$, after training scale as follows in the infinite-width limit $n \to \infty$. The hidden constants in $O_*, \Omega_*$ and $\omega_*$ below can depend on the training routine, $t$, $\xi$, $l$ and the initial function $f_0$.*

*Under cross-entropy (CE) loss, three qualitatively distinct regimes arise:*

(a) ***Stable regime*** *($\alpha \geq 1$): For any training routine, all $l \in [L]$ and any $\xi \in \mathbb{R}^{d_{in}}$, it holds that $\|f_t(\xi)\|_{RMS} = O_*(1)$, $|\mathcal{L}(f_t(\xi_t), y_t)| = O_*(1)$, $\|\chi_t\|_{RMS} = O_*(1)$, $\|\nabla_t^l\|_{RMS} = O_*(n^{-1/2})$ and $\|x_t^l(\xi)\|_{RMS} = O_*(1)$.*

(b) ***Controlled divergence*** *($\frac{1}{2} \leq \alpha < 1$): For any training routine, all $l \in [L]$ and any $\xi \in \mathbb{R}^{d_{in}}$, it holds that $\|n^{\alpha-1} \cdot f_t(\xi)\|_{RMS} = O_*(1)$, $\|x_t^l(\xi) - x_0^l(\xi)\|_{RMS} = O_*(1)$, $|\mathcal{L}(f_t(\xi_t), y_t)| = O_*(1)$, $\|\chi_t\|_{RMS} = O_*(1)$ and $\|\nabla_t^l\|_{RMS} = O_*(n^{-1/2})$. In addition, there exists a training routine and input $\xi$ such that $\|n^{\alpha-1} \cdot f_t(\xi)\|_{RMS} = \Omega_*(1)$.*

(c) ***Catastrophic instability*** *($\alpha < \frac{1}{2}$): For any $l \in [L]$, there exists a training routine and a $\xi \in \mathbb{R}^{d_{in}}$, such that $\|f_t(\xi)\|_{RMS} = \omega_*(1)$, $\|x_t^l(\xi)\|_{RMS} = \omega_*(1)$ and $\|\nabla_t^l\|_{RMS} = \omega_*(1)$.*

*Under mean-squared error (MSE) loss, a stable regime as in (a) above arises if $\alpha \geq 1$. If $\alpha < 1$, training is catastrophically unstable as in (c) above and, in addition, there exists a training routine such that $|\mathcal{L}(f_t(\xi_t), y_t)| = \omega_*(1)$ and $\|\chi_t\|_{RMS} = \omega_*(1)$.*

**Proposition C.13** (**Under CE loss, SP with large learning rates learns features at large width**). *Consider the setting of Proposition 2 of training a $(L+1)$-layer MLP with SGD in SP with global learning rate $\eta_n = \eta \cdot n^{-\alpha}$, $\alpha \in \mathbb{R}$, in the infinite-width limit $n \to \infty$. The hidden constants in $O_*, \Omega_*$ and $\omega_*$ below can depend on the training routine, $t$, $\xi$, $l$ and the initial function $f_0$.*

(a) *Under both MSE and CE loss in the stable regime ($\alpha \geq 1$), for any training routine, $l \in [L]$ and $\xi \in \mathbb{R}^{d_{in}}$ it holds that $\|\Delta x_t^l(\xi)\|_{RMS} = O_*(n^{-1/2})$.*

(b) *Under CE loss in the controlled divergence regime ($\frac{1}{2} \leq \alpha < 1$), for any training routine, $l \in [L]$ and $\xi \in \mathbb{R}^{d_{in}}$ it holds that $\|\Delta x_t^1(\xi)\|_{RMS} = O_*\left(n^{-1/2-\alpha}\right)$, and $\|\Delta x_t^l(\xi)\|_{RMS} = O_*\left(n^{1/2-\alpha}\right)$. For any $l \in [L]$, there exists a training routine and $\xi \in \mathbb{R}^{d_{in}}$ such that $\|\Delta x_t^1(\xi)\|_{RMS} = \Omega_*\left(n^{-1/2-\alpha}\right)$, and $\|\Delta x_t^l(\xi)\|_{RMS} = \Omega_*\left(n^{1/2-\alpha}\right)$.*

**Remark C.14** (**2-layer networks recover stable training dynamics and width-independent feature learning at** $\alpha = 0$). Similarly, it can be shown that 2-layer MLPs remain activation stable under width-independent learning rate scaling $\eta_n = \Theta(1)$. The controlled divergence regime is given by $0 \leq \alpha < 1/2$, with width-independent input layer feature learning at $\alpha = 0$. Experimental evidence in Appendix F.4 supports this maximal stable learning rate scaling prediction. ◄

**Remark C.15** (**Adam recovers stable training dynamics and width-independent hidden-layer feature learning at** $\alpha = 1$). For Adam and $L \geq 2$, an analogous NE⊗ORT-based proof (Yang and Littwin, 2023) would show that $\eta_n = \Theta(n^{-1})$ recovers feature learning in all hidden layers $l \in [2, L]$, stable activations and loss-logit gradients, while logits blow up only through $W_0^{L+1} \Delta x_t^L = \Theta(n^{1/2})$. To avoid logit blowup, $\eta_n = \Theta(n^{-3/2})$ would be necessary. In that case, only the term $W_0^{L+1} \Delta x_t^L$ would contribute non-vanishingly to the logit updates. Hence, for Adam under CE loss, the controlled divergence regime is given by $1 \leq \alpha < 3/2$, with hidden-layer feature learning at $\alpha = 1$. ◄

### C.3.3 Proof of Propositions 1 and 2

The proof in Yang and Hu (2021) for general stable $abc$-parameterizations directly covers the stable regimes of both losses, showing a kernel regime and vanishing feature learning for $\alpha \geq 1$.

For the controlled divergence regime under CE loss, however, note that the TP framework does not allow diverging computations. Here, we need to replace the logit updates by rescaled logit updates, before computing the softmax limit outside of the TP framework.

Formally, under standard initialization, $W_0^{L+1} \sim N(0, 1/n)$ is replaced in the TP by $\hat{W}_\varepsilon^{L+1}$ constructed via Nonlin, conditioning on $f_0(\xi)$ (see Yang and Hu (2021, Appendix H) for all details). For stable parameterizations, the function updates are defined in the TP as

$$\delta \hat{f}_t = \theta'_{L+1} \frac{\delta W_t^{L+1} x_t^L}{n} + \theta'_{Lf} \frac{\hat{W}_{t-1}^{L+1} \delta x_t^L}{n},$$

where $\theta'_{L+1} = n^{1-\alpha}$ and $\theta'_{Lf} = n^{1-r-b_{L+1}}$. In the controlled divergence regime $\alpha < 1$, we now define rescaled logit updates in the TP as

$$\delta \hat{f}_t = \hat{\theta}_{L+1} \frac{\delta W_t^{L+1} x_t^L}{n} + \hat{\theta}_{Lf} \frac{\hat{W}_{t-1}^{L+1} \delta x_t^L}{n},$$

by replacing $\theta'_{L+1}$ by $\hat{\theta}_{L+1} := \theta_\alpha \theta'_{L+1}$ and replacing $\theta'_{Lf}$ by $\hat{\theta}_{Lf} := \theta_\alpha \theta'_{Lf}$, where $\theta_\alpha := n^{\alpha-1}$. The adapted pre-factors ensure that $\delta \hat{f}$ remains $O_*(1)$ for a well-defined TP. The TP master theorem now implies almost sure convergence of the rescaled logit updates $\delta \hat{f}_t \to \mathring{\delta} \hat{f}_t \in \mathbb{R}^{d_{\text{out}}}$ a.s.

Now we compute the softmax limit outside of the TP framework, as we want to recover the softmax values of the original diverging logits. Thus, given the convergent sequence $\delta \hat{f}_t \to \mathring{\delta} \hat{f}_t \in \mathbb{R}^{d_{\text{out}}}$ a.s., due to the smoothness and saturation properties of the softmax it follows that there exists a $\mathring{\chi}_t \in \mathbb{R}^{d_{\text{out}}}$ such that $\sigma(\theta_\alpha^{-1} \cdot \delta \hat{f}_t) - y_t \to \mathring{\chi}_t$ a.s. Since $|\sigma(\theta_\alpha^{-1} \cdot \delta \hat{f}_t) - y_t| \leq 1 + |y_t|$ and $|\mathring{\chi}_t| \leq 1 + |y_t|$, this sequence can again be used as a TP scalar. Now the last-layer weights are TP vectors updated with $\delta W_t^{L+1} = -\eta_n \chi_t x_t^L$ which do not change the scaling of $\hat{W}_t^{L+1} = \hat{W}_{t-1}^{L+1} + \theta_{L+1/f} \cdot \delta W_1^{L+1}$ with $\theta_{L+1/f} \leq 1$ as long as $\alpha \geq 1/2$. Thus the backward pass scalings are not affected and the rest of the TP can remain unchanged.

For larger learning rates $\alpha < 1/2$ under CE loss, we provide heuristic scaling arguments. Observe that preactivations diverge after the first update step $\delta h_1^2 = -\eta_n \frac{\partial f_0}{\partial h^2}(x_0^1)^\top x_1^1 = \Theta(n^{1/2-\alpha})$. The updates of the next hidden layer's preactivations scale even larger, that is $\delta h_1^3 = -\eta_n \frac{\partial f_0}{\partial h^3}(x_0^2)^\top x_1^2 = \Theta(n^{2(1/2-\alpha)})$. In this way, the exponent growth continues throughout the forward pass. But even if there is only a single hidden layer, the scaling of the backpropagated gradient is increased after the second step, $\frac{\partial f_2}{\partial x^L} = W_0 - \eta_n \chi_0 x_0^L - \eta_n \chi_1 x_1^L = \Omega(\eta_n \chi_1 \delta x_1^L) = \Omega(n^{1/2-2\alpha}) = \omega(n^{-1/2})$. This, in turn, increases the preactivation update scaling $\delta h_3^2 = -\eta_n \frac{\partial f_2}{\partial h^2}(x_2^1)^\top x_3^1 = \Omega(n^{-\alpha} \frac{\partial f_2}{\partial x^L} n) = \Omega(n^{3(1/2-\alpha)})$, which in turn increases the gradient scaling in the next step, inducing a feedback loop of cascading exponents between diverging activations and gradients, inducing fast training divergence.

Under MSELoss, observe how, already for $\alpha < 1$, diverging logits $\delta f_1 = W_0^{L+1} \delta x_1^L - \eta_n \chi_0 (x_0^L)^\top x_0^L = \Theta(n^{1-\alpha})$ increase the gradient scaling through $\chi_1 = f_1 - y_1 = \Theta(n^{1-\alpha})$ which in

turn increases the activation as well as logit scaling in the next step, and induces a divergent feedback loop even worse than above.

## C.4 Scaling dynamics in 2-layer linear networks

Here, we rederive the training dynamics of the minimal model from Lewkowycz et al. (2020) that shows an initial catapult mechanism in NTP. They observe that the training dynamics of repeatedly updating a 2-layer linear network in NTP on the same training point is fully captured by update equations of the current function output $t_t$ and the current sharpness $\lambda_t$.

### C.4.1 Deriving the update equations for SP, NTP and $\mu$P

**NTP.** The original model by Lewkowycz et al. (2020) is given by

$$f = n^{-1/2}vux,$$

where $u \in \mathbb{R}^{n \times d}, v \in \mathbb{R}^n$ are initialized as $u_{ij}, v_i \sim N(0, 1)$ and trained with MSE loss $L(f, x, y) = \frac{1}{2}(f(x) - y)^2$, loss derivative $\chi_t = f(x) - y$ and a global learning rate $\eta$.

Repeated gradient descent updates using $(x, y)$, then results in the update equations,

$$f_{t+1} = f_t(1 + n^{-1}\eta^2\chi^2\|x\|^2) - \eta\chi_t\lambda_t,$$
$$\lambda_{t+1} = \lambda_t + n^{-1}\eta\chi_t\|x\|^2(\eta\chi_t\|x\|^2\lambda_t - 4f_t),$$

where the *update kernel* is defined as

$$\tilde{\Theta}(x, x') = n^{-1}(\|u\|^2 + \|v\|^2).$$

Note that the width-dependence in $f_t$ and $\lambda_t$ results in qualitatively different behaviour in the infinite-width limit. In particular, in the limit, the sharpness cannot evolve over the course of training, $\lambda_t = \lambda_0$.

**Maximal Update Parameterization.** We define a *2-layer linear network in $\mu$P with arbitrary weight multipliers* as

$$f = \bar{v}\bar{u}x,$$

with *reparameterization-invariant weights* $\bar{u}_{ij} \sim N(0, 1/d_{in})$ and $\bar{v}_i \sim N(0, 1/n^2)$, $\bar{u} = n^{-a_u}u, \bar{v} = n^{-a_v}v$, and the *original weights* $u, v$ are trained with MSE loss and layerwise learning rates $\eta_u = \eta n^{1+2a_u}$ and $\eta_v = \eta n^{-1+2a_v}$, which results in reparameterization-invariant layerwise learning rates $\bar{\eta}_u = \eta n$ and $\bar{\eta}_v = \eta n^{-1}$.

Formally, we now perform updates on $u$ and $v$, but we can work with $\bar{u}$ and $\bar{v}$ instead. For gradients it holds that $\frac{\partial f}{\partial u} = \frac{\partial f}{\partial \bar{u}}\frac{\partial \bar{u}}{\partial u} = \frac{\partial f}{\partial \bar{u}}n^{-a_u}$; this width scaling has to be accounted for when transitioning between representatives of the $\mu$P equivalence class. For updates $\bar{\eta}_u, \bar{\eta}_v$ should be used instead of $\eta_u, \eta_v$, as the layerwise learning rate rescaling was exactly chosen to cancel out the effect of the weight rescaling,

$$\bar{u}_{t+1} - \bar{u}_t = -n^{-a_u}\eta_u\frac{\partial f}{\partial \bar{u}}\frac{\partial \bar{u}}{\partial u} = -n^{-2a_u}\eta_u\frac{\partial f}{\partial \bar{u}} = -\bar{\eta}_u\frac{\partial f}{\partial \bar{u}}.$$

The derivatives for backpropagation are given by,

$$\chi_t := \frac{\partial L}{\partial f} = f(x_t) - y_t, \qquad \frac{\partial f}{\partial \bar{v}} = x^\top\bar{u}^\top, \qquad \frac{\partial f}{\partial \bar{u}} = \bar{v}^\top x^\top.$$

The updated weights are then given by

$$\bar{v}_{t+1} = \bar{v}_t - \eta n^{-1}\chi_t x^\top\bar{u}^\top, \qquad \bar{u}_{t+1} = \bar{u}_t - \eta n\chi_t\bar{v}^\top x^\top.$$

In the case $d_{in} = 1$, the updated function is then given by

$$f_{t+1} = \bar{v}_{t+1}\bar{u}_{t+1}x = f_t + \eta^2\chi_t^2 x^\top\bar{u}^\top\bar{v}^\top x^\top x - \bar{\eta}_u\chi_t\bar{v}\bar{v}^\top x^\top x - \bar{\eta}_v\chi_t x^\top\bar{u}^\top\bar{u}x$$
$$= f_t(1 + \eta^2\chi_t^2\|x\|^2) - \eta\chi_t(n\|\bar{v}\|^2 + n^{-1}\|\bar{u}\|^2)\|x\|^2$$

$$= f_t\big(1 + \eta^2\chi_t^2\|x\|^2\big) - \eta\chi_t\tilde{\Theta}(x,x),$$

where we call $\tilde{\Theta}$ the reparameterization-invariant *update kernel* defined as

$$\tilde{\Theta}(x,x') = \sum_l \frac{\eta_l}{\eta}\frac{\partial f(x)}{\partial W^l}\frac{\partial f(x')}{\partial W^l} = x^\top\big(n^{-1}\|\bar{u}\|^2 + n\|\bar{v}\|^2\big)x'.$$

The update kernel evolves via the reparameterization-invariant update equation

$$
\begin{aligned}
\lambda_{t+1} = \quad & \tilde{\Theta}_{t+1}(x,x) = \|x\|^2\big(n\|\bar{v}_{t+1}\|^2 + n^{-1}\|\bar{u}_{t+1}\|^2\big) \\
= \quad & \|x\|^2\Big(n\|\bar{v}_t\|^2 + n^{-1}\|\bar{u}_t\|^2 + n^{-1}\bar{\eta}_u^2\chi_t^2\bar{v}\bar{v}^\top x^\top x \\
& -2n\bar{\eta}_v\chi_t\bar{v}\bar{u}x + n\bar{\eta}_v^2\chi_t^2 x^\top\bar{u}^\top\bar{u}x - 2n^{-1}\bar{\eta}_u\chi_t\bar{v}\bar{u}x\Big) \\
= \quad & \lambda_t + \|x\|^2\big(\eta^2\chi_t^2\|x\|^2(n^{-1}\|\bar{u}\|^2 + n\|\bar{v}\|^2) - 4\eta\chi_t f_t\big) \\
= \quad & \lambda_t + \|x\|^2\Big(\eta^2\chi_t^2\lambda_t - 4\eta\chi_t f_t\Big) \\
= \quad & \lambda_t + \|x\|^2\eta\chi_t\Big(\eta\chi_t\lambda_t - 4f_t\Big).
\end{aligned}
$$

Now note that, even under $f_0 = 0$, we get non-trivial, width-independent dynamics. Due to the LLN, at initialization, we have $n^{-1}\|\bar{u}_0\|^2 \approx 1$ and $n\|\bar{v}_0\|^2 \approx 1$ (=$n^{-1}$ times sum over $n$ iid $\chi^2$ variables), hence $\lambda_0 \approx 2$.

To conclude, the training dynamics for repeatedly updating with the same training point $(x,y)$ are fully described by the update equations,

$$f_{t+1} = \qquad f_t\big(1 + \eta^2\chi_t^2\|x\|^2\big) - \eta\chi_t\lambda_t, \qquad\qquad (C.1)$$

$$\lambda_{t+1} = \qquad \lambda_t + \|x\|^2\eta\chi_t\Big(\eta\chi_t\lambda_t - 4f_t\Big). \qquad\qquad (C.2)$$

This can be rewritten in terms of the error (or function-loss derivative under MSE loss) $\chi_t = f_t - y$, akin to Kalra et al. (2025), as

$$\chi_{t+1} = \qquad \chi_t\big(1 - \eta\lambda_t + \eta^2\|x\|^2\chi_t(\chi_t + y)\big), \qquad\qquad (C.3)$$

$$\lambda_{t+1} = \qquad \lambda_t + \|x\|^2\eta\chi_t\Big(\eta\chi_t\lambda_t - 4(\chi_t + y)\Big). \qquad\qquad (C.4)$$

First observe that all terms in the update equations become width-independent in $\mu$P. Only the initial conditions are width-dependent with vanishing variance, $f_0 = \Theta(n^{-1/2})$. As opposed to NTP, the sharpness update term $\eta^2\chi_t^2\|x\|^2$ is not vanishing anymore. While Lewkowycz et al. (2020) simply use labels $y = 0$, non-trivial dynamics in $\mu$P require $y \neq f_0 \to 0$.

Importantly, $\eta\chi_t$ always appear jointly, so that interpolation effectively reduces the learning rate.

**Remark C.16** (**Characterizing sharpness increase: Critical threshold depends on the labels.**). When both sharpness and the loss increase, then training diverges as the learning rate lies even further from its edge of stability. In $\mu$P, since $f_0 \to 0$, $\lambda_t$ will grow in the first step. For subsequent steps, the sharpness update equation (C.4) implies that sharpness increases ($\lambda_{t+1} \geq \lambda_t$) if and only if $\lambda_t \geq \frac{4}{\eta}(1 + \frac{y}{\chi}) = \frac{4}{\eta}\frac{f_t}{\chi_t}$. Kalra et al. (2025) provides a more extensive analysis of the dynamics and fixed points of this model in $\mu$P. ◀

**Remark C.17** (**Weight multipliers**). A natural choice of weight multipliers for $\mu$P can be considered to be $a_l = 1/2\cdot\mathbb{I}(l = L+1) - 1/2\cdot\mathbb{I}(l = 1)$, as this choice allows using a width-independent global learning rate $\eta_n = \eta\cdot n^0$, and the update kernel does not require width-dependent scaling factors, $\tilde{\Theta}(x,x') = x^\top\big(\|u\|^2 + \|v\|^2\big)x'$. In other words, under these weight multipliers, width-independence in parameter space translates into width-independence in function space. ◀

**Standard Parameterization.** We define training a 2-layer linear network in SP with global learning rate scaling $n^{-c}$ as

$$f = \bar{v}\bar{u}x,$$

with initialization $\bar{u} \sim N(0, 1/d_{in}), \bar{v} \sim N(0, 1/n)$ and global learning rate $\bar{\eta}_u = \bar{\eta}_v = \eta n^{-c}$. Parameter multipliers affect all scalings in the same way as for $\mu$P. Only the learning rate has a different prefactor, and the last layer has larger initialization. The adapted update equations become

$$f_{t+1} = f_t(1 + n^{-2c}\eta^2\chi^2\|x\|^2) - n^{1-c}\eta\chi_t\lambda_t,$$

$$\lambda_{t+1} = \lambda_t + \|x\|^2 n^{-c}\eta\chi_t(n^{-c}\eta\chi_t\lambda_t - 4n^{-1}f_t),$$

where we define, as for NTP,

$$\tilde{\Theta}(x, x') = n^{-1}(\|\bar{u}\|^2 + \|\bar{v}\|^2),$$

where $n^{-1}\|\bar{v}\|^2 \approx n^{-1}$ at initialization ($n^{-2}$ times sum over $n$ iid $\chi^2$-variables with positive mean).

Choosing $c < 1$ results in output blowup of the term $n^{1-c}\eta\chi_t\lambda_t$. While this can in principle be counteracted by shrinking $\lambda_t$ at finite width, a well-defined stable and non-trivial infinite-width limit is only attained at $c = 1$, where $f_{t+1} = f_t - \eta\chi_t\lambda_t$ and $\lambda_{t+1} = \lambda_t$. We now show that also at finite width, stable training with a constant learning rate in SP requires $\eta = O(n^{-1})$.

### C.4.2 Finding the maximal stable learning rate scaling by characterizing the conditions for loss and sharpness decrease

The following proposition characterizes the choices of $\eta$ that result in a decrease in loss at any present state.

Writing $n_{sp} = \begin{cases} n, & \text{in SP,} \\ 1, & \text{else,} \end{cases}$ $n_{ntp} = \begin{cases} n, & \text{in NTP,} \\ 1, & \text{else,} \end{cases}$, we can write the update equations of parameterizations jointly as

$$\chi_{t+1} = \chi_t(1 - n_{sp}\eta\lambda_t + n_{ntp}^{-1}\eta^2\chi\|x\|^2(\chi_t + y)),$$

$$\lambda_{t+1} = \lambda_t + \eta\chi_t\|x\|^2 n_{ntp}^{-1}(\eta\chi_t\lambda_t - 4n_{sp}^{-1}(\chi_t + y)).$$

**Proposition C.18** (**Characterizing loss decrease in SP and NTP**). *Let $\eta \geq 0$. For $n_{sp}$ or $n_{ntp}$ large enough, we write the update equations of repeatedly updating the uv-model with SGD on the training point $(x, y)$ with $\|x\| = 1$ in SP or NTP jointly as provided above. The loss decreases at any step, omitting time $t$,*

1. *in the case $f(f - y) \geq 0$, if and only if $\eta \leq \frac{2}{n_{sp}\lambda} + O(n_{sp}^{-3}n_{ntp}^{-1})$ or $\eta \in [\frac{n_{sp}n_{ntp}\lambda}{\|x\|^2\chi f} - \frac{2}{n_{sp}\lambda} - O(n_{sp}^{-3}n_{ntp}^{-1}), \frac{n_{sp}n_{ntp}\lambda}{\|x\|^2\chi f}]$,*
2. *in the case, $f(f - y) < 0$, if and only if $\eta \leq \frac{2}{n_{sp}\lambda} - O(n_{sp}^{-3}n_{ntp}^{-1})$.*

*It holds that $\lambda_{t+1} \geq \lambda_t$ if and only if $\lambda_t \geq \frac{4}{n_{sp}\eta_t}(1 + \frac{y}{\chi_t})$.*

**Remark C.19** (**Instability in SP**). The crucial insight from Proposition C.18 for SP is that both loss and sharpness increase early in training as soon as $\eta = \omega(n^{-1})$, unless an extensively large learning rate that depends on the current sharpness, training point and output function, is accurately chosen at each time step in a slim interval of benign large learning rates, which is unlikely to hold in practice. Figure C.2 shows that in simulated training with constant learning rates, the maximal stable learning rate indeed scales as $\Theta(n^{-1})$. This instability prediction is in line with the infinite-width prediction from Yang and Hu (2021), and hence does not explain large learning rate stability in SP in practice. Figure C.3 shows that the learning rate scaling $\eta_n = \eta_0 \cdot n^{-1}$ induces an asymptotically width-independent instability threshold for $\eta_0$, asymptotically constant sharpness in the kernel regime, and that small widths diverge at smaller $\eta_0$ due to diverging sharpness. ◄

*Proof.* From the update equations, it holds that $\lambda_{t+1} \geq \lambda_t$ if and only if $\eta_t n_{ntp}\chi_t(\eta_t\chi_t\lambda_t - \frac{4}{n_{sp}}f_t) \geq 0$ if and only if $\lambda_t \geq \frac{4}{n_{sp}\eta_t}(1 + \frac{y}{\chi_t})$.

Observe that the loss decreases if and only if $|\chi_{t+1}| \leq |\chi_t|$, which holds if and only if $|1 - n_{sp}\eta\lambda_t + \eta^2 n_{ntp}^{-1}\chi\|x\|^2(\chi_t + y)| \leq 1$, which can be written as, omitting all subscripts $\cdot_t$,

$$\eta^2 n_{ntp}^{-1}\|x\|^2\chi f - \eta n_{sp}\lambda \in [-2, 0].$$

Assuming $\eta \geq 0$, the above holds if and only if $\left(\eta \chi f \leq \frac{n_{sp}n_{ntp}\lambda}{\|x\|^2} \text{ and } n_{ntp}^{-1}\eta^2\|x\|^2\chi f - \eta n_{sp}\lambda \geq -2\right)$. The first constraint is a mild one that states $\eta = O(n)$. We will now focus on the second one. Solving for the roots of this polynomial in $\eta$, we get

$$\eta_{1,2} = \frac{1}{2\|x\|^2\chi f}\left(n_{ntp}n_{sp}\lambda \pm \sqrt{n_{ntp}^2 n_{sp}^2\lambda^2 - 8\|x\|^2 n_{ntp}\chi f}\right).$$

Assuming $n_{sp}^2 n_{ntp}\lambda^2 \gg 8\|x\|^2\chi f =: C$, we get $n_{sp}n_{ntp}\lambda\sqrt{1 - \frac{C}{n_{sp}^2 n_{ntp}\lambda^2}} \approx n_{sp}n_{ntp}\lambda(1 - \frac{C}{2n_{sp}^2 n_{ntp}\lambda^2} - \frac{1}{4}(\frac{C}{n_{sp}^2 n_{ntp}\lambda^2})^2)$. In that case $\eta_1 \approx \frac{2}{n_{sp}\lambda}$ and $\eta_2 \approx \frac{n_{sp}n_{ntp}\lambda}{\|x\|^2\chi f} - \frac{2}{n_{sp}\lambda}$.

Hence, if $\chi f \geq 0$, we get loss decrease if $\eta \leq \frac{2}{n_{sp}\lambda} + O(n_{sp}^{-3}n_{ntp}^{-1})$ or $\eta \in [\frac{n_{sp}n_{ntp}\lambda}{\|x\|^2\chi f} - \frac{2}{n_{sp}\lambda} - O(n_{sp}^{-3}n_{ntp}^{-1}), \frac{n_{sp}n_{ntp}\lambda}{\|x\|^2\chi f}]$.

If $\chi f < 0$, we get loss decrease if $\eta \in [\frac{n_{sp}n_{ntp}\lambda}{\|x\|^2\chi f} - \frac{2}{n_{sp}\lambda} + O(n_{sp}^{-3}n_{ntp}^{-1}), \frac{2}{n_{sp}\lambda} - O(n_{sp}^{-3}n_{ntp}^{-1})]$, where the left end of the interval is negative. The upper end resembles the edge of stability that vanishes as $n_{sp}^{-1}$ for SP but not for NTP.

$\square$

Note the interesting slim regime of benign large learning rates $\eta \approx \frac{n\lambda}{\|x\|^2\chi f} - \frac{1}{n_{sp}\lambda} = \Theta(n)$ when $f(f - y) > 0$. As all of the involved quantities are known at training time, an adaptive learning rate schedule may significantly speed up training by stable learning with excessive learning rates. However it remains unclear whether a similar regime exists in practical architectures under CE loss. In that case, the sharpness computation would also be much more computationally expensive.

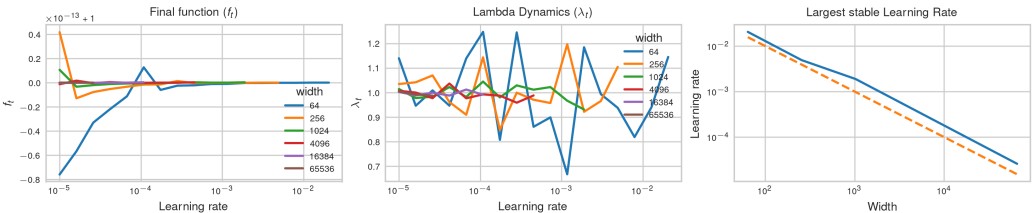

Figure C.2: **Stable SP.** $f_t$ (left) and $\lambda_t$ (center) after training to convergence for several widths. The largest stable learning rate for SP indeed scales as $n^{-1}$ (*right*). When lines end, training diverged for larger learning rates. The first subplot shows that training has succeeded in memorizing the training label $y = 1$ at the optimal learning rate at all widths. The second subplot shows that the randomness in $\lambda_t$ due to random initial conditions vanishes with increasing width, as SP is approaching its kernel regime.

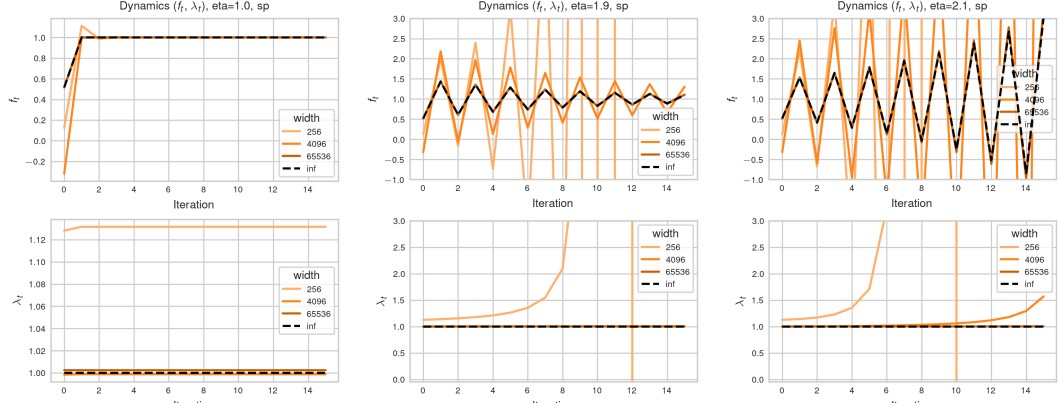

Figure C.3: **Stable SP dynamics.** $f_t$ (top) and $\lambda_t$ (bottom) over the course of training in SP with $\eta_n = \eta_0 \cdot n^{-1}$ for several widths, where $\eta_0 = 1$ in the gradient flow regime (left), $\eta_0 = 1.9$ close to the edge of stability (center) and $\eta_0 = 2.1$ above the edge of stability (right). Smaller widths tend to diverge at smaller learning rates due to sharpness divergence due to the non-vanishing term in the sharpness update. The learning rate scaling $\eta_n = \Theta(n^{-1})$ leads to asymptotically consistent dynamics. This shows that larger learning rate scaling would induce divergence. Asymptotically SP with $\eta_n = \Theta(n^{-1})$ lies in the kernel regime and the sharpness stays constant over the course of training. Edge of stability oscillations do not enable larger learning rate scaling under MSE loss.

# D    Experimental details

If not otherwise specified, we train a single epoch to prevent confounding from multi-epoch overfitting effects.

## D.1    MLPs

We implement our MLP experiments on MNIST (Deng, 2012) and CIFAR-10 (Krizhevsky et al., 2009) in PyTorch (Paszke et al., 2019). We train ReLU MLPs with the same width $n$ in all hidden dimensions with plain SGD/Adam with a single learning rate for all trainable parameters, batch size 64 without learning rate schedules, weight decay or momentum to prevent confounding. We use Adam with the PyTorch standard hyperparameters. By standard initialization we mean He initialization variance $c_\phi/\texttt{fan\_in}$ with $c_\phi = 2$ for the ReLU activation function (He et al., 2015).

## D.2    Multi-index data

For some experiments in Appendix F, we generate multi-index teacher data, inspired by Kunin et al. (2024), but setting a deterministic teacher for ensuring a balanced classification task.

We draw the covariates $\xi \sim \mathcal{U}(\mathbb{S}^{d_{in}-1})$ i.i.d. from the uniform distribution on the unit sphere in $\mathbb{R}^{d_{in}}$ with input dimension $d_{in} = 100$. The training set consists of $10^3$ training points. We also draw a test set consisting of $10^4$ test points.

For the target function $f^*$, drawing 3 random directions as in Kunin et al. (2024) results in heavily unbalanced classes and $f^* = 0$ on large part of the support with high probability. Instead, we set 4 teacher neurons deterministically for less noisy results. The teacher net is a shallow ReLU network given by $f^*(\xi) = \text{sign}(\sum_{i=1}^4 s_i \phi(w_i^\top \xi))$ with unit vectors $w_1 = e_1$, $w_2 = e_2$, $w_3 = -e_1$, $w_4 = -e_2$ and signs $s_1 = s_3 = +1$ and $s_2 = s_4 = -1$. This results in the nonlinear target function $f^*(\xi) = \text{sign}(\xi_1 - \xi_2)$ for all $\xi \in \mathbb{R}^{d_{in}}$ with $\xi_1 > 0$ or $\xi_2 > 0$, but $f^*(\xi) = \text{sign}(\xi_2 - \xi_1)$ for all $\xi \in (-\infty, 0) \times (-\infty, 0)$. We do not use label noise.

This dataset requires learning to align with the first 2 covariate dimensions $(\xi_1, \xi_2)$, where all of the signal for the labels $f^*(\xi)$ lies. If the input layer does not learn to align with these dimensions, the sparse signal is obscured in the activations (random features) after the first layer due to the large variance in the remaining covariate dimensions.

### D.3 Language modeling

We train small Transformer models (Vaswani et al., 2017) using LitGPT (Lightning AI, 2023). We adapt the Pythia (Biderman et al., 2023) architecture with 6 Transformer blocks, standard $\texttt{d\_head}^{-1/2}$ attention scaling, pre-attention and qk-Layernorm (Wortsman et al., 2024). We purely scale width, proportionally scaling the number of attention heads and the MLP hidden size while keeping the number of layers and head dimension $\texttt{d\_head} = 32$ fixed. For widths 256, 1024 and 4096, this results in 8, 32 and 128 heads per Transformer block and a total of $30M$, $167M$ and $1.4B$ parameters.

Standard training means AdamW with a single, tuned maximal learning rate, $(\beta_1, \beta_2) = (0.9, 0.95)$, $\varepsilon = 10^{-12}$, sequence length 512, batch size 256, 700 steps of warmup followed by cosine learning rate decay to $10\%$ of the maximal learning rate, weight decay 0.1, gradient clipping. We train for 10681 steps in mixed precision on the DCLM-Baseline dataset (Li et al., 2024). We train all models on the same number of tokens to prevent confounding effects from increased training time.

### D.4 Figure Details

**Figure 1**: The training accuracy of 8-layer MLPs is averaged over 4 runs to reduce noise from random initialization. The training loss of GPT trained with SGD is averaged over 3 runs. GPT with Adam was only run once.

**Figure 2**: In the left subplot, $c_l = 1$ for normalization layers, since they act like diagonal matrices and do not accumulate $\texttt{fan\_in}$-dependent scaling. For all other layers, we expect LLN-like scaling $c_l = \texttt{fan\_in}$, which is the length of the inner products between weight updates and incoming activations. For determining the correct layerwise learning rate scaling, what matters is how much of the expected alignment exponent is lost at finite width in a single update step. For the first step, we measure the barely width-dependent exponents 0.01, $-0.08$ and $-0.08$ for the last Layernorm, the MLP and readout layer, respectively, suggesting that maximal width-dependent alignment approximately holds, so that infinite-width scaling predictions are useful for determining layerwise learning rate scaling rules. The readout layer and last Layernorm layer are chosen due to their particular importance for logit blowup. The MLP layer was chosen to add a layer that scales hidden-like. This layer was not cherry picked. We observe other MLP layers to have similar scaling properties.

**Figure 4**: 3-layer MLP trained with SGD on CIFAR-10 with width-dependent learning rate $\eta_n = 0.0001 \cdot n^{-0.5}$. Averages over 4 random seeds.

**Figure 5**: Learning rate sweeps for GPT. SGD runs are averaged over 3 random seeds, due to noisy individual outcomes. The results for Adam stem from a single random seed due to limited computational resources. Minimal unstable learning rates are defined as the smallest learning rates to produce loss worse than (optimal CE loss +1) at each width. The x-axes showing learning rates are scaled as $(n/256)^\alpha$. In this way, the learning rate at base width 256 remains the same for comparability of the constants. If the optimal or maximal stable learning rate indeed scales as $\eta_n = \eta \cdot n^{-\alpha}$, then the width-dependent scaling of the x-axis $\eta_n \cdot n^\alpha$ shows learning rate transfer.

**Figure 6**: Exponents are based on the learning rate curves provided in Appendix F.2 for GPT on DCLM-Baseline and in Appendix F.3 for 8-layer MLPs on image data sets. For MLPs on MNIST and CIFAR-10, we measure minimal unstable learning rates as the smallest learning rate larger than the optimal one to produce NaN entries when using MSE loss, or accuracy $< 20\%$ under CE loss. For GPT, we measure minimal unstable learning rates as the smallest learning rates that are larger than the optimal one to produce loss worse than (optimal CE loss +1) at each width.

**Figure 7**: Shown is the training accuracy at the end of training for one epoch for the optimal learning rate at each width.

## E Refined coordinate checks

The standard coordinate check as provided in the readme of the $\texttt{mup}$-package Yang et al. (2022) may be considered the plot of activation norms $\|x_t^l\|_{RMS}$ after $t$ steps of training for all layers $l$ and the network output norm $\|f\|_{RMS}$ with $f := W_t^{L+1} x_t^L$ as a function of width. Completely width-independent dynamics under $\mu$P then result in an approximately width-independent coordinate check of all layers. However, width-dependence in the activations of previous layers would confound the $l$-th layer activation scaling, so that measuring the effective $l$-th layer updates requires measuring

$\|\Delta W_t^l x_t^l\|_{RMS}$ in each layer, where one may be interested in the weight updates accumulated over the entire course of training $\Delta W_t^l = W_t^l - W_0^l$ or the update in a single step $\delta W_t^l = W_t^l - W_{t-1}^l$. In standard architectures, one can equivalently measure the operator norm of the weight updates $\sqrt{\texttt{fan\_in}(W_t^l)/\texttt{fan\_out}(W_t^l)} \cdot \|\Delta W_t^l\|_{2\to 2} \overset{!}{=} \Theta(1)$ (Yang et al., 2023a); however in non-standard architectures such as Mamba this spectral condition has been shown to fail, so that, in the general case, care should be taken in how exactly weight updates affect the output function (Vankadara et al., 2024). The difference between $\|\Delta W_t x_t\|$ and the preactivation updates $\|\Delta(W_t x_t)\|$ is precisely $\|W_0 \Delta x\|_{RMS}$ which measures the effect of updates propagating from previous layers.

All coordinate checks are run over 4 random seeds either at small learning rate or the optimal learning rate $\eta_{256}$ at base width 256 (after 1 epoch of training) on CIFAR-10. The learning rate is then scaled in relation to that base width $\eta_n = \eta \cdot (n/256)^\alpha$ with a clean exponent $\alpha \in \{-1, -0.5, 0\}$.

### E.1 SGD

Figure E.1 shows the refined coordinate check for a 3-layer MLP in SP with global learning rate scaling $\eta_n = \eta \cdot n^{-1/2}$. As predicted by Proposition 4, the input layer updates decay as $n^{-1}$, the hidden layer learns features width-independently, and the output scales as $n^{1/2}$ which results in one-hot predictions after the softmax in wide models, but not necessarily unstable training dynamics.

Both $\|\Delta W_t^l x_t^l\|_{RMS}$ and $\|\Delta W_t^l\|_{RMS \to RMS}$ measure the effective update effect in the $l$-th layer equivalently and accurately even in narrow MLPs of width 64. Naively tracking the activation updates $\Delta x_t^l = x_t^l - x_0^l$ however is confounded by non-vanishing feature learning in narrow models, and only shows the correct hidden- and last-layer scaling exponents for $n \geq 4000$, even after only a single update step.

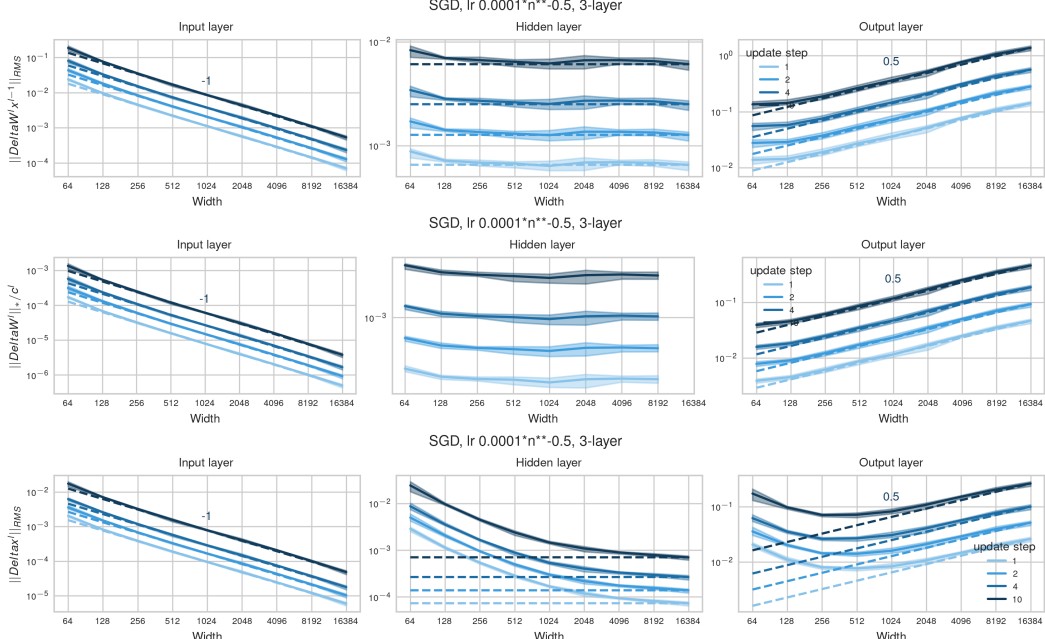

Figure E.1: **(Hidden layer feature learning in SP under intermediate learning rate scaling)** Effective $l$-th layer update scalings $\|\Delta W_t^l x_t^{l-1}\|_{RMS}$ (top), weight update spectral norm $\|\Delta W_t^l\|_*$ (2nd row) and activation updates $\delta x^l$ (bottom) of MLPs trained in SP with small learning rate $\eta_n = 0.0001 \cdot (n/256)^{-1/2}$ scaled to preserve hidden-layer feature learning. The TP scaling predictions are accurate. Hidden layers learn features width-independently, and input layers have vanishing feature learning. At moderate widths, activation updates are confounded by previous layer updates, and thus do not provide an accurate metric for effective update scaling.

Figure E.2 shows a refined coordinate check for a 3-layer MLP in SP with width-independent global learning rate scaling $\eta_n = 0.0001 \cdot n^0$. While infinite-width theory predicts the input layer to learn width-independently and the hidden layer to explode as $\Theta(n)$, both empirical exponents are $n^{-1/2}$ smaller, so that the input layer has vanishing feature learning and the hidden layer is still exploding. This ostensible contradiction is resolved when repeating the coordinate check but initializing the last layer to $0$ (Figure E.3). Now the predicted scaling exponents are recovered, already at small width. The reason for this subtle but important difference is that the gradient that is back-propagated is given by the last-layer weights, $\partial f/\partial x^L = W_t^{L+1} = W_0^{L+1} + \Delta W_t^{L+1}$. Under standard initialization at the optimal learning rate, the initialization $W_0^{L+1} = \Theta(n^{-1/2})$ still dominates the updates $\Delta W_t^{L+1} = \Theta(\eta_n)$ in absolute terms after a few update steps at widths up to $16384$. Comparing the absolute scales of $\|\Delta W_t^l x_t^l\|_{RMS}$ or $\|W_t^l\|_*$ in both figures confirms this hypothesis. The pure update effects in Figure E.3 have lower order of magnitude in the constant before the scaling law, but follow clear scaling exponents. Therefore the faster scaling law under last-layer zero initialization can be extrapolated with certainty to induce a phase transition under standard initialization around width $4 \cdot 10^7$. We do not have sufficient computation resources to validate this but arrive at this order of magnitude irrespective of whether we extrapolate the scaling laws of $\|\Delta W_t^l x_t^l\|_{RMS}$ or $\|W_t^l\|_*$ as well as of the input or hidden layer laws. For base width $n_0$ and width-dependent statistics $\Delta_n^1$ and $\Delta_n^2$ with differing scaling exponents $c_1$ and $c_2$, $\Delta_n^1$ and $\Delta_n^2$ intersect at width $n_0 \cdot (\Delta_{n_0}^2/\Delta_{n_0}^1)^{1/(c_1-c_2)}$.

This consequential difference in empirical scaling exponents at realistic widths due to a subtle difference in last-layer initialization highlights the attention to detail that is required to make accurate scaling predictions from infinite-width limit theory, but, as we show in this paper, apparent contradictions can often be reconciled with enough attention to detail, and the clean scaling laws we arrive at as a result already hold at moderate scales and prove the usefulness of investing this extra effort.

**Hence, one reason why scaling exponents in SGD can be larger than predicted up to very large widths, is due to differing orders of magnitude in the constant pre factors in the initialization versus update terms in the backward pass. Without our refined coordinate check, the phase transition around width $10^7$ is hard to predict.**

As predicted, the width-exponents of 2-layer nets behave like the input and output layer in 3-layer nets (Figure E.4).

When choosing the optimal learning rate $\eta_{256} = 0.03$ at width $256$, stronger finite-width effects due to non-vanishing input layer feature learning already occur after a few steps and make the update scaling exponents after 10 steps only visible at larger width $n \geq 2048$ (Figure E.5). As long as divergence is prevented in the first few steps, self-stabilization mechanisms such as activation sparsification can quickly contain the initial catapult (Figure E.6). In deeper networks, explosion of several hidden layers is increasingly difficult to stabilize, and finite width effects are reduced.

Figure E.6 shows the effective update rank and the alignment between activations at initialization versus at time $t$ for the same input training points under unstable width-independent learning rate scaling. The updates in each layer are remarkably strongly dominated by a single direction. As hidden-layer activations are slowly diverging, their alignment is only beginning to decrease at large widths $n \geq 4096$. The beginning instability of $\|\Delta x^2\|_{RMS}$ will eventually induce training instability and suboptimal accuracy at large width, which is hard to predict without tracking the layerwise effective update scaling across widths.

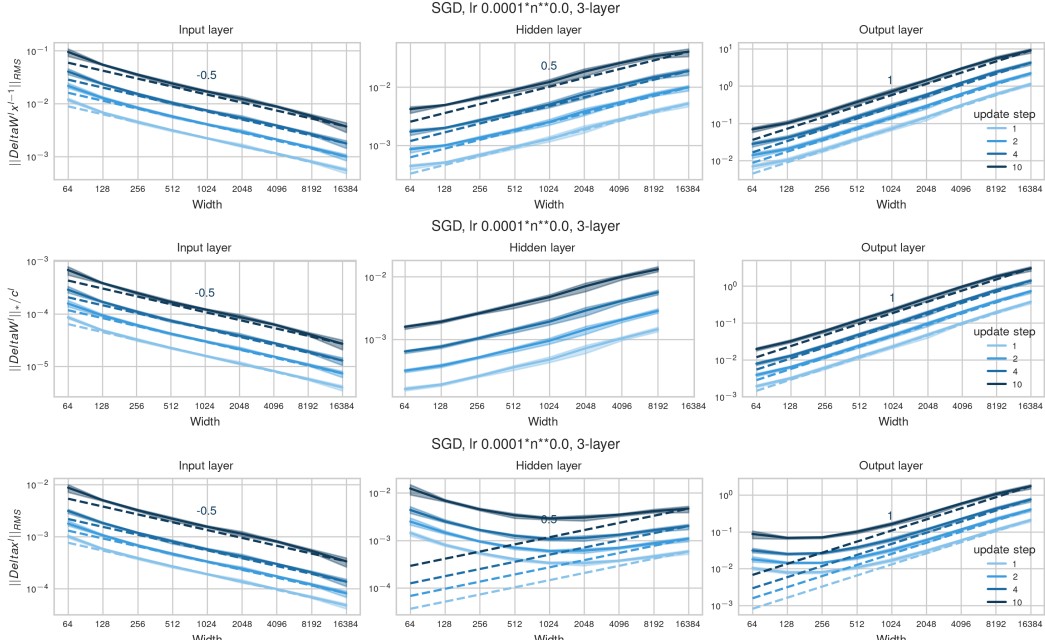

Figure E.2: (**Inaccurate exponent predictions under standard initialization with large learning rate scaling**) Effective $l$-th layer update scalings $\|\Delta W_t^l x_t^{l-1}\|_{RMS}$ (top), weight update spectral norm $\|\Delta W_t^l\|_*$ (2nd row) and activation updates $\delta x^l$ (bottom) of 3-layer MLPs trained in SP with width-independent $\eta_n = 0.0001$. Hidden layer activation updates explode, and input layers have vanishing feature learning. By TP scaling predictions, however, the input layer should learn features width-independently. Instead, the TP scaling exponents are **only accurate under last-layer zero initialization, not under standard initialization** (see Figure E.3 for last-layer zero initialization) as the initialization scaling $W_0^{L+1} = \Theta(n^{-1/2})$ still dominates the update scaling $\Delta W_t^{L+1} = \Theta(\eta_n)$ at realistic widths after a few updates under the optimal learning rate. Hence, the backpropagated gradient $\partial f / \partial x^L = W_t^{L+1}$, relevant for the hidden and input layer updates, behaves for a several steps like it should only behave in the first step. By comparing the absolute scales here versus those in Figure E.3 it becomes apparent that this is indeed a finite-width effect, as the absolute scale of $\|\Delta W x\|_2$ here is on the order $10^{-1}$ and $10^{-2}$ for input and hidden layer, respectively, whereas the pure update effects under last-layer zero initialization are of at most order $10^{-4}$ for both layer types. Clearly for sufficient width, the differing scaling exponents will induce a phase transition toward the predicted scaling exponents. While the input layer learns features width-independently under last-layer zero initialization, as predicted by TP theory, this is not the case at realistic scales under standard initialization. The qualitative statement that standard parameterization with width-independent learning rates is not activation stable in deep networks is still accurate at moderate width.

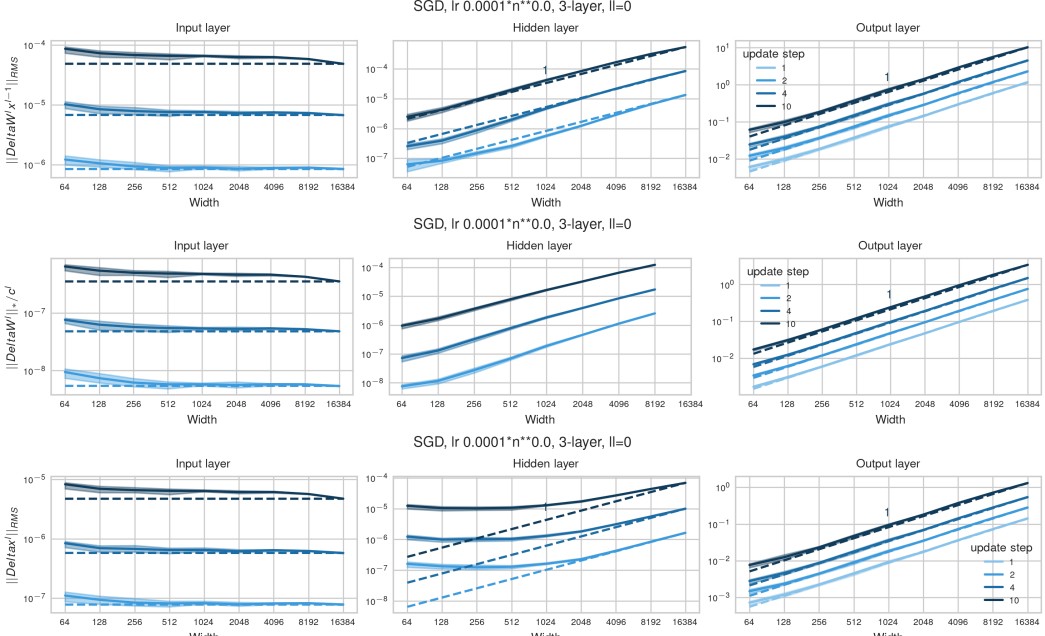

Figure E.3: (**Accurate exponent predictions in SP with last-layer zero initialization under large learning rate scaling**) Same as Figure E.2 with width-independent $\eta_n = 0.0001$ but initializing the last layer to zero. Here, the TP scaling predictions are accurate. Hidden layer activation updates explode as $n^1$, and input layers learn features width-independently. Observe a smaller absolute scale of the pure update effects here versus in Figure E.1 that explains the differing exponents there. The updates in the input and hidden layers vanish in the first step, as the gradient for backprop is $W_0^{L+1} = 0$.

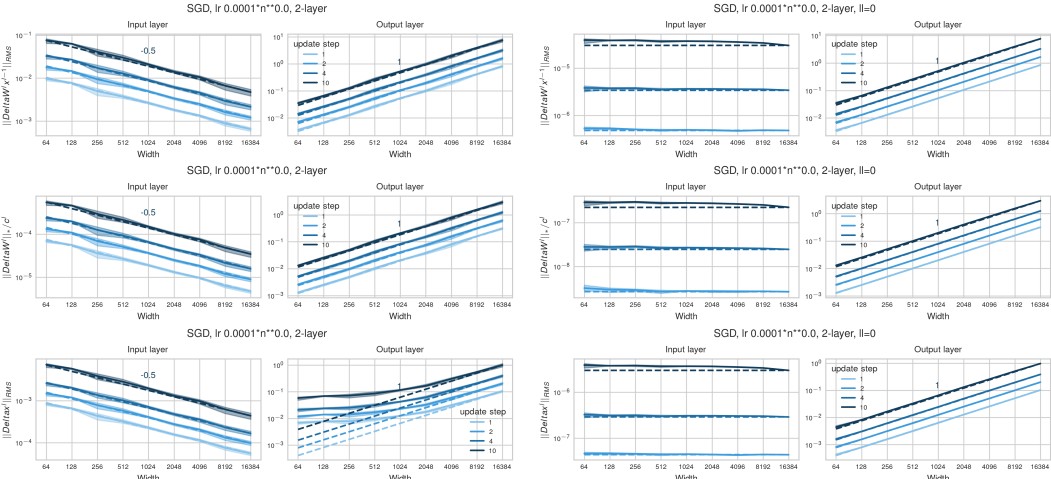

Figure E.4: (**Shallow nets learn features width-independently under large learning rate scaling**) Same as Figure E.1 but for 2-layer MLPs trained in SP with width-independent $\eta_n = 0.0003$ with standard initialization (left) and last-layer initialized to $0$ (right). The input layer and output layer scalings behave as in the 3-layer nets. Since there is no exploding hidden layer, activation stability is preserved in 2-layer nets under $\eta_n = \Theta(1)$.

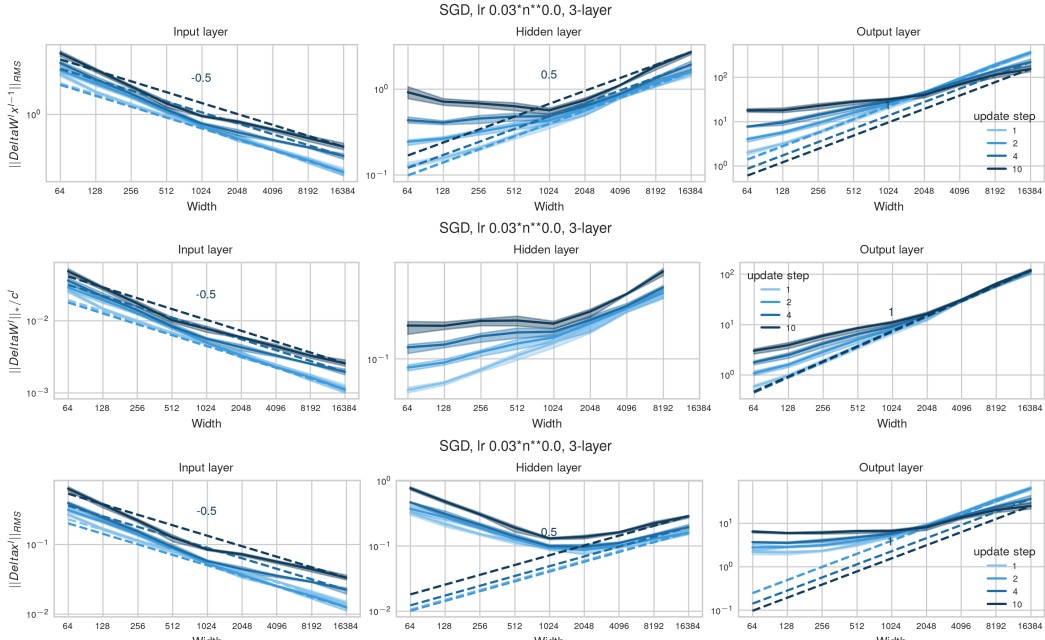

Figure E.5: (**Large finite-width effects at optimal learning rate in shallow 3-layer MLPs**) At the optimal learning rate $\eta_{256} = 0.03$ with width-independent scaling, non-vanishing input layer feature learning confounds the scalings after few update steps up to moderate widths $n \leq 1024$, similar to Adam (Figure E.9).

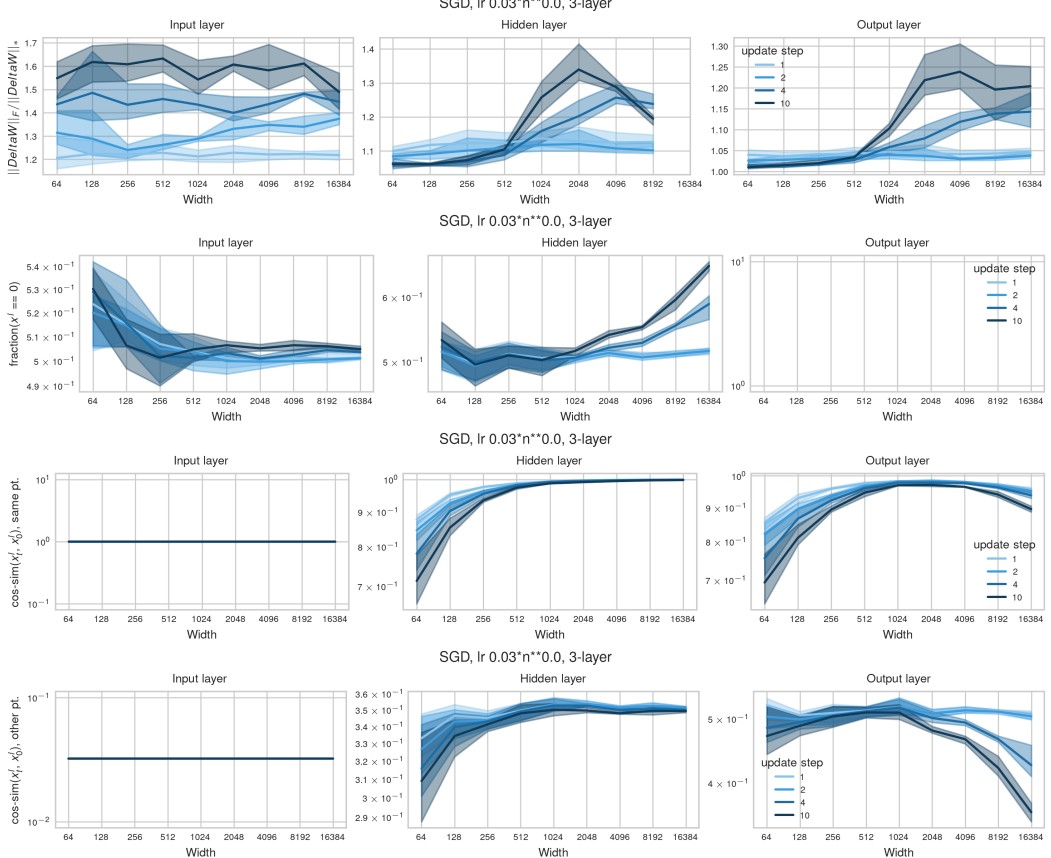

Figure E.6: **(Activation sparsification at the optimal learning rate)** Effective $l$-th layer update ranks $\|\Delta W_t^l\|_F / \|\Delta W_t^l\|_*$, activation sparsity and cosine similarity between activations to each layer comparing time $0$ and time $t$ on the same input training point and on differing training points in the same batch of 3-layer MLPs trained with SGD in SP with width-independent learning rate $\eta_n = 0.03$ as in Figure E.5. As opposed to the gradient flow regime, at the optimal learning rate, there are significant self-stabilization effects at large width already after 10 steps through activation sparsification but less through activation rotation.

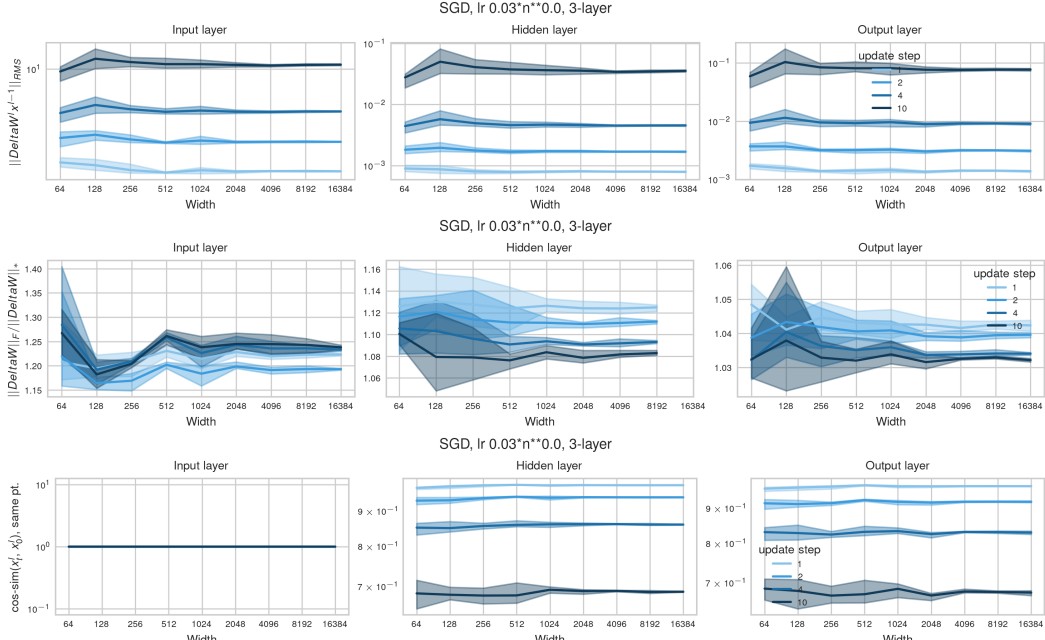

Figure E.7: **(Full width-independence in $\mu$P)** Effective $l$-th layer updates (top), effective update ranks $\|\Delta W_t^l\|_F/\|\Delta W_t^l\|_*$ (second row) and cosine similarity between activations to each layer comparing time $0$ and time $t$ on the same input training point (bottom) of 3-layer MLPs trained with SGD in $\mu$P with width-independent learning rate $\eta_n = 0.03$. As expected, all statistics behave width-independently. The effective update rank is remarkably small, as for SP. The activation are rotated quite quickly.

## E.2 Adam

With $\eta_n = \Theta(n^{-1/2})$, the optimal learning rate scaling for 3-layer MLPs with Adam on CIFAR-10 is larger than predicted (Figure F.30). Figure E.9 shows that this may be due to large finite-width effects for Adam at optimal learning rate multiplier $\eta_{256} = 0.0003$ and moderate width $n \leq 8192$. While the weight update spectral norm scales as predicted, the input-layer gets large updates at moderate width (Figure E.10) and induces a strong rotation of the activations. As a result, the activation explosion only sets in at large width $n \geq 8192$. This qualitative change toward vanishing input layer feature learning will result in a phase transition toward unstable scaling at large widths which is hard to predict at small scale from measurements alone, except when measuring both $\|\Delta W^l\|_*$ and the alignment $\|\Delta W^l x^{l-1}\|_{RMS}$.

As opposed to SGD, observe large finite-width effects in the activation updates even under small absolute learning rate $10^{-6}$ at moderate width $n \leq 8192$ (Figure E.8).

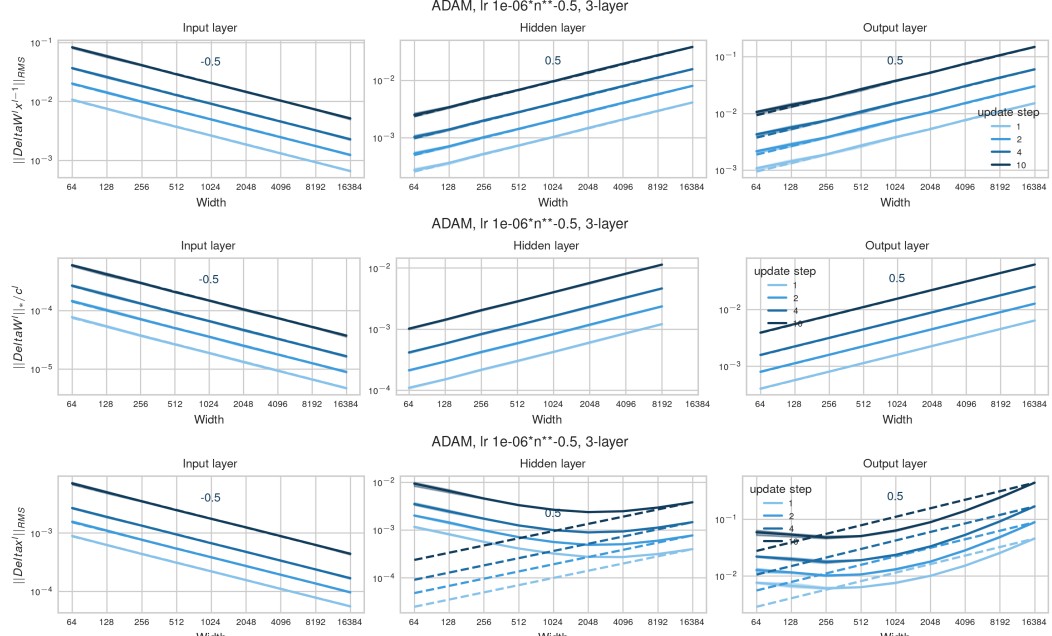

Figure E.8: **(Large finite-width effects from input-layer updates in Adam)** Effective $l$-th layer update scalings $\|\Delta W_t^l x_t^{l-1}\|_{RMS}$ (top), weight update spectral norm $\|\Delta W_t^l\|_*$ (2nd row) and activation update norm $\|\delta x^l\|_{RMS}$ (bottom) of 3-layer MLPs trained with Adam in SP with $\eta_n = 10^{-6} \cdot n^{-1/2}$. Observe the theoretically predicted exponents in $\|\Delta W^l\|_*$ do not transfer to the activation updates at moderate width $n < 8192$ due to large non-vanishing input layer updates at moderate width. Even the effective updates $\|\Delta W_t^l x_t^{l-1}\|_{RMS}$ do not perfectly align with the scaling law at infinite width, indicating that the alignment between $\Delta W_t^l$ and $x_t^{l-1}$ evolves non-trivially across width and that the spectral norm $\|\Delta W^l\|_*$ and pure infinite-width predictions are less useful for explaining the behaviour of Adam at moderate width.

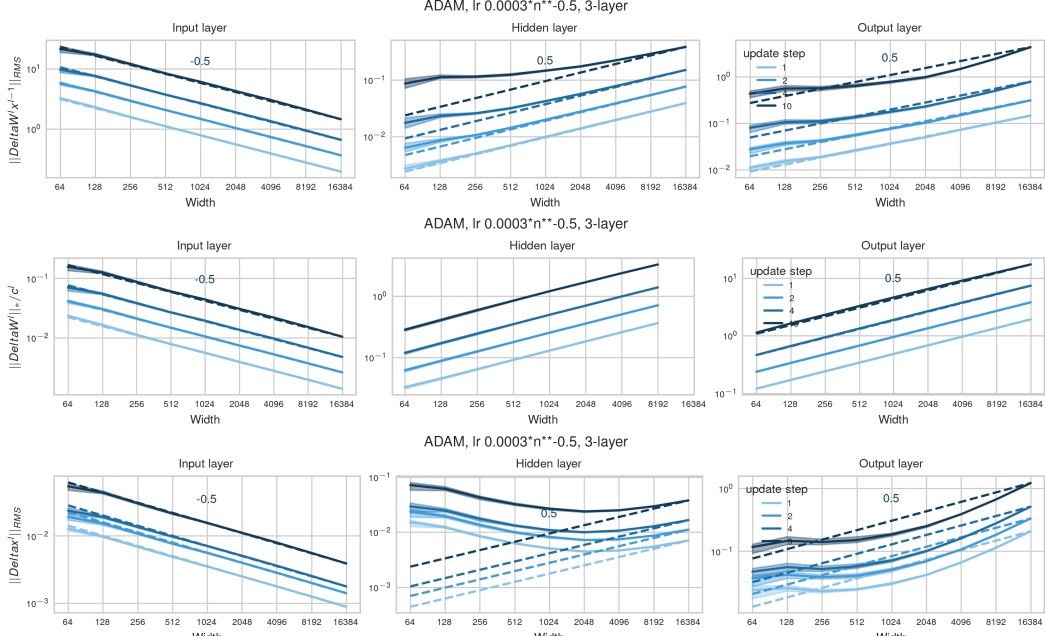

Figure E.9: **(Large finite-width effects from input-layer updates in Adam)** Effective $l$-th layer update scalings $\|\Delta W_t^l x_t^{l-1}\|_{RMS}$ (top), weight update spectral norm $\|\Delta W_t^l\|_*$ (2nd row) and activation update norm $\|\delta x^l\|_{RMS}$ (bottom) of 3-layer MLPs trained with Adam in SP with large $\eta_n = 0.0003 \cdot n^{-1/2}$. Observe the theoretically predicted exponents in $\|\Delta W^l\|_*$ do not transfer to the activation updates at moderate width $n < 8192$ due to large non-vanishing input layer updates at moderate width. Even the effective updates $\|\Delta W_t^l x_t^{l-1}\|_{RMS}$ do not perfectly align with the scaling law at infinite width, indicating that the alignment between $\Delta W_t^l$ and $x_t^{l-1}$ evolves non-trivially across width and that the spectral norm $\|\Delta W^l\|_*$ and pure infinite-width predictions are less useful for explaining the behaviour of Adam at moderate width.

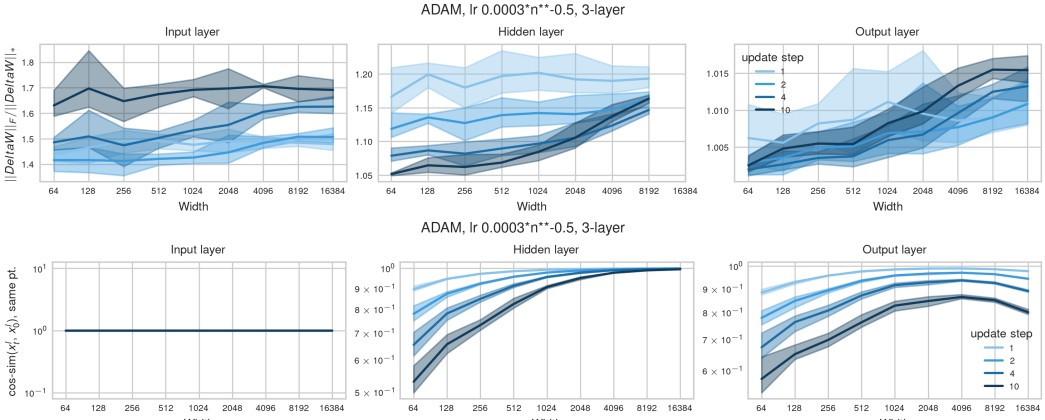

Figure E.10: **(Strong activation rotation under Adam at moderate width)** Effective $l$-th layer update ranks $\|\Delta W_t^l\|_F / \|\Delta W_t^l\|_*$ (top) and cosine similarity between activations to each layer comparing time 0 and time $t$ on the same input training point (bottom) of 3-layer MLPs trained with ADAM in SP with large $\eta_n = 0.0003 \cdot n^{-1/2}$. The effective update rank is mostly growing in time in the input layer. Already after a few steps, the first-layer activation coordinates are drastically rotated at moderate widths. This induces a u-curve in the hidden-layer activations that inherit large rotation from the input layer at moderate width and update too much at large width under $\eta_n = \Theta(n^{-1/2})$.

## E.3 Normalization layers and Adam provide robustness to miss-initialization

For MLPs trained with SGD, initialization greatly impacts the training dynamics as both the forward and the backward pass are affected (Figure E.11). Large input layer initialization induces update instability at large width, which is stabilized by extreme activation sparsification (Figure E.13).

By adding normalization layers, the forward pass can be enforced to scale width-independently. This may affect the gradients. But the gradient norms become irrelevant under Adam with sufficiently small $\varepsilon$. Adding both normalization layers and Adam to MLPs, observe that initialization is barely relevant for update scalings (Figure E.12), and other downstream statistics such as activation sparsity (Figure E.14). Here we use RMSNorm to fairly compare activation sparsity, but we expect LayerNorm to induce the same scaling behaviour.

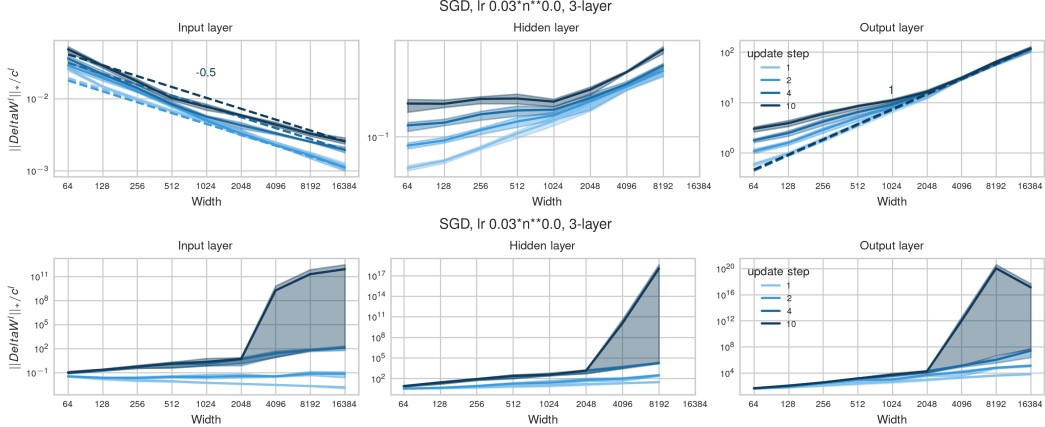

Figure E.11: **(Initialization matters in MLPs with SGD)** SP (top), SP with large input layer variance 2 (bottom). The initializations induce significant differences in the training dynamics. Large input layer normalization becomes unstable at large width.

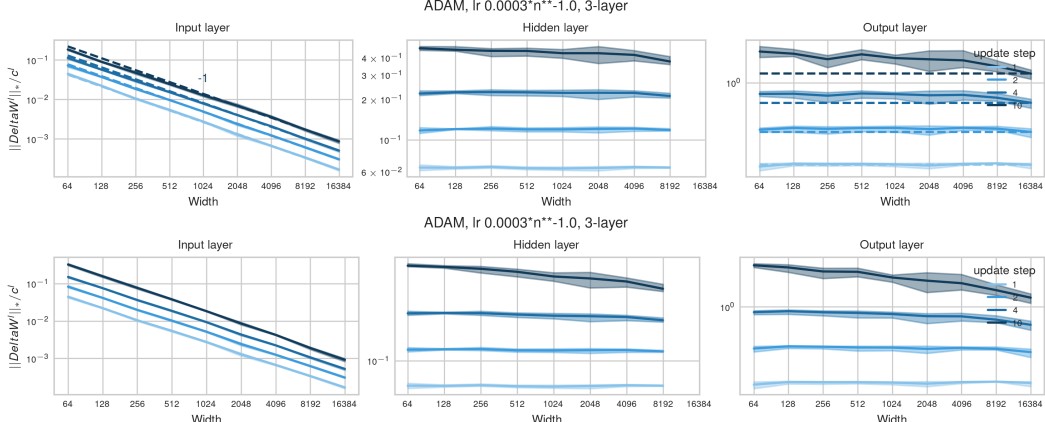

Figure E.12: **(Differing initialization barely matters with normalization layers and Adam)** Update spectral norms of MLPs with the most basic normalization layer RMSNorm after every layer trained with Adam and initialized with SP (top) versus SP with large input layer variance 2 (bottom). Here, initialization barely impacts the update scaling.

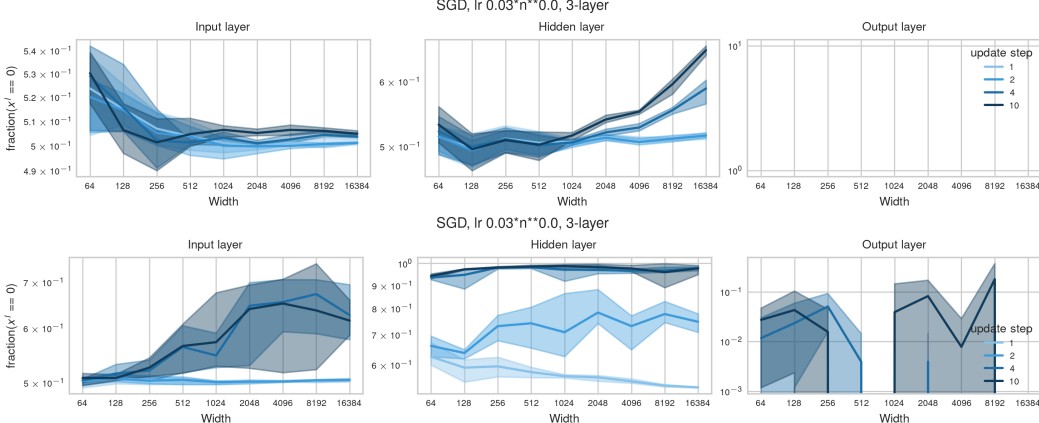

Figure E.13: **(Big difference in activation sparsity under SGD)** SP (top), SP with large input layer variance 2 (bottom). Large input variance has to be stabilized by increased activation sparsity.

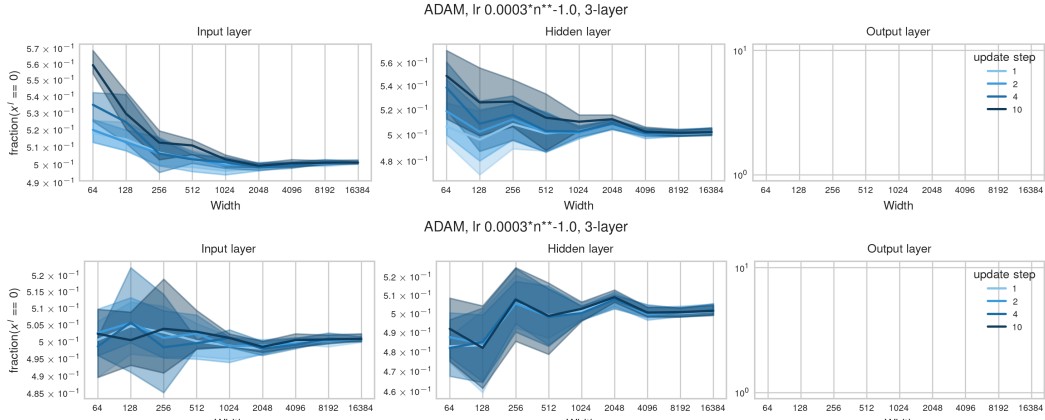

Figure E.14: **(Activation sparsity barely affected under normalization)** Same as Figure E.12 but showing the fraction of activation entries that equal $0$. Both initializations do not significantly sparsify activations beyond $50\%$.

## E.4 Alignment and update scaling in Transformers

For GPT in SP, alignment and update scaling follows the theoretical predictions: Hidden and output layers diverge with width at constant learning rate (Figure E.15, top). At the optimal learning rate where $\eta_n \to 0$ (here only $\eta_n = \Theta(n^{-1})$ is shown), embedding and normalization layer updates vanish with width (Figure E.16 and Figure E.15, bottom).

SP-full-align achieves approximately width-independent signal propagation in the language setting $d_{\text{out}} \gg n$. Since we measure width-independent alignment $\alpha_{W_0^{L+1} \Delta x_t^L} = \Theta(1)$ (Figures 2 and E.18), under large output dimension $d_{\text{out}} \gg n$, the initial output layer operator norm $\|W_0^{L+1}\|_{RMS \to RMS}$ approximately scales $\Theta(1)$ (Vershynin, 2010), as opposed to $\Theta(n^{1/2})$ at sufficient width $n \gg d_{\text{out}}$. The term $W_0^{L+1} \Delta x_t^L$ therefore induces approximately width-independent logit updates even under standard last-layer initialization, in the regime $d_{\text{out}} \gg n$ (cf. Figure E.17), but it induces logit divergence at sufficient width $d_{\text{out}} \ll n$.

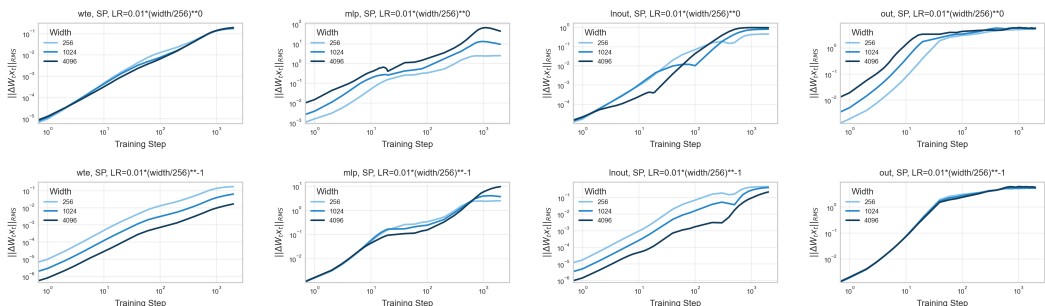

Figure E.15: **(Effective updates follow predictions)** Effective updates $\|\Delta W_t x_t\|$ for constant learning rate scaling $\eta_n = 0.01$ (top) and stable learning rate scaling $\eta_n = 0.01 \cdot (n/256)^{-1}$ (bottom) in GPT models of varying width (the darker, the wider) for the embedding layer, the first MLP layer in the Transformer block 2, the last Layernorm before the readout layer and the readout layer (from left to right). At constant learning rate, hidden and output layers diverge with width. At optimal learning rate, embedding and normalization layer updates vanish with width.

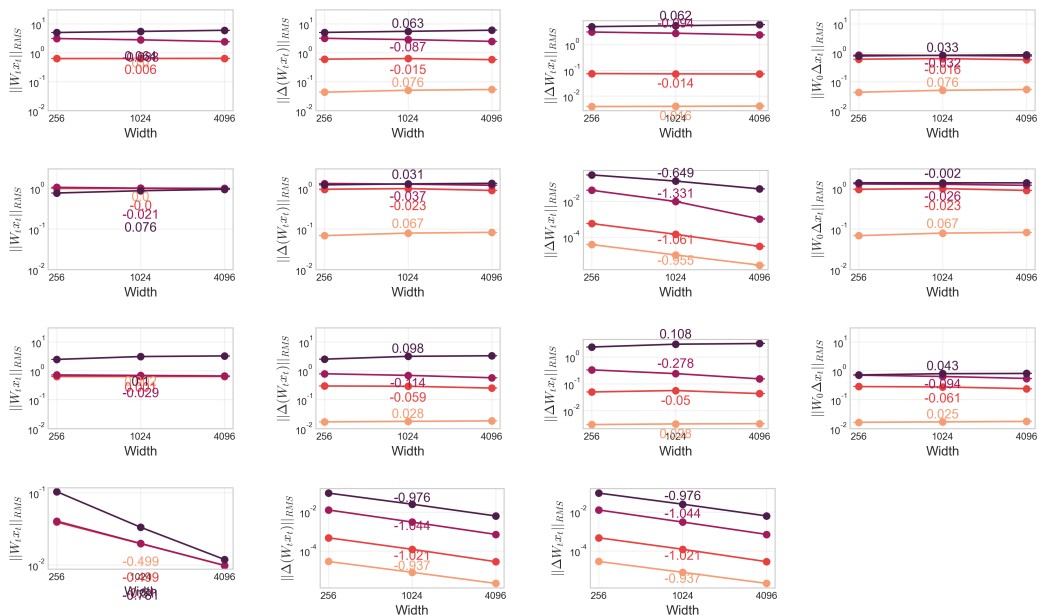

Figure E.16: **(Refined coordinate checks for GPT in SP with Adam and $\eta_n = 0.01 \cdot n^{-1}$)** *From left to right:* Activation norm, activation updates, effective updates $\|\Delta W_t x_t\|_{RMS}$, propagating updates $\|W_0 \Delta x_t\|_{RMS}$ after 2, 10, 100 and 700 batches of training (the darker, the more batches). *Layers from top to bottom:* readout, last Layernorm, first MLP layer in Transformer block 2, embedding layer. Infinite width-scaling predictions are accurate in all effective update terms $\|\Delta W_t x_t\|_{RMS}$: Embedding and Layernorm layers scale input-like and their updates vanish as $\Theta(n^{-1})$, all hidden and output layers are effectively updated width-independently. Against the infinite-width prediction, logit updates do not explode, not because of miss-alignment but because output dimension is much larger than width $d_{out} \gg n$, which changes the approximate scaling of $\|W_0^{L+1}\|_{RMS \to RMS}$ from $\Theta(n^{1/2})$ in the infinite-width limit, to $\Theta(1)$ in the large output dimensional regime.

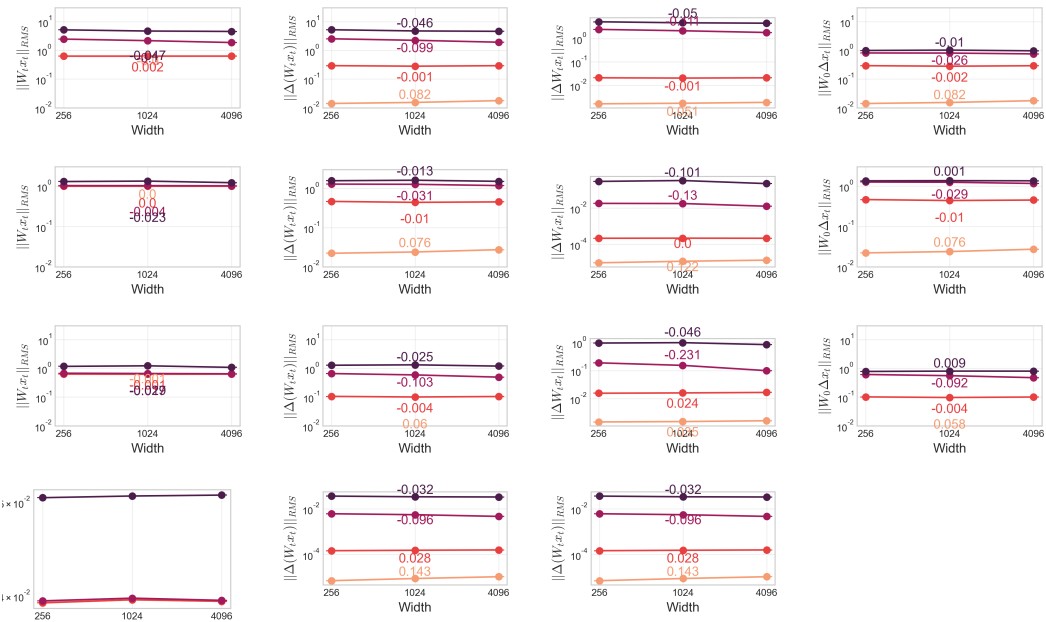

Figure E.17: (**Refined coordinate checks for GPT in SP-fullalign with Adam and width-independent** $\eta_n = 0.003162$) Same as Figure E.16 but for SP-full-align. The propagating update and effective update terms in all layers scale approximately width-independently. The output layer propagating updates do not diverge as in the large width regime $n \gg d_{\text{out}}$ (Figure F.32). Here, instead, due to the large output dimension $d_{\text{out}} \gg n$, $\|W_0^{L+1}\|_{RMS \to RMS}$ is dominated by its other summand, which induces approximately width-independent signal propagation at realistic widths.

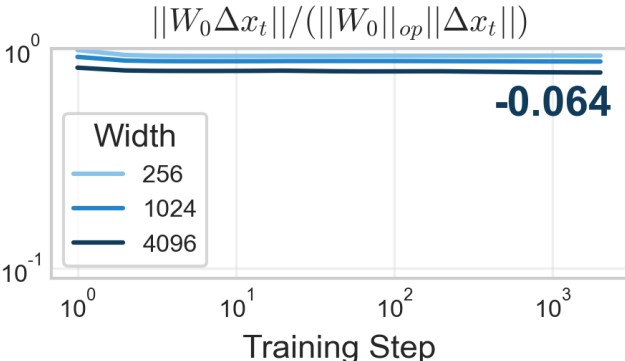

Figure E.18: (**Updates propagate maximally in the readout layer in SP-full-align**) The operator norm ratio for propagating activations in the readout layer for training GPT with AdamW in SP-full-align with near-optimal learning rate $\eta_n = 0.00316$. The ratio is barely width dependent so that propagated activations can be computed when knowing both $\|W_0\|_{op} = \|W_0\|_{RMS \to RMS}$ and $\|\Delta x_t\|_{RMS}$.

# F   Empirical learning rate exponents

## F.1   Summary of the MLP experiments in this section

In general, the optimal learning rate exponent appears to be architecture- as well as data-dependent. We conjecture that the optimal learning rate scaling is subject to opposing objectives. Ideally, the effective updates in all layers scale width-independently. Since this cannot be achieved with a single learning rate for input, hidden and output layers, the layer types act on the optimal learning rate scaling as opposing forces.

**SGD under MSE loss.** For SGD under MSE loss, output blowup results in unstable training dynamics so that the maximal stable and optimal learning rate robustly scales as $\eta_n = \Theta(n^{-1})$ across architectures and datasets. As a consequence of vanishing feature learning, neither training nor test loss monotonically improve with scale under MSE loss.

**Random feature models.** When only training the last layer, fully width-independent training dynamics are achieved with $\eta_n = \eta \cdot n^{-1}$. Figure F.19 shows that this exponent clearly results in learning rate transfer for 2-layer ReLU random feature networks on CIFAR-10. Also observe that since all learning rate scalings recover activation-stability, larger than optimal learning rates still result in non-trivial classification accuracy.

**Deep MLPs.** With an increasing amount of hidden layers, their width-independence eventually outweighs input layer feature learning in vision datasets. For at least 6 layers, we see approximate learning rate transfer under $\eta_n = \Theta(n^{-1/2})$ for SGD and $\eta_n = \Theta(n^{-1})$ for Adam as predicted for width-independent hidden layer feature learning for both CIFAR-10 and MNIST.

**Shallow ReLU MLPs at moderate width and (deep) linear networks are not useful proxy models for deep nonlinear networks.** For shallow MLPs, we often observe stronger finite-width effects than for deeper networks causing larger than predicted optimal learning rate scaling at moderate width, as divergence in fewer hidden layers can be stabilized over the course of training up to larger widths (cf. Appendix E). In linear networks, on the other hand, feature learning is not essential as the learned function always remains linear. Consequently we often observe that optimal learning rates decay faster than maximal stable learning rates in (deep) linear networks even under CE loss (Figures F.20 and F.40). These differences between deep non-linear networks and toy architectures suggest that shallow MLPs and deep linear networks do not serve as useful proxy models for practical non-linear networks in terms of optimal learning rate exponents at moderate width.

**Input layer task.** Under multi-index data with a sparse signal and high-dimensional isotropic covariates (explained in Appendix D.2), learning the two signal input dimensions is particularly useful for good generalization. Appendix F.4 shows the predicted exponent $\eta_n = \eta \cdot n^0$ for input layer learning in 2-layer MLPs. Deeper MLPs recover hidden layer stability with optimal learning rate scaling $\eta_n = \Theta(n^{-1/2})$. Observe that generalization suffers when the input layer does not learn to align with the signal dimensions, so that only the 2-layer MLP with CE loss generalizes well at large width.

**Standard initialization with $\mu$P learning rates (SP-full-align).** While Everett et al. (2024) report good transfer properties of SP-full-align, Appendix F.7 shows that the optimal learning rate clearly shrinks across image datasets and our multi-index data. We also introduce a variant of this parameterization that matches the $n^{1/2}$ logit blowup rate from the term $W_0^{L+1}\Delta x_t$ in the effective last-layer updates by increasing the last-layer learning rate. This variant performs similarly well as SP-full-align. In particular, both variants seem to be less learning rate sensitive than $\mu$P.

**Adam learns features with $\eta_n = \eta \cdot n^{-1}$.** Adam simplifies the learning rate scaling for weight $W$ to $\eta_W = \eta/\texttt{fan\_in}(W)$, because the gradient is normalized but still correlated with the incoming activations since the sign is preserved in each entry. Thus $\eta_n = \eta/n$ is expected to induce width-independent hidden- and output-layer learning, but vanishing input-layer learning since here $\texttt{fan\_in}$ is fixed and hence would require constant learning rate scaling. As for SGD, we still observe the optimal learning rate scaling $\eta_n = \eta \cdot n^{-1}$ in deep MLPs on MNIST and CIFAR-10 (Appendix F.6), indicating that width-independence in hidden- and output-layer dominates input layer feature learning.

## F.2 Transformer experiments

As we consider single-pass training, training and validation loss approximately coincide, so that statements about the training loss transfer to statements about the validation loss irrespective of the optimizer. All figures in this section show training loss on the left and validation loss on the right.

Stabilizing techniques like gradient clipping can improve the absolute learning rate multiplier in front of the scaling law, but do not seem to change the width-scaling exponent for SGD (Figure F.4 vs Figure F.5).

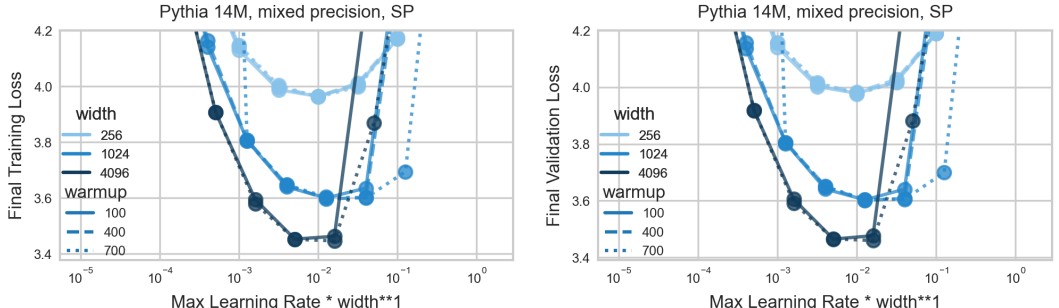

Figure F.1: **(Instability without qk-Layernorm)** Train loss (left) and validation loss (right) of single-pass AdamW training without qk-Layernorm. Training and validation loss approximately coincide. Optimal learning rate scaling is dominated by the maximal stable learning rate scaling that is at most $\Theta(n^{-1})$. But without qk-Layernorm, the stability threshold is decreasing faster than $\Theta(n^{-1})$ even when increasing warmup length, so that it may be that the instability threshold would decay beyond the ideal learning rate and performance suffers. As our computational budget does not allow us to scale further, this setting remains inconclusive.

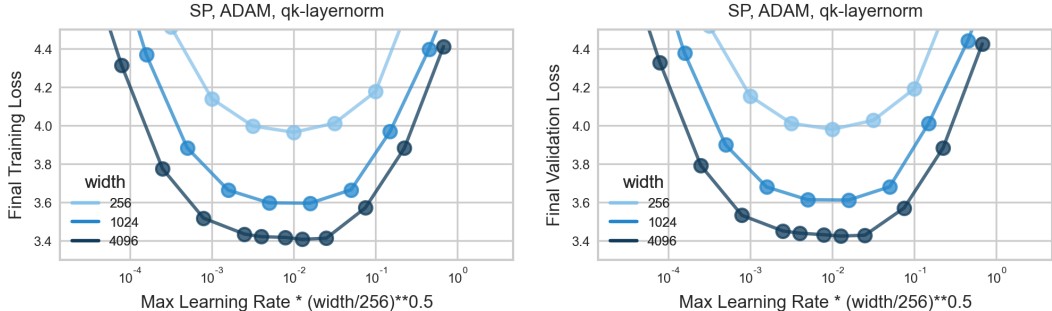

Figure F.2: **(Large learning rate stability with qk-Layernorm)** Same as Figure F.1 but with qk-Layernorm as recommended by Wortsman et al. (2024). Training and validation loss approximately coincide. The optimal learning rate seems to approximately transfer under $\eta_n = \eta \cdot n^{-1/2}$, so the added Layernorm appears to stabilize learning at larger learning rate scaling, similar to the softmax in CE loss.

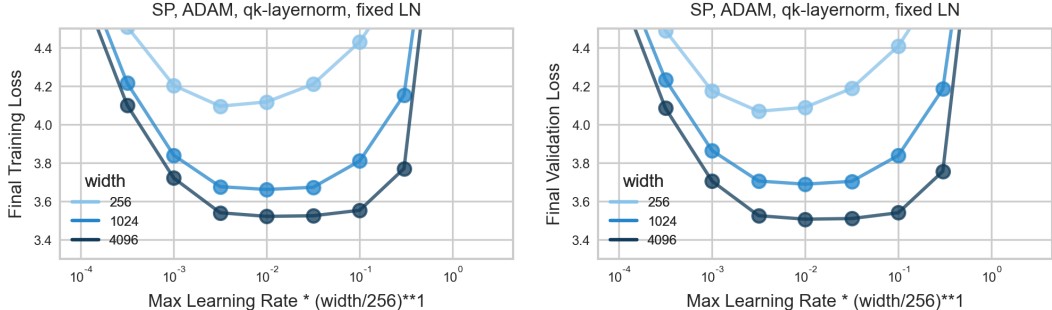

Figure F.3: **(Smaller optimal learning rate exponents under fixed Layernorms)** Same as [Figure F.1](#) with qk-Layernorm as recommended by [Wortsman et al. (2024)](#), but all trainable Layernorm parameters are fixed to initialization. Here only the embedding layer behaves input-like, so that all other parameters learn width-independently under learning rate scaling $\Theta(n^{-1})$. While the optimum is drifting toward larger learning rates, an increasingly large plateau of near-optimal learning rates emerges at large width. $\Theta(n^{-1})$ still approximately captures the maximal stable learning rate scaling.

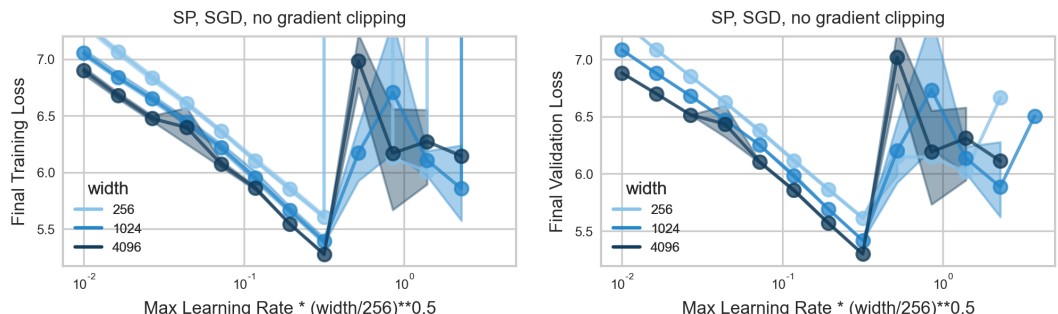

Figure F.4: **(GPT trained with SGD has $\Theta(n^{-1/2})$-learning rate scaling)** Train loss (left) and validation loss (right) of single-pass SGD training (averaged over 3 random seeds affecting weight initialization and data shuffling). Training and validation loss approximately coincide. Hence also validation-optimal learning rate scaling is dominated by maximal stable learning rate scaling $\Theta(n^{-1/2})$ for hidden-layer stability.

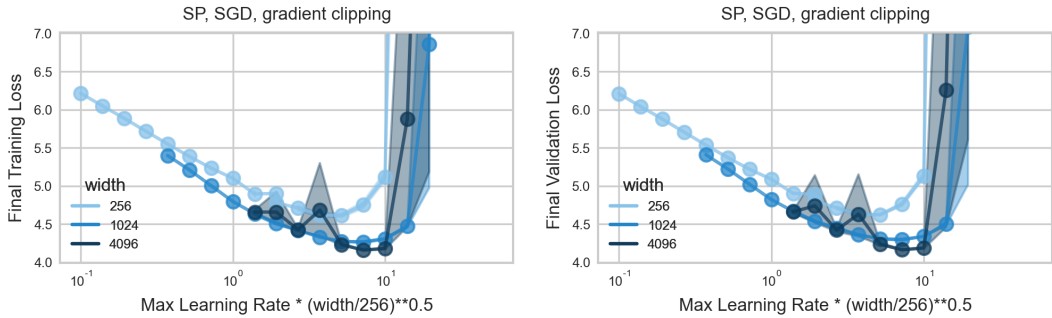

Figure F.5: **(Larger learning rates remain stable under gradient clipping)** Same as [Figure F.4](#) but with gradient clipping. Performance is significantly improved as larger learning rate constants are stable (observe similar performance as without gradient clipping at the same learning rate). Optimal learning rate scaling is still dominated by the maximal stable learning rate scaling $\eta_n = \eta \cdot n^{-1/2}$ for hidden-layer stability.

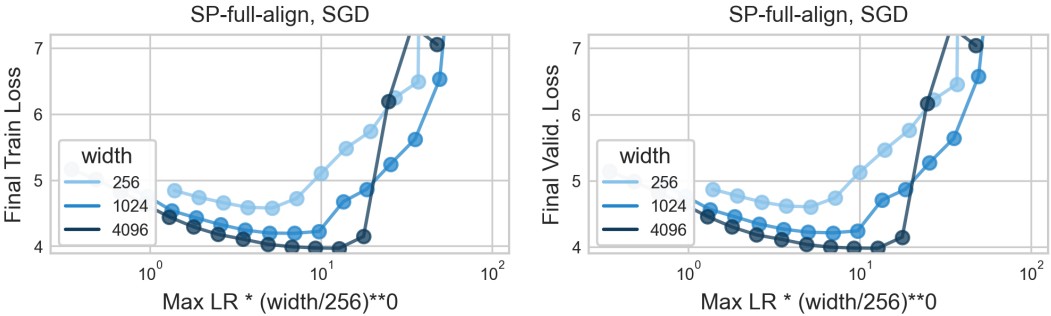

Figure F.6: (**No clean transfer under SGD in SP-full-align**) Same as Figure F.5 but in SP-full-align. The optimal learning rate tends to grow and saturate at the maximal stable learning rate which appears to be roughly width-independent as predicted, but transfer is not ideal.

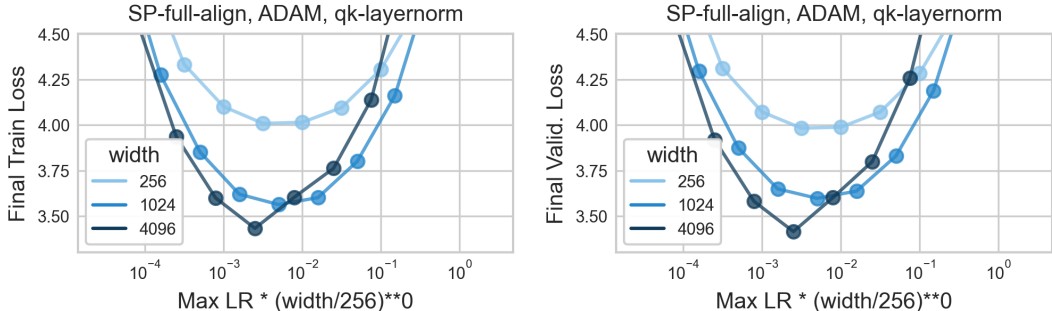

Figure F.7: (**Optimal learning rate shrinks at large width under AdamW in SP-full-align**) Same as Figure F.2 (AdamW and trainable qk-Layernorm) but in SP-full-align. The optimal learning rate initially transfers but starts to shrink at sufficient width. In Appendix F.7, the optimal learning rate already decays at small width on image datasets with small output dimension.

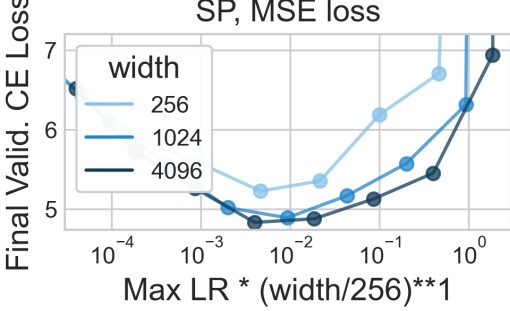

Figure F.8: (**Optimal learning rate exponent** $-1$ **as predicted under MSE loss**) Same as Figure F.5 (SGD in SP with gradient clipping) but under MSE loss. The optimal learning rate appears to follow $\eta_n = \eta \cdot n^{-1}$, as expected. Loss is worse than under CE loss.

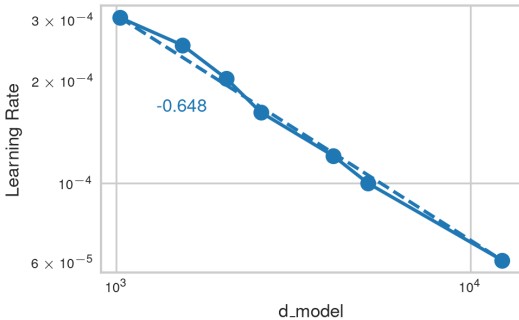

Figure F.9: **(Large learning rate exponent in original GPT paper)** Just plotting the reported learning rate and d_model values from Brown et al. (2020) results in quite a stable scaling law with exponent $-0.648$, which is larger than $-1$ required for hidden-layer stability but significantly smaller than $0$ required for width-independent embedding and normalization layer learning. But note that jointly increasing batch size, n_layers and n_heads might be confounding factors here.

## F.3  Learning rate sweeps corresponding to 8-layer MLPs

Here we provide all learning rate curves that correspond to the learning rate exponent tables of 8-layer MLPs trained on image datasets (Figure 6). The curves corresponding to GPT on DCLM-Baseline are provided in Appendix F.2. For SP under CE loss and MSE loss, Figures F.10 and F.11 show that maximal stable learning rates approximately follow the predicted exponent $-0.5$ for CE loss and $-1$ for MSE loss, and even for the worst-fitting dataset CIFAR-10 the optimal learning rate saturates at the maximal stable learning rate at realistic width $16384$ (Figure F.13). Thus, for CIFAR-10, we expect a regime transition when scaling further where the maximal stable learning rate constrains the optimal learning rate to scale with exponent close to $-0.5$.

In all datasets, performance improves much more with scale under CE loss than under MSE loss, under which feature learning is lost and the accuracy tends to asymptote at moderate scale. For GPT on the DCLM-Baseline dataset with MSE loss, gradient clipping and normalization layers might stabilize larger learning rates, but optimal learning clearly requires exponent $-1$. For SP-full-align under CE loss, Figure F.12 shows that the maximal stable learning rate remains remarkably width-independent (as predicted) on all four image datasets, but that the optimal learning rate decays with differing exponents.

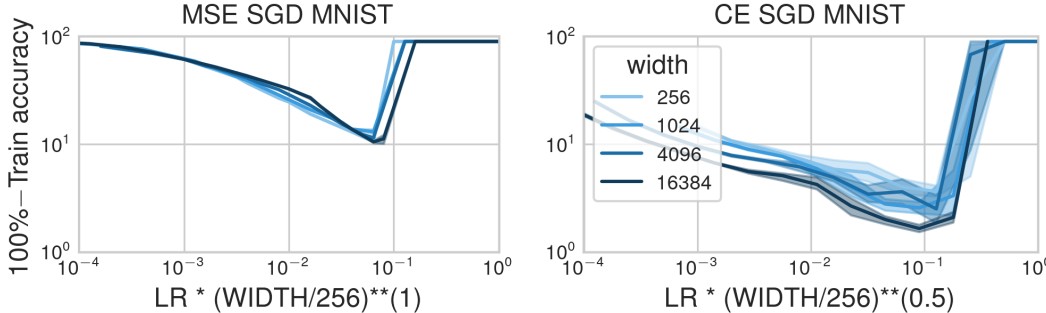

Figure F.10: **(Learning rates decay slower under CE loss than under MSE loss)** Width-scaled learning rate versus training error for MNIST showing approximate transfer with $\eta_n = \eta \cdot n^{-1}$ under MSE loss versus with $\eta_n = \eta \cdot n^{-1/2}$ under CE loss.

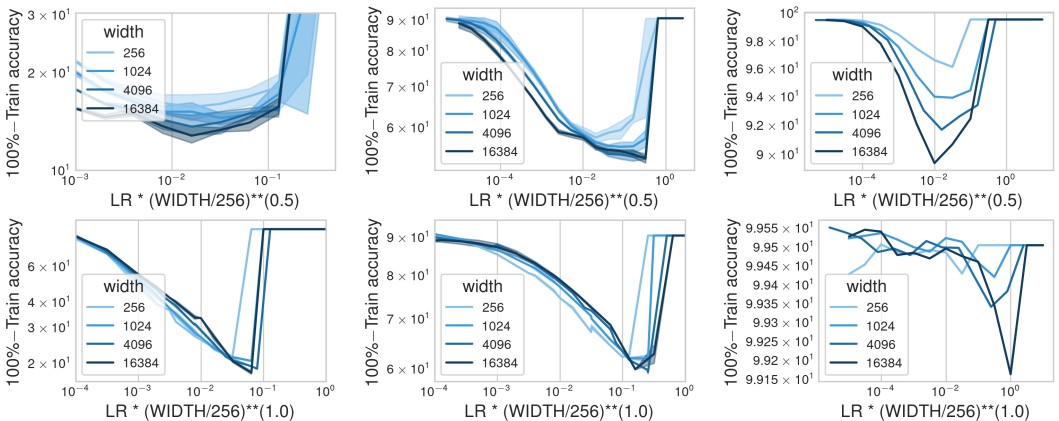

Figure F.11: (**Maximal stable learning rate dominates optimal learning rate at sufficient width**) Same as Figure F.10 but for FashionMNIST (left), CIFAR-10 (center) and TinyImagenet (right): Training accuracy as a function of width-scaled learning rate of 8-layer ReLU MLPs trained with SGD in SP under CE loss (top) and MSE loss (bottom). The maximal stable learning rate approximately follows the predicted exponent $-0.5$ for CE loss and $-1$ for MSE loss, and the optimal learning rate saturates close to the maximal stable learning rate at sufficient width. MLPs barely learn on TinyImagenet when using MSE loss.

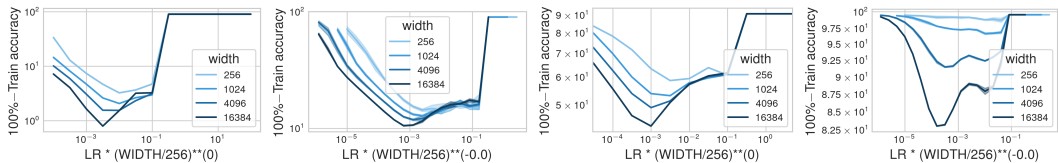

Figure F.12: (**Optimal learning rate decays in SP-full-align**) Training accuracy as a function of learning rate of 8-layer ReLU MLPs trained with SGD in SP-full-align under CE loss on MNIST, FashionMNIST, CIFAR-10 and TinyImagenet (from left to right). As predicted, the maximal stable learning rate remains strikingly width-independent in all datasets, but the optimal learning rate decays in all datasets.

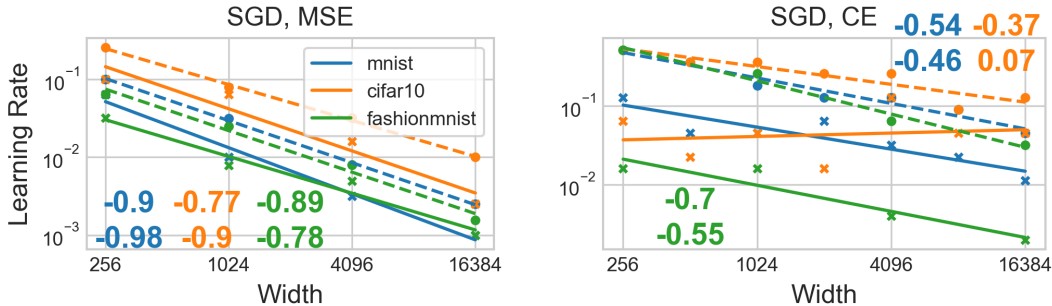

Figure F.13: **Learning rates decay slower under CE loss than under MSE loss.** Optimal learning rate (solid) and minimal unstable learning rate (dashed) for 8-layer MLPs on CIFAR-10, MNIST and Fashion-MNIST corresponding to Figure F.11. Optimal learning rates are often close to max-stable learning rates. Theoretical instability predictions $\eta_n = \mathcal{O}(n^{-1})$ for MSE loss and $\eta_n = \mathcal{O}(n^{-1/2})$ for CE loss are surprisingly accurate. Even for CIFAR-10, the optimal learning rate saturates at the maximal stable learning rate at realistic width $16384$, so that it is expected to follow the maximal stable learning rate scaling at larger widths.

|  |  | MNIST | Fashion-MNIST | CIFAR-10 | Tiny ImageNet | DLCM-Baseline |
|---|---|---|---|---|---|---|
| CE loss | Max-stable LR expon. | -0.54 | -0.7 | -0.37 | -0.33 | -0.38 |
|  | Optimal LR expon. | -0.46 | -0.55 | 0.07 | -0.33 | -0.38 |
| MSE | Max-stable LR expon. | -0.9 | -0.89 | -0.77 | - | -0.50 |
|  | Optimal LR expon. | -0.98 | -0.83 | -0.9 | - | -1.05 |
| SP-full-align | Max-stable LR expon. | 0.0 | 0.0 | 0.0 | 0.0 | 0.09 |
|  | Optimal LR expon. | -0.25 | -0.35 | -0.33 | -1.45 | 0.21 |

Table F.1: **Learning rate exponents at the edge of controlled divergence.** Same data as in Figure 6, but as a table. We do not compute exponents on TinyImageNet under MSE loss because the accuracy of MLPs remains too close to random guessing.

## F.4 MSE loss with softmax does not enable feature learning, but Adam does

**CE versus MSE versus MSE+softmax.** Here we train 2-layer and 3-layer ReLU MLPs on generated multi-index teacher data as detailed in Appendix D. These data crucially differ from the other considered datasets in that the target function only depends on the first 2 input dimensions. Due to the isotropic covariate distribution, input layer feature learning is necessary for good generalization. Hence we observe an approximate $\eta_n \approx \Theta(1)$ scaling for 2-layer MLPs with CE loss, necessary for preserving input layer feature learning (Figure F.14, left). 3-layer MLPs attain the maximal activation-stable exponent $\eta_n = \Theta(n^{-1/2})$ in CE loss (Figure F.16, left). 2-layer MLPs preserve a better validation accuracy compared to their training accuracy than deeper nets, as input layer learning gets increasingly inhibited by $\Theta(1)$-learning rate instability in the presence of hidden layers. Both for shallow and deeper MLPs with MSE loss, we lose feature learning under the maximal output-stable scaling $\eta_n = \Theta(n^{-1})$, as expected.

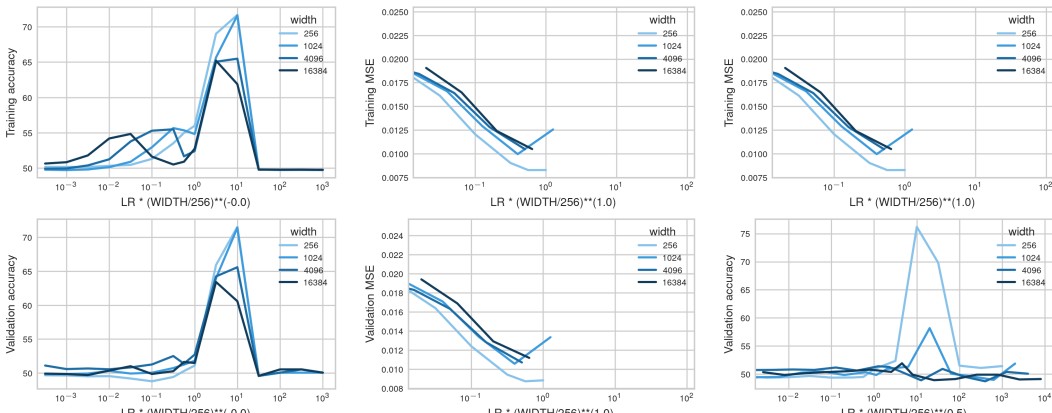

Figure F.14: **(CE loss increases maximal stable learning rate scaling to $\Theta(1)$ in 2-layer nets)** Training accuracy (top) and validation accuracy (bottom) for a 2-layer MLP on generated multi-index teacher data (mean over 4 seeds) with CE loss (left), MSE loss (center) and MSE loss with softmax (right). The x-axis scales the learning rate with width-dependent exponents; observe approximate transfer under $\Theta(1)$, $\Theta(n^{-1})$ and $\Theta(n^{-1})$ scaling, respectively. In the MSE plot, ending lines indicate divergence for larger learning rates. MSE loss with softmax on the output does not increase optimal learning rate scaling due to vanishing gradients and gets worse due to a lack of input layer feature learning.

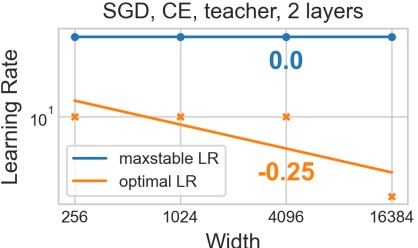 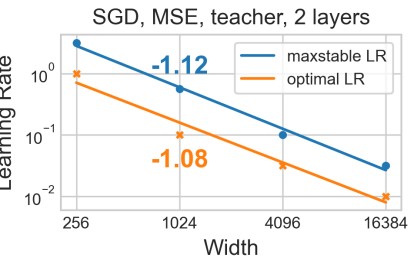

Figure F.15: **(CE loss increases maximal stable learning rate scaling to $\Theta(1)$ in 2-layer nets)** Optimal and maximal stable learning rate exponents corresponding to Figure F.14. Since 2-layer MLPs do not contain hidden layers, the maximal stable learning rate under CE loss is determined by input-layer stability $\eta_n = \Theta(1)$ (see Remark C.14). Under MSE loss, the maximal stable learning rate is constrained by logit stability $\eta_n \approx \Theta(n^{-1})$.

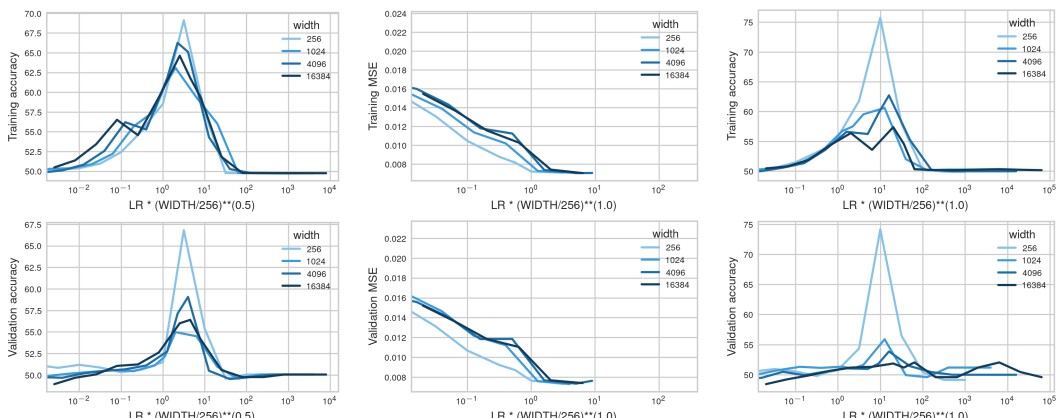

Figure F.16: **(Cross-entropy loss increases maximal stable learning rate scaling to approximately** $\Theta(n^{-1/2})$ **in 3-layer nets)** Same as Figure F.14 but for a 3-layer MLP. The x-axis scales the learning rate with width-dependent exponents; observe approximate transfer of the maximal stable learning rate under $\Theta(n^{-1/2})$, $\Theta(n^{-1})$ and $\Theta(n^{-1})$ scaling, respectively. In the MSE plot, ending lines indicate divergence for larger learning rates. Observe that wider networks generalize worse with scale as they lose input layer feature learning.

In this setting, it becomes particularly apparent that using the MSE loss with a softmax applied to the output of the network is not desirable (Figures F.14 and F.16, right). Ultimately, the only difference to CE loss is that the loss derivative with respect to the network output $f(\xi) := W^{L+1}x^L(\xi)$ becomes

$$\left(\frac{\partial \mathcal{L}}{\partial f}\right)_j = \sum_{i \in [C]} (\sigma(f)_i - y_i)\sigma(f)_i(\delta_{ij} - \sigma(f)_j),$$

where the inner derivative of the softmax $\sigma_i(\delta_{ij} - \sigma_j)$ vanishes as soon as the outputs diverge $|f_i(\xi) - f_j(\xi)| \to \infty$ on a training point $\xi$. Hence, while the softmax still mitigates output blowup in the forward pass, the gradients vanish under output blowup. The CE loss, on the other hand, is exactly the correct choice of loss function to cancel out the inner derivative of the softmax and effectively view $\sigma(f)$ as the output of the network, resulting in $\left(\frac{\partial \mathcal{L}}{\partial f}\right)_j = \sigma(f)_j - y_j$.

Here vanishing gradients under output blowup in the MSE+softmax setting is so severe that output blowup prevents learning under large learning rates and the optimal learning rate scales as $\Theta(n^{-1})$.

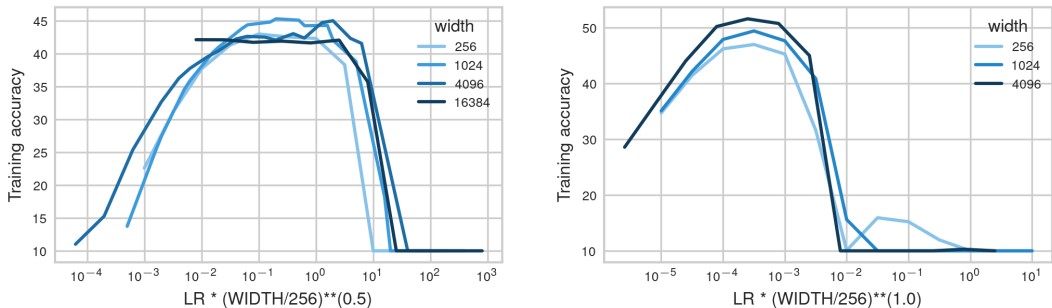

Figure F.17: **(Layernorm on logits also stabilizes large learning rates under MSE loss)** 8-layer MLPs trained with SGD (left) and Adam (right) on CIFAR-10 with MSE loss with Layernorm applied to the logits. The Layernorm has a similar stabilizing effect as CE loss and allows learning with logit blowup under $\eta_n = \Theta(n^{-1/2})$, but performance does not monotonically improve with width. For Adam, hidden and output layers learn width-independently with $\eta_n = \Theta(n^{-1})$.

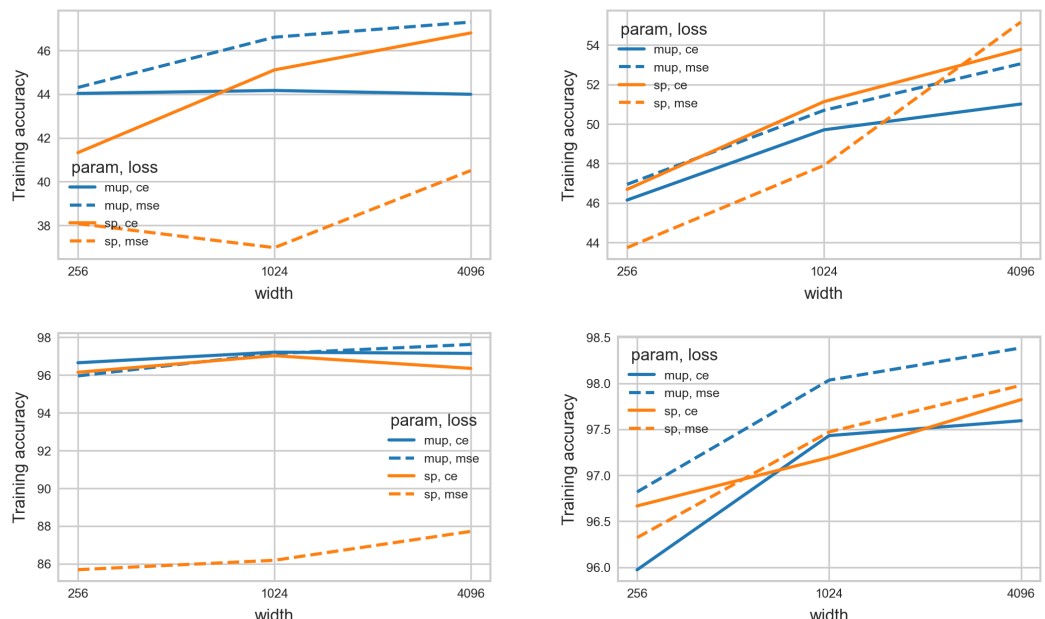

Figure F.18: **(Large performance difference between losses for SGD in SP but not Adam)** Optimal training accuracy of 8-layer MLPs trained with SGD (left) and Adam (right) on CIFAR-10 (top) and MNIST (bottom) with MSE loss (dashed lines) and CE loss (solid lines) in $\mu$P (blue) and SP (orange). For SGD in SP, CE loss performs much better than MSE loss as large learning rates recover feature learning at large widths. The performance in $\mu$P depends much less on the loss function since features are always learned width-independently. In $\mu$P, MSE loss slightly outperforms CE loss, suggesting that $\mu$P might benefit from exploring other loss functions. For ADAM, $\eta_n = \Theta(n^{-1})$ in SP remains stable even under MSE loss and recovers hidden-layer feature learning so that the difference between losses is much smaller.

In Figure F.17, we apply a Layernorm to the logits instead of a softmax. The Layernorm allows learning despite logit blowup under $\eta_n = \Theta(n^{-1/2})$, but performance does not monotonically improve with width either.

**Adam reduces sensitivity to loss function.** Figure F.18 shows that the big performance difference between MSE loss and CE loss for SGD in SP is reduced if either (a) $\mu$P is applied to ensure width-independent feature learning or (b) Adam in SP is used, which remains stable with $\eta_n = \Theta(n^{-1})$ due to stable update scaling even under MSE loss, allowing width-independent hidden-layer feature

learning. Observe that MSE loss outperforms CE loss in both $\mu$P and Adam in SP. This suggests that exploring loss functions beyond CE loss might be promising when using $\mu$P and/or Adam. A more detailed evaluation of Adam is provided in Appendix F.6.

## F.5 SGD

### F.5.1 Random feature models are optimal under stable learning rates

Figure F.19 shows that 2-layer random feature ReLU MLPs transfer under $\eta_n = \eta \cdot n^{-1}$ learning rate scaling under any loss. Under CE loss, also larger learning rates result in non-trivial learning as saturating the softmax does not harm training stability. Under MSE loss on the other hand, training diverges above the edge of stability and results in trivial accuracy of $10\%$. Under MSE with softmax on the output logits, a exploding logits induce vanishing gradients which also inhibits learning (see Appendix F.4 for more details), and results in worse accuracy than under CE loss.

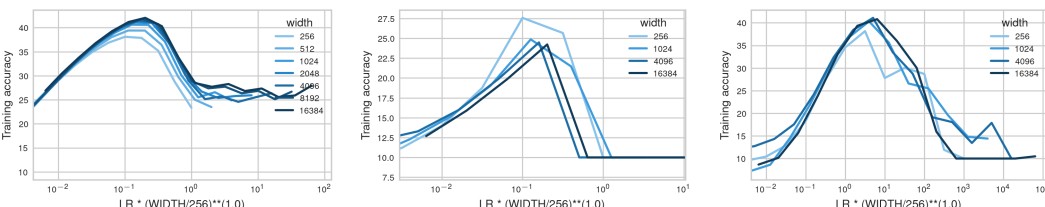

Figure F.19: **(Random feature models approximately transfer under $\eta_n = \Theta(n^{-1})$ for SGD)** Training accuracy after one epoch of only training the last layer of 2-layer MLPs on CIFAR 10 with CE loss (left), MSE loss (center) and MSE loss with softmax (right). Observe approximately width-independent dynamics with $\eta_n = \eta \cdot n^{-1}$ independent of the loss function or architecture used. Note that also larger learning rates result in non-trivial generalization because there is no instability caused by activation blowup. The larger learning rates are not optimal, because the usual benefits of larger learning rates like increased feature learning or activation sparsity do not apply to random feature models.

### F.5.2 Linear networks are optimal under stable learning rates

Linear networks lack the ability to learn non-linear features for improved generalization at large learning rates. Hence, small learning rates $\eta_n \approx \Theta(n^{-1})$ are optimal for MNIST (Figure F.20), where feature learning is lost, but activations and logits remain stable. Observe that also deeper linear MLPs are only stable under $\mathcal{O}(n^{-1/2})$ as theoretically predicted.

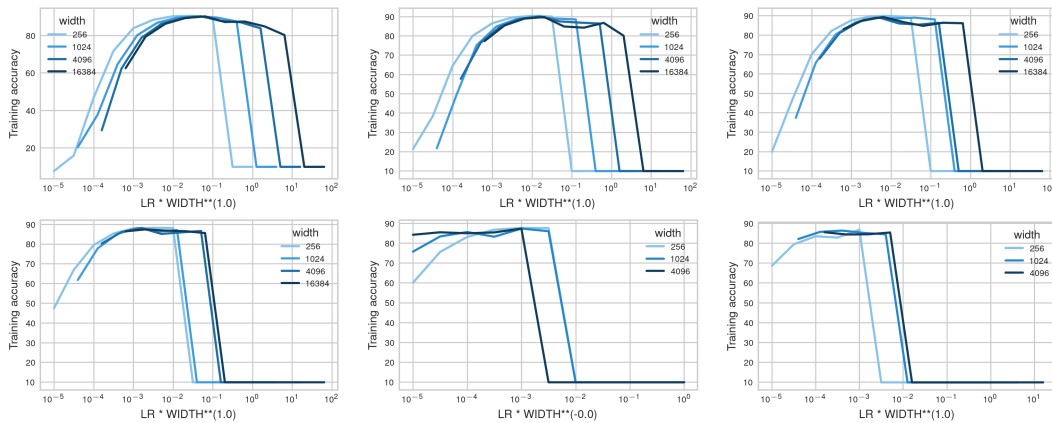

Figure F.20: **(In linear nets on MNIST, the optimal learning rate shrinks faster than the maximal stable learning rate)** Same as Figure F.22 but for linear nets. The maximal stable learning rate scales similarly as for the non-linear nets, but the optimum approximately follows $\Theta(n^{-1})$. Irrespective of the depth, linear MLPs can only learn a linear transformation; hence under sufficient width, feature learning under large learning rates does not provide a benefit over mere last-layer learning.

### F.5.3 Small optimal learning rates under MSE loss

With MSE loss, observe a clear $\eta_n = \mathcal{O}(n^{-1})$ optimal and maximal stable learning rate exponent irrespective of MLP depth (Figure F.21). Any blowup induces catastrophically cascading updates, so that the stability threshold $\eta_n = \mathcal{O}(n^{-1})$ is strictly enforced at all depths, and the loss converges to its kernel limit at moderate width for MLPs with at least 3 layers.

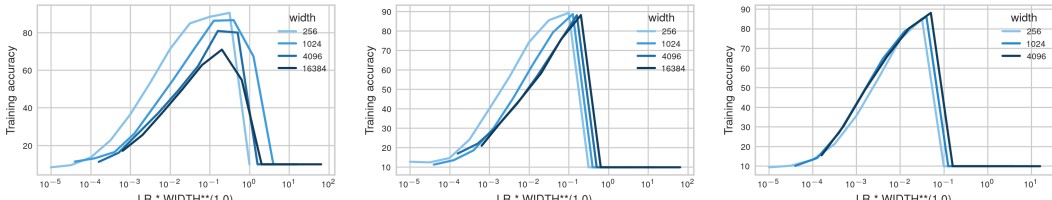

Figure F.21: **(MSE loss on MNIST approximately transfers under $\Theta(n^{-1})$ learning rate scaling)** Both the optimal as well as the maximal stable learning rate approximately transfer under global $\Theta(n^{-1})$ learning rate scaling when training 2, 3 or 10 layer MLPs (from left to right) with MSE loss on MNIST. Loss is not improving as feature learning is lost under $\Theta(n^{-1})$ scaling. Especially in 2 layer nets, the input layer is learning features at small width, but not at large width, so that the loss worsens with width. In 3 and 10 layer MLPs, on the other hand, the loss quickly converges with width, which suggests fast convergence to the kernel limit under MSE loss.

### F.5.4 Large optimal learning rates under CE loss

As theoretically predicted in Remark C.14, 2-layer MLPs transfer the optimal and maximal stable learning rate $\eta_n \approx \Theta(n^0)$, where the input layer learns width-independently.

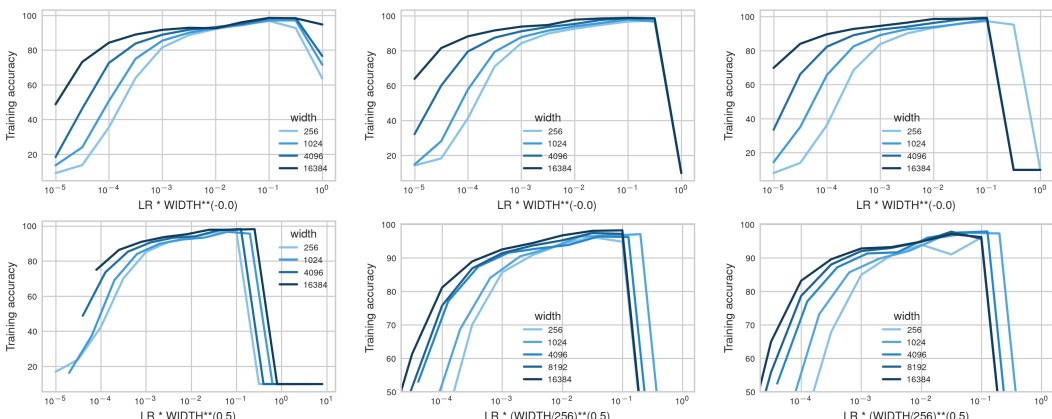

Figure F.22: **(Deeper nets follow infinite width theory increasingly accurately)** Training accuracy after 1 epoch of training MLPs with 2, 3, 4, 6, 8 and 10 layers (from top-left to bottom right) on MNIST. While 2, 3 and 4 layer MLPs self-stabilize under large learning rates $\Theta(n^0)$ and approximately transfer the optimum as well as max-stable learning rate, in 6, 8 and 10 layer MLPs it becomes increasingly apparent that the maximal stable learning rate transitions towards $\Theta(n^{-1/2})$ to prevent hidden layer blowup, which also forces the optimal learning rate to be $\mathcal{O}(n^{1/2})$ for at least feature learning in the hidden layers. Hence the theoretical activation stability predictions hold more accurately in deeper nets, with too many hidden layers to stabilize blowup in all of them.

For 3-layer RELU MLPs with CE loss on CIFAR-10, the maximal stable learning rate and the optimal learning rate transfer over many widths before beginning to shrink at width $16384$. At moderate width and depth, strong self-stabilization mechanisms such as activation sparsification stabilize an initial catapult at large learning rate (Figure E.6).

In increasingly deep networks, the maximal stable learning rate scaling $\eta_n \approx \mathcal{O}(n^{-1/2})$ becomes increasingly pronounced, as it becomes increasingly difficult to stabilize activation blowup in an

increasing amount of hidden layers, and width-independent learning of increasingly many hidden layers dominates (MNIST Figure F.22, CIFAR-10 Figure F.23).

Observe in all figures here that the optimal learning rate saturates at the maximal stable learning rate at sufficient width.

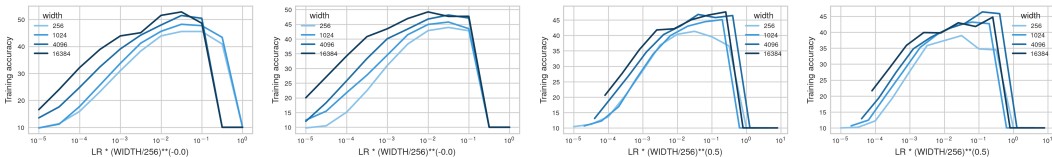

Figure F.23: **(Hidden-layer stability determines learning rate scaling in deep MLPs for SGD on CIFAR-10)** MLPs trained with SGD on CIFAR-10 with 4, 6, 8 and 10 layers (from left to right). First two x-axes are width-independent, last two scaled by $n^{1/2}$. While 4- and 6-layer MLPs self-stabilize sufficiently for approximate transfer under width-independent learning rate scaling, 8- and 10-layer MLPs have a max-stable learning rate scaling $\eta_n \approx \mathcal{O}(n^{-1/2})$. The optimal learning rate saturates at the maximal stable learning rate at sufficient width.

## F.6 Adam stabilizes updates similar to CE loss

We first show $\mu$P as a baseline for learning rate transfer in MLPs on MNIST (Figure F.24).

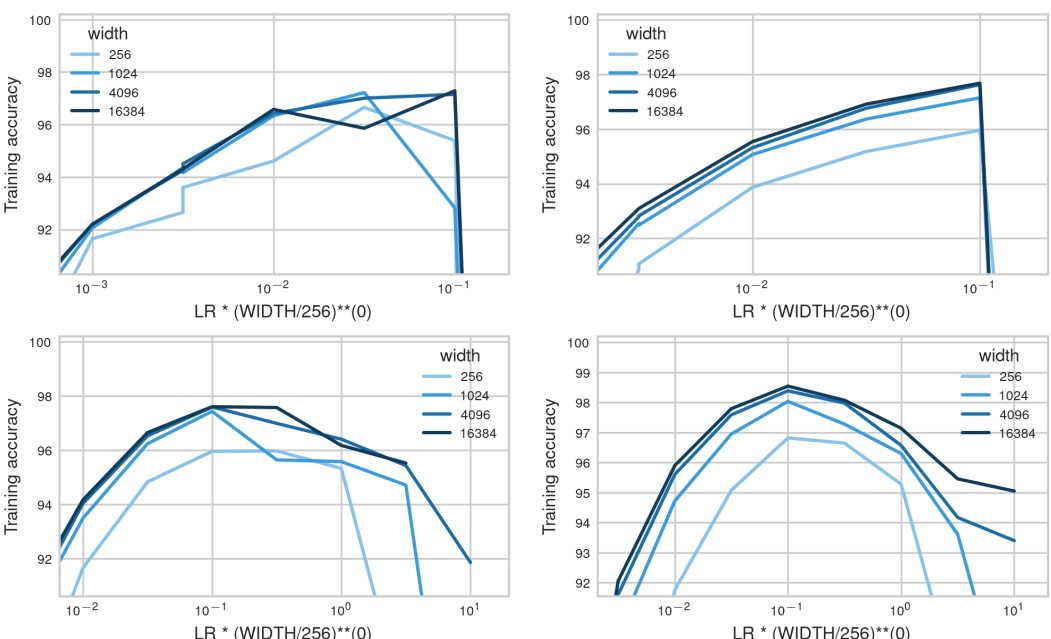

Figure F.24: **($\mu$P as a baseline for transfer)** 8-layer MLPs trained on MNIST with SGD (top) and ADAM (bottom) under CE loss (left) and MSE loss (right). No systematic learning rate shifts in $\mu$P; saturating drifts may occur. Transfer and monotonic improvement looks less noisy under MSE loss.

**Controlled divergence through stabilized updates.** The closest clean optimal learning rate exponent of 8-layer MLPs trained with Adam under both MSE loss as well as CE loss is $-1$ for most of the evaluated image datasets (Figure F.25). Validation-optimal learning rates tend to be larger than train-optimal learning rates, suggesting a well-generalizing bias of large learning rates. The fact that Adam with MSE loss (Figure F.26) can have optimal learning rates as large as CE loss indicates that the crucial effect of CE loss in SGD is stabilizing the updates. A crucial difference to SGD is that activation blowup does not affect the updates in Adam since the gradient is normalized. For SGD, exploding gradients induce even larger explosion in the next forward pass, which in turn induces even larger explosion in the next backward pass. Hence, without activation stability, even the divergence

exponent grows over time in SGD resulting in catastrophically cascading updates. For Adam, on the other hand, gradients are normalized, so that the forward pass always accumulates the same width-dependent exponent that is stabilized when passed through the softmax. Thus under sufficient numerical precision, from a stability point of view, Adam can even tolerate larger learning rates than the hidden-layer feature learning $\eta_n = \Theta(n^{-1})$, and the optimal learning rate may also be pushed toward input layer feature learning.

**In shallow networks the input layer enforces slower optimal learning rate decay.** As for SGD, with increasing depth, the optimal learning rate decays faster, from around $\eta_n \approx \Theta(n^{-1/2})$ in shallow networks to small $\eta_n = \mathcal{O}(n^{-1})$ in deep MLPs in both MNIST (Figure F.27) and CIFAR-10 (Figure F.28), and in both train and validation accuracy (Figure F.29). Figure F.30 confirms the hypothesis that the optimal learning rate is subject to opposing objectives for width-independent learning of input-like and hidden-like layers by showing that a 3-layer MLP trained with Adam with fixed input layer follows the clean exponent $\eta_n \approx \Theta(n^{-1})$, where the hidden and output layer learn fully width-independently.

**Less clear maximal stable learning rate.** For Adam, the optimal learning rate typically does not saturate at the maximal stable learning rate. Instead, a regime of suboptimal large learning rates emerges where its moments are already harmed (Kalra and Barkeshli, 2024), and the maximal stable learning rate threshold is often less clear cut compared to SGD. For all depths, the instability threshold appears to scale around $\eta_n \approx \Theta(n^{-1/2})$.

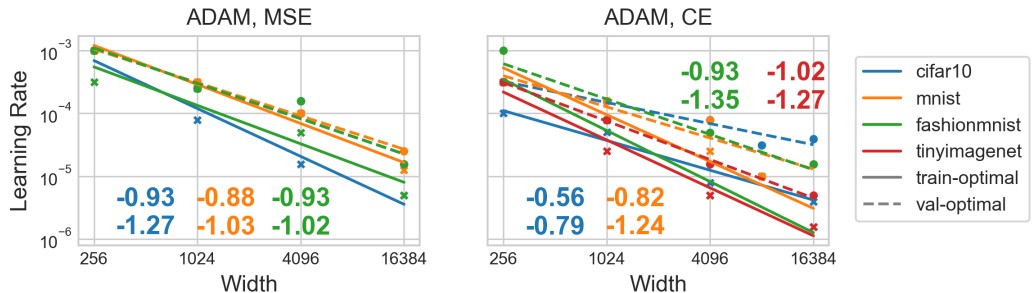

Figure F.25: **(MSE and CE loss share similar optimal learning rate exponents under Adam)** Train-optimal (solid) and validation-optimal (dashed) learning rate as a function of width for several image datasets for 8-layer MLPs trained with Adam under MSE loss (left) and CE loss (right). Generally observe exponents around $-1$. The MSE-optimal learning rate does not decay faster than under CE loss, which indicates that Adam's parameterwise normalization of the gradient stabilizes exploding updates, recovering feature learning with $\eta_n \approx \Theta(n^{-1})$ under both MSE and CE loss.

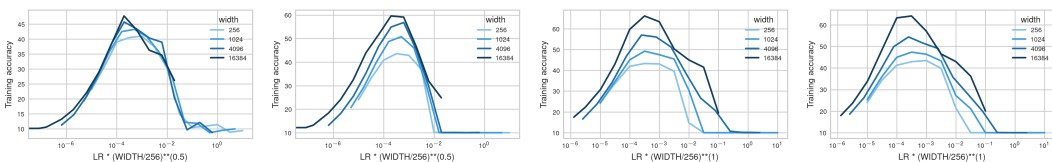

Figure F.26: **(Adam stabilizes the backward pass even under MSE loss)** MLPs trained with ADAM on CIFAR-10 under MSE loss with 2, 3, 6, 8 layers (from left to right). 2 and 3 layers show approximate transfer under $n^{-1/2}$ learning rate scaling, 6 and 8 layers show approximate transfer under $n^{-1}$.

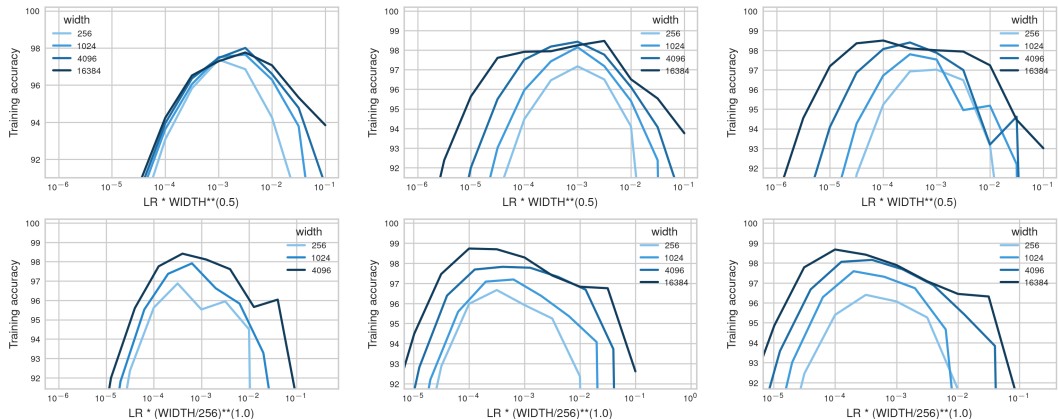

Figure F.27: **(Learning rate transfer in deep MLPs for ADAM on MNIST)** MLPs trained with ADAM on MNIST with 2, 3, 4, 6, 8, 10 layers (from top left to bottom right). In the first row, the x-axis is width-dependently scaled to show approximate transfer under $n^{-1/2}$ learning rate scaling. In the bottom row, the x-axis is width-dependently scaled to show approximate transfer under $\eta_n \approx \Theta(n^{-1})$. Observe the optimal learning rate scaling transitioning from larger than $\Theta(n^{-1/2})$ in 2-layer MLPs toward at most $\Theta(n^{-1})$ with increasing depth.

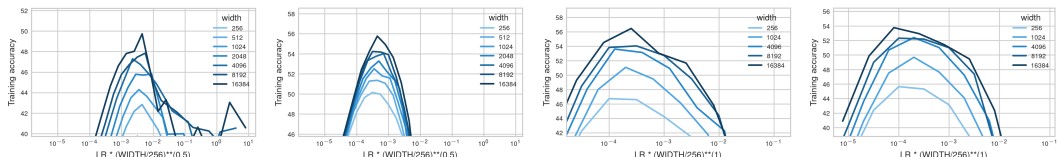

Figure F.28: **(Learning rate exponent $\eta_n = \Theta(n^{-1})$ for ADAM in deep MLPs on CIFAR-10)** MLPs trained with ADAM on CIFAR-10 with 2-layer random features, 2, 8, 10 layers (from left to right). The first 2 x-axes show approximate transfer under $n^{-1/2}$ learning rate scaling, the last 2 under $n^{-1}$ learning rate scaling. As for SGD, in deeper nets hidden-layer width-independence dominates input-layer width-independence and induces optimal learning rate scaling $\eta_n \approx \Theta(n^{-1})$.

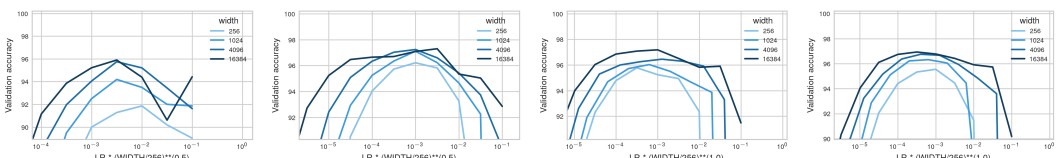

Figure F.29: **(Transfer in validation accuracy in deep MLPs for ADAM on MNIST)** Validation accuracy of MLPs trained with ADAM on MNIST with 2 layer random feature, 3, 8, 10 layers (from left to right). Validation-optimal learning rate in deep MLPs scales as $\eta_n = \mathcal{O}(n^{-1})$. 2 layer RF and 3 layer nets appear to approximately transfer under $\eta_n \approx \Theta(n^{-1/2})$ but lose monotonic improvement and predictability at scale.

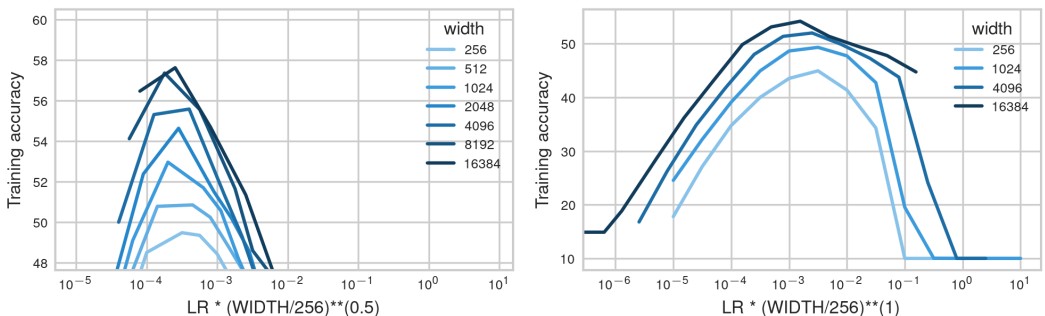

Figure F.30: **(Trade off between input- and hidden-layer width-independence)** 3-layer MLPs trained with ADAM on CIFAR-10 (left) and not training the first layer (right). 3-layer MLPs approximately transfer under $\eta_n = \Theta(n^{-1/2})$, being pushed toward input-layer feature learning. As there are no conflicting goals like preserving input layer feature learning, 3-layer MLPs with fixed input layer follow the width-independent exponent $\eta_n = \Theta(n^{-1})$ that yields hidden-and output-layer width-independent feature learning.

### F.7 Effective update parameterizations beyond $\mu$P

The logit updates can be decomposed into

$$f_t(\xi) - f_0(\xi) = W_0^{L+1}\Delta x_t^L(\xi) + \Delta W_t^{L+1}x_t^L(\xi),$$

for arbitrary inputs $\xi \in \mathbb{R}^{d_{in}}$ and $\Delta W_t^{L+1} = \sum_{t'=0}^{t-1} \chi_{t'} \cdot x_{t'}^L(\xi')$.

In this section, we consider vision and generated data sets in the regime $n \gg d_{\text{out}}$. First note that under large last-layer initialization $(W_0^{L+1})_{ij} \sim N(0, n^{-1})$ as in SP, fully width-independent training dynamics are impossible, since width-independent feature learning $\Delta x_t^L = \Theta(1)$ implies logit blowup through the term $W_0^{L+1}x_t^L = \Theta(n^{1/2})$ for both SGD and Adam. The fact that logit blowup does not prevent stable training under CE loss explains why we can achieve non-vanishing feature learning under SP last-layer initialization. When dropping the logit stability constraint, we can ask which is the optimal layerwise learning rate scaling under standard last-layer initialization. Following the $\mu$P desiderata, we still want to effectively update all layers, meaning a non-vanishing effect of the weight updates in each layer on the output function. With the correct choice of layerwise learning rates, we can still satisfy these desiderata for all scalings of last-layer initialization variance, which also implies that there is not a unique $abc$-equivalence class to fulfill these effective update desiderata when not requiring logit stability. We will see that SP full-align in Everett et al. (2024), which just uses the $\mu$P layerwise learning rates for SP initialization (which they promote as their overall best-performing parameterization without identifying stability under logit blowup as the key mechanism), fulfills these desiderata, except for vanishing last-layer update effect on the output function. We will introduce another variant with larger last-layer learning rate that recovers effective updates of all layers. For avoiding confusion with SP, meaning using a global learning rate, and with $\mu$P, meaning also achieving width-independence in the logits, we call this last variant *Maximal Update Parameterization under Standard Output-layer Initialization (MUSOLI)*.

For deriving the optimal layerwise learning rate exponents, first consider the scaling of hidden-layer pre-activation updates $\delta h_l$, $l \in [2, L]$, and input-layer pre-activation updates $\delta h_1$ (Yang and Hu, 2021, p. 51),

$$\delta h^l(\xi) = \Theta\Big(W_0^l \delta x_t^{l-1} + \eta_l \chi_{t-1}\frac{\partial f}{\partial h_{t-1}^l}\underbrace{(x_{t-1}^{l-1})^\top x_t^{l-1}(\xi)}_{\Theta(n)}\Big),$$

$$\delta h^1(\xi) = \Theta\Big(\eta_1 \chi_{t-1}\frac{\partial f}{\partial h_{t-1}^1}\underbrace{(\xi_{t-1})^\top \xi}_{\Theta(1)}\Big),$$

where it holds that $\partial f/\partial h^l = \Theta(\partial f/\partial x^L) = W_t^{L+1} = \Theta(W_0^{L+1} - \eta_{L+1}\chi_t x^L)$ (at latest in the second step) (Yang and Hu, 2021, p. 52). Hence the correct $l$-th layer learning rate $\eta_l$ for

achieving a width-independent effect on the next layer's pre-activations needs to cancel out the backpropagated gradient scaling $\partial f/\partial h^l$ and for hidden layers additionally the LLN-like scaling from the inner product between activations. As we still require activation stability $x_T^L = \Theta(1)$, we have $\partial f/\partial x^L = \Theta(n^{-\min(b_{L+1}, c_{L+1})})$. While under standard $\mu$P, it holds that $\partial f/\partial x^L = \Theta(n^{-1})$, the changed gradient scaling must be counteracted by choosing hidden layer learning rate $\eta_l = \Theta(n^{\min(b_{L+1}, c_{L+1})-1})$, $l \in [2, L]$, and input layer learning rate $\eta_1 = \Theta(n^{\min(b_{L+1}, c_{L+1})})$. In words, under larger last-layer initialization or learning rate, the hidden and input layer learning rates should be scaled down by the same amount. Finally, SP-full-align achieves a width-independent effect of the last-layer weight updates on the logits. But as the width-independent feature updates $\Delta x^L = \Theta(1)$ induce logit blowup $W_0^{L+1}\Delta x_t^L = \Theta(n^{1/2})$, the effect of the last-layer weight updates on the softmax output is actually vanishing. For last-layer weight updates to affect the softmax output in the same scaling as the updates propagated forward, the last-layer learning rate needs to be $\eta_{L+1} = \Theta(n^{-b_{L+1}})$, hence $c_{L+1} = b_{L+1}$. Hence MUSOLI is defined as SP-full-align but setting $\eta_{L+1} = \Theta(n^{-b_{L+1}})$. This last-layer learning rate is larger than in $\mu$P or Everett et al. (2024) under standard last-layer initialization $b_{L+1} = 1/2$, but necessary for fulfilling the desideratum that the weight updates in all layers affect the output function non-vanishingly.

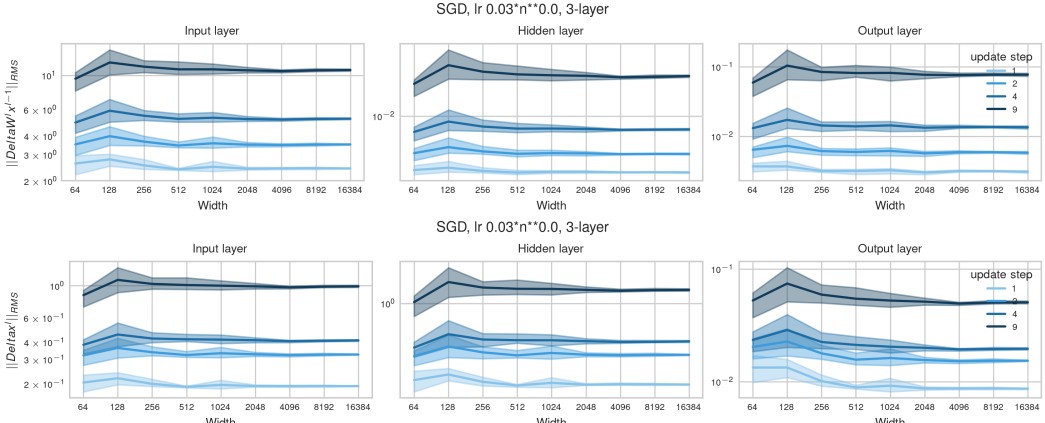

Figure F.31: **(Coordinate check for $\mu$P for SGD on CIFAR-10)** $\mu$P induces fully width-independent update dynamics.

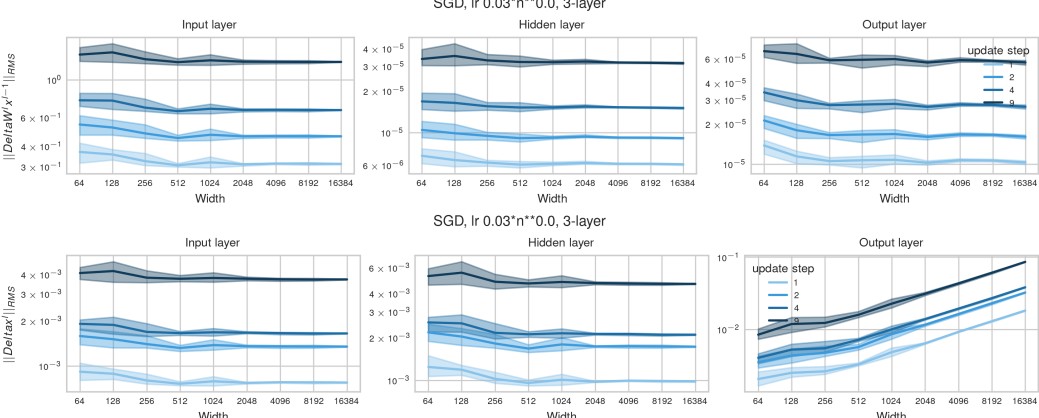

Figure F.32: **(Coordinate check for SP-full-align for SGD on CIFAR-10)** Effective updates $\|\Delta W^l x^{l-1}\|_{RMS}$ and activation updates $\|\Delta x^l\|_{RMS}$ as a function of width. The theoretically predicted scaling exponents hold: All layers update width-independently, but due to the large last-layer initialization, the activation updates correlated with $W_0^{L+1}$ propagated forward induce output logits exploding as $W_0^{L+1}\delta x_t^L = \Theta(n^{1/2})$. This motivates increasing the last-layer learning rate to $\eta_{L+1} = \Theta(n^{-1/2})$ so that last-layer updates contribute with the same scaling. Note that in absolute terms, the updates are much smaller than under $\mu$P (Figure F.31).

By definition, the effective update and propagating update terms in all layers scale width-independently in $\mu$P (Figure F.31). For SP-full-align, Figure F.32 shows that indeed all weight updates behave width-independently, but the output logits are dominated by the activations propagated forward as $W_0^{L+1}\delta x_t^L = \Theta(n^{1/2})$, since $\delta x_t^L$ and $W_0^{L+1}$ are highly correlated. Consequently, the last-layer updates have vanishing effect on the output function, which induces width dependence. By additionally scaling up the last-layer learning rate $\eta_{L+1} = \Theta(n^{-1/2})$, the logit scaling exponent in the term $W_0^{L+1}\delta x_t^L = \Theta(n^{1/2})$ is matched in the last-layer update term $\Delta W_t^{L+1}x_t^L = \Theta(n^{1/2})$ so that $b_{L+1} = 1/2$ and $c_{L+1} = 1/2$ recovers a balanced influence of all layer updates in the softmax output.

Figure F.33 and Figure F.34 show that after single-pass SGD or Adam, for both SP-full-align and MUSOLI the optimal learning rate shrinks with width for both generated 2-class multi-index teacher data as well as MNIST. The optimal learning rate exponent is often closer to $-0.5$ as we consistently observe under MSE loss, preventing logit blowup. Figure F.35 shows the same for CIFAR-10. This behaviour persisting across 3 data sets suggests that neither SP-full-align nor MUSOLI can be expected to transfer the optimal learning rate in general. An interesting question for future work remains why logit divergence introduces a width-dependence in the optimal learning rate in these parameterizations.

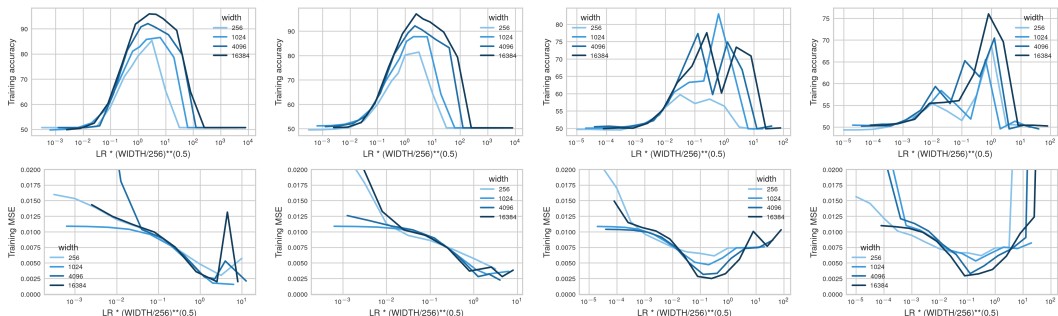

Figure F.33: **(Effective update variants do not transfer optimal learning rates on multi-index data)** Training accuracy of 8-layer MLPs trained for 1 epoch on multi-index teacher data under CE loss (top) and MSE loss (bottom) with SGD in SP-full-align, SGD in MUSOLI, Adam in SP-full-align and Adam in MUSOLI (from left to right). In all cases, logit blowup is avoided by optimal learning rates shrinking as $\eta_n = \Theta(n^{-1/2})$. Under CE loss the maximal stable learning rate remains width-independent, for SGD under MSE loss the maximal stable learning rate decays as $n^{-1/2}$, as necessary for stability.

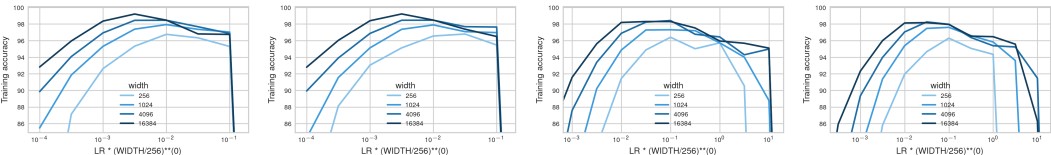

Figure F.34: **(Effective update variants do not transfer optimal learning rates on MNIST)** Training accuracy of 8-layer MLPs trained for 1 epoch on MNIST under CE loss with SGD in SP-full-align, SGD in MUSOLI, Adam in SP-full-align and Adam in MUSOLI (from left to right). In all cases, the optimal learning rate decays with width, while the maximal stable learning rate stays constant.

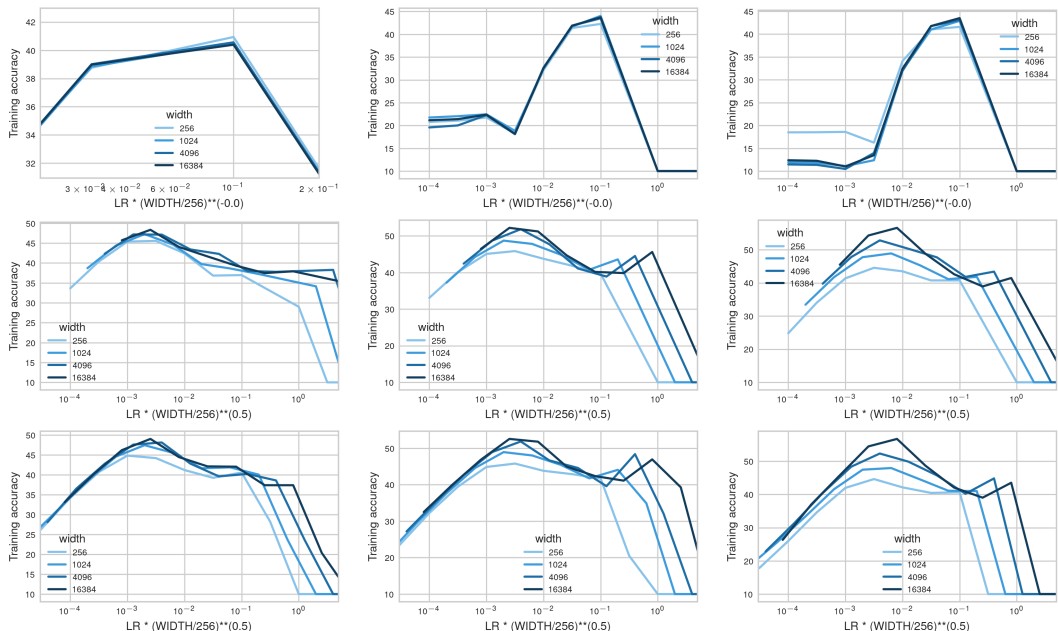

Figure F.35: **(Effective update variants for SGD on CIFAR-10)** MLPs with 2, 3 an 6 layers (from left to right) trained with SGD on CIFAR-10 in $\mu$P (top) versus SP full-align from Everett et al. (2024) (2nd row) versus SP full-align with larger last-layer learning rate (MUSOLI) (bottom row). While $\mu$P transfers with low variance as expected (left), $\mu$P with large standard last-layer initialization $b_{L+1} = 1/2$ and large last-layer learning rate $c_{L+1} = 1/2$ (right) have a non-trivial optimal learning rate scaling between $\Theta(n^{-1/2})$ and $\Theta(1)$, while the maximal stable learning rate scales width-independently.

As expected from parameterizations in the controlled divergence regime, Figure F.35 also shows that the maximal stable learning rate scales width-independently, since activation and gradient stability is preserved. Over the course of 20 epochs, the training dynamics under large learning rates in MLPs with at least 3 layers are stabilized and the optimal learning rate indeed scales width-independently under standard last-layer initialization. Hence width-dependence in parameterizations can induce optimal learning rate scaling that varies over the course of long training. But often the optimal learning rate scales like the maximal stable learning rate. In such cases our theory is predictive. For MSE loss, Figure F.36 shows that logit divergence needs to be avoided through optimal and max-stable learning rate scaling $\eta_n = \Theta(n^{-1/2})$, irrespective of single or multi-epoch settings, because training diverges in the first steps under larger learning rates. This shows that $\mu$P is necessary for making MSE loss a viable alternative to CE loss, avoiding logit divergence while recovering feature learning. Under CE loss, SP full-align and MUSOLI are more robust to poor tuning of the learning rate than $\mu$P, both in terms of training and test accuracy (Figures F.37 and F.38). We leave a closer analysis of the multi-epoch setting to future work.

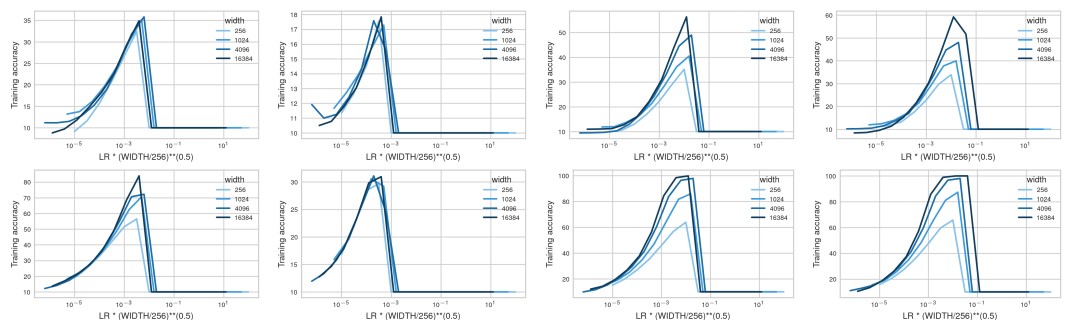

Figure F.36: **(Effective update variants with SGD under MSE loss avoid logit blowup)** Training accuracy of 2-layer, 3-layer linear, 6-layer and 8-layer MLPs (from left to right) trained with SGD for 1 epoch (top) and 20 epochs (bottom) on CIFAR-10 in SP full-align from Everett et al. (2024). Optimal learning rates shrinking as $\eta_n = \Theta(n^{-1/2})$ persists, avoiding logit blowup through $W_0^{L+1}\Delta x_t^L$. Only in 8-layer MLPs is the optimal learning rate saturating at the width-independent stability threshold.

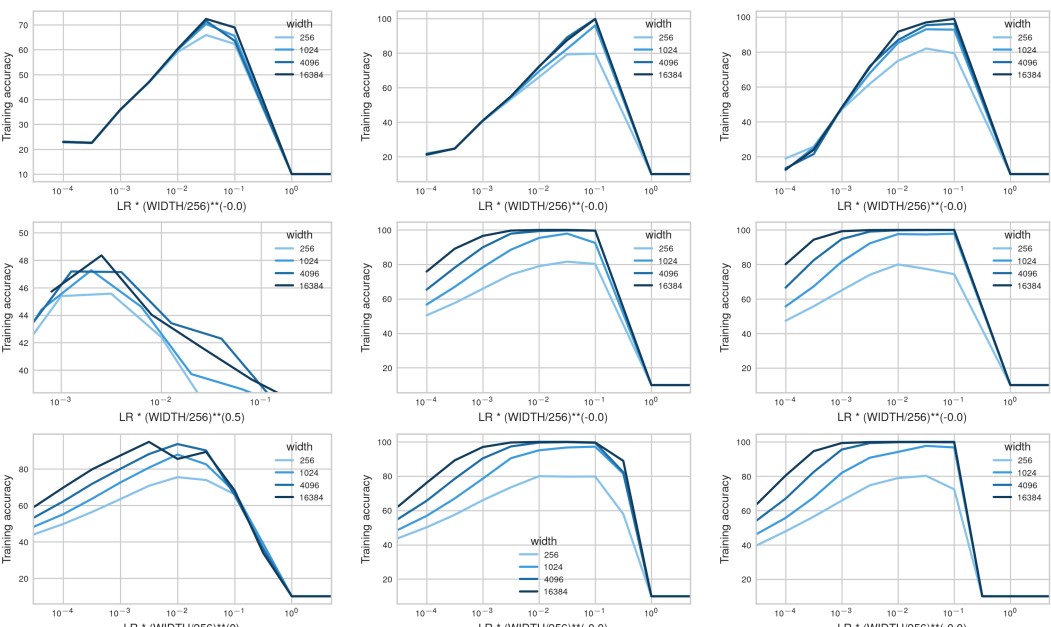

Figure F.37: **(Effective update variants for SGD on CIFAR-10 after convergence)** MLPs with 2, 3 an 6 layers (from left to right) trained with SGD in $\mu$P (top) versus SP full-align from Everett et al. (2024) (2nd row) versus SP full-align with larger last-layer learning rate (MUSOLI) (bottom row) as in Figure F.35 but trained for 20 epochs. After sufficiently long training the large learning rate dynamics stabilize in MUSOLI so that the optimum indeed scales width-independently. MUSOLI strictly dominates original $\mu$P in training accuracy, and robustness to badly tuned learning rate is strongly improved under SP last-layer initialization compared to original $\mu$P. In sufficiently deep MLPs, the larger last-layer learning rate barely matters, but in 2-layer nets SP-full align avoids output blowup and feature learning by transferring under $\eta_n = \Theta(n^{-1/2})$.

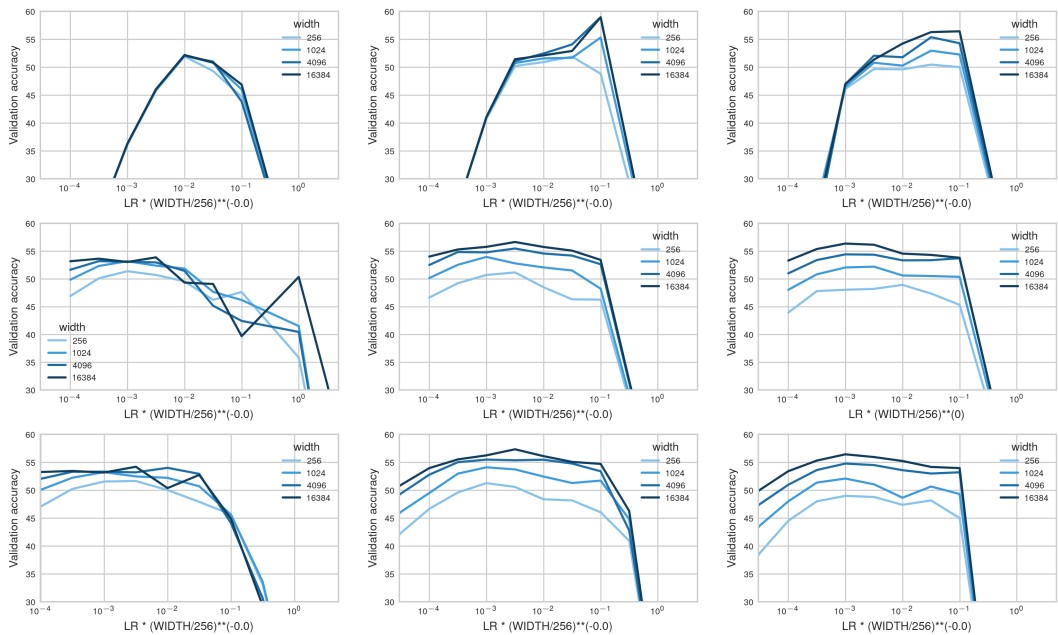

Figure F.38: **(Test accuracy of effective update variants for SGD on CIFAR-10 after convergence)** Test accuracy of 2-layer, 3-layer and 6-layer (from left to right) MLPs trained with SGD for 20 epochs on CIFAR-10 in $\mu$P (top) versus SP full-align from Everett et al. (2024) (2nd row) versus SP full-align with larger last-layer learning rate (MUSOLI) (bottom row). The validation-optimal learning rate scales width-independently in all cases. Observe that, while all variants generalize similarly well, the susceptibility to poorly tuned learning rates is much larger in $\mu$P than under parameterizations with large last-layer initialization.

For ADAM, the gradient is normalized in the backward pass, so that input- and hidden-layer learning rates remain the same as in $\mu$P under large last-layer initialization. This is again equivalent to the SP full-align parameterization from Everett et al. (2024). The logit update term $W_0^{L+1}\Delta x_t^L = \Theta(n^{1/2})$ should again be balanced with a larger output layer learning rate $\eta_{L+1} = \Theta(n^{-1/2})$ if the weight updates of all layers should have a non-vanishing effect on the softmax output in the infinite-width limit (MUSOLI). Figure F.39 shows that nonlinear networks trained with Adam and large last-layer initialization already tend to transfer better under MUSOLI than under SP full-align after 1 epoch. Linear networks again have smaller optimal learning rate exponent, indicating that avoiding logit blowup improves over feature learning in this case, where feature learning does not even add expressivity. Generalization, learning rate transfer and learning rate sensitivity after 20 epochs tends to be similar in all 3 considered parameterizations in deep ReLU MLPs (Figure F.40), showing again that parameterizations with logit blowup are a viable alternative.

Especially in deep ReLU MLPs, the last-layer learning rate does not seem to have a big impact, and SP full-align and MUSOLI overall behave similarly for both SGD and Adam.

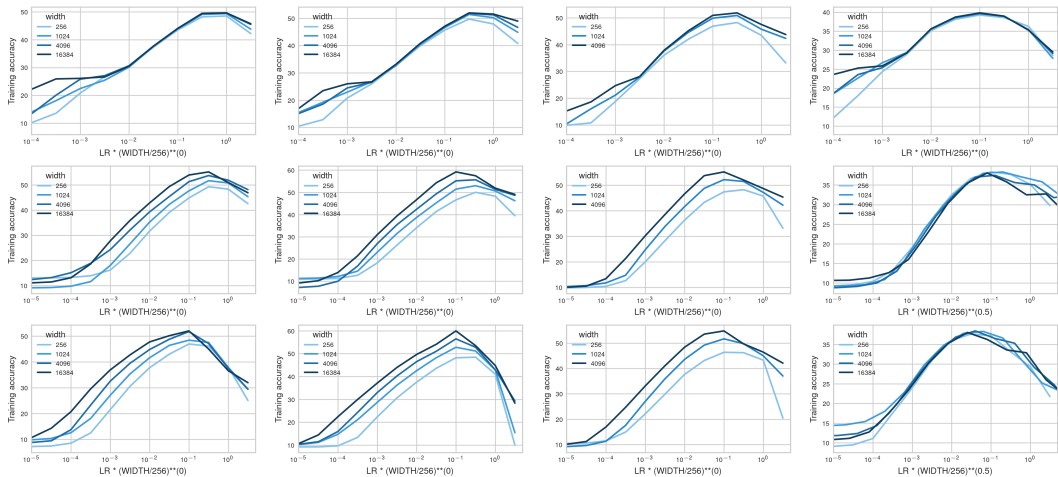

Figure F.39: **(Train accuracy of effective update variants for ADAM on CIFAR-10)** Train accuracy of 2-layer, 3-layer, 6-layer and 3-layer-linear MLPs (from left to right) trained with ADAM for 1 epoch on CIFAR-10 in $\mu$P (top row) versus SP full-align from Everett et al. (2024) (2nd row) versus MUSOLI (bottom row). The learning rate transfers irrespective of the architecture in $\mu$P. Large last-layer learning rate improves transfer in MUSOLI over SP full-align. The optimal learning rate scales as $\eta_n = \Theta(n^{-1/2})$ in both parameterizations with large last-layer initialization, as feature learning does not improve expressivity.

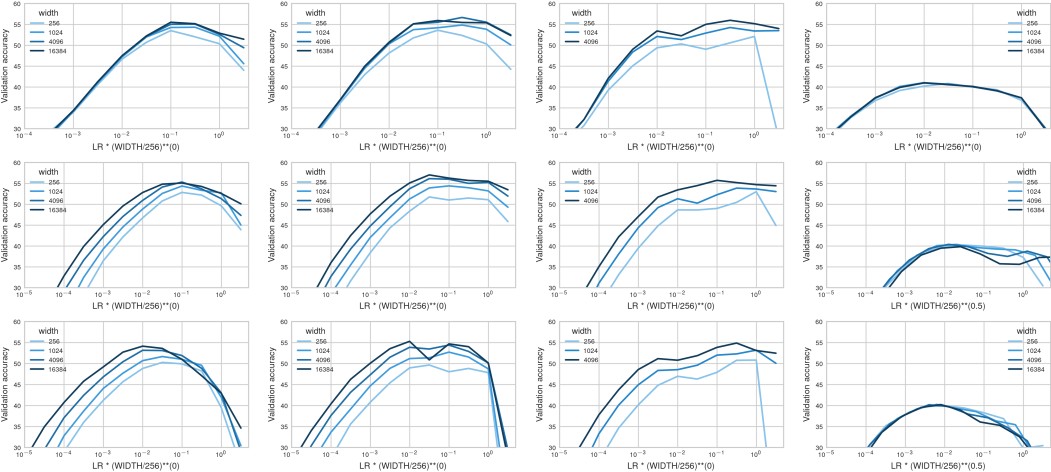

Figure F.40: **(Test accuracy of effective update parameterizations for ADAM on CIFAR-10 after convergence)** Test accuracy of 2-layer, 3-layer, 6-layer and 3-layer-linear MLPs (from left to right) trained with ADAM for 20 epochs on CIFAR-10 in $\mu$P (top row) versus SP full-align from Everett et al. (2024) (2nd row) versus MUSOLI (bottom row). The validation-optimal learning rate scales width-independently in all ReLU MLPs with at least 3 layers. 3-layer linear networks clearly transfer under $\eta_n = \Theta(n^{-1/2})$ in SP full-align and MUSOLI, as for sufficient width learning features does not add expressivity, and instead avoiding logit blowup dominates the learning rate scaling.

