# OpenReview forum: "On the Surprising Effectiveness of Large Learning Rates under Standard Width Scaling"
_NeurIPS.cc/2025/Conference — NeurIPS 2025 spotlight_

### Official Review · Reviewer_r9tC · 2025-06-14

**Clarity:** 3
**Significance:** 3
**Originality:** 4
**Rating:** 5
**Confidence:** 3

**Summary:**

This paper tackles the discrepancy between theoretically predicted optimal (stable) learning rates (LRs) and empirically observed ones as model scale increases under standard parameterization (SP). The authors particularly highlight the role of cross-entropy (CE) loss, which in contrast to MSE loss allows stable training under large LRs even with divergent logits. Interestingly, as shown in the paper, neither long-training dynamics nor the catapult mechanism arguments are able to explain the stability of large LRs in SP. Instead, thanks to the properties of the CE loss derivative, training dynamics remains stable with LR exponent up to $-1/2$, which aligns well with practice. Experiments confirm the theoretical predictions and help further elucidate the intuition behind different choices of architectural elements, optimizers, and parameterizations.

**Questions:**

Please refer to Weaknesses.

**Ethical Concerns:**

["NO or VERY MINOR ethics concerns only"]

**Final Justification:**

In light of the authors rebuttal and other reviews, I decide to keep my initial high score as I consider this work a strong submission.

**Limitations:**

Yes

**Quality:**

4

**Strengths And Weaknesses:**

# Strengths

This paper provides a solid foundation for the empirically observed LR scaling laws, bridging the gap between theory and practice and enabling better understanding and more accurate prediction of optimal LR values at scale.
The idea to particularly consider the role of the loss function seems novel and interesting, which is a solid contribution of this work.
It is indeed counterintuitive to anticipate stable training dynamics and even feature learning when model outputs (logits) enter the divergence regime.
The authors did a great job of explaining the practical benefits of cross-entropy loss, and I specifically like the part demonstrating that MSE loss can possess similar benign properties under $\mu$P, which makes it both theoretically attractive and practically usable.

The authors clearly position their work in relation to prior art, explicitly focusing on missing parts of the puzzle and suggesting possible explanations.

The theoretical results are supported by extensive practical experiments and detailed discussions.

Writing is clear and consistent.
The paper is well-written, although it is still relatively hard to follow by a nonspecialist in deep learning scaling theory.

# Weaknesses

Since I am not an expert in deep learning scaling theory, I cannot fully assess the possible technical limitations of this work.
For me, the main weakness of this work is the unclear practical implications of the proposed theory, which has a descriptive rather than predictive nature.
I would suggest that the authors pay more attention to this aspect in a future revision.

---

> ### Author Rebuttal · Authors · 2025-07-30
>
> Thank you for your thoughtful review and constructive criticism. We are delighted about your positive evaluation of our work, acknowledging **clear writing and extensive experiments**, in particular that you find our **theoretical contributions novel and interesting with solid contributions and counterintuitive insights into stable feature learning despite logit divergence**. We were also surprised by this surprisingly simple resolution of a long-standing and seemingly hard puzzle, and are also excited about the potential for practically relevant theory in the controlled divergence regime as well as for the practical potential of using MSE loss with muP. We will now address your main point of criticism.
>
> ### **Practical implications**
>
> We are grateful for this criticism and agree that it is important to further clarify how useful and surprisingly accurate our width-scaling theory is for large-scale model training. We will state these implications more clearly in the revision. We believe that our work has the following practical implications:
>
> 1. **Constraining the search space for the optimal learning rate.**
>
> Our experiments (pre-existing as well as novel experiments are summarized in the table in the Response to Reviewer dfje) strongly support the prediction from Proposition 1 that the maximal stable learning rate scales as $\Theta(1/\sqrt{n})$. This result provides a concrete constraint on the learning rate search space, significantly reducing the range practitioners must consider. Rather than exhaustive or heuristic searches, practitioners may narrow their search around the theoretically justified scaling, leading to substantial computational savings and more efficient hyperparameter tuning. Furthermore, we observe that, particularly in deep nonlinear architectures, the optimal learning rate often saturates at this maximal stable scaling (see e.g. Figures 5 and 6 (left), Figures F.4 and F.5, we will add all learning rate curves corresponding to the added learning rate exponent table to the appendix). Thus, employing the correct scaling often effectively enables approximate transfer of optimal learning rates.
>
> 2. **Potential performance gains at scale using alternative loss functions with muP.**
>
> Correctly attributing the observed performance gap between MSE and cross-entropy (CE) loss in standard parameterization (SP) to scaling considerations reveals a concrete recommendation: practitioners could leverage stable parameterizations like muP for image datasets or SP-full-align for language datasets to effectively utilize loss functions beyond cross-entropy at large scales. Specifically, your point about correct width-scaling via muP enabling alternative loss functions to excel at large scales is further validated by our extended experiments. *We will extend Figure 7 to width 16384 and add results for FashionMNIST* (which closely resemble CIFAR-10). In the extended experiments, we still consistently observe that CE loss greatly outperforms MSE under SP, whereas MSE consistently outperforms CE loss under muP, with differences becoming particularly pronounced at large widths. These results highlight the potential for substantial performance gains from further investigating the interplay between the loss function and parameterization. They also suggest that it is worth exploring the use of further loss functions in conjunction with muP.
>
> 3. **Identifying the correct scaling mechanisms enables principled improvements of scaling practice.**
>
> *Identifying the correct causal mechanism for training stability under large learning rates enables principled and informed exploration of the extended search space of practically relevant potentially best-performing parameterizations for efficiently finding improvements in model scaling practice:* The controlled divergence regime had previously been neglected, even though common scaling practice as well as the best-performing parameterization SP-full-align from Everett et al. (2024) exactly operate in this regime. We now detail two concrete implications together with exciting avenues for future work.
>
> **Identifying the correct mechanism for training instability at large scales.** Training instability due to logit divergence at large scales is a *widely-established empirical phenomenon under SP* (Chowdhery et al., 2022; Wortsman et al., 2023) which has motivated popular interventions like the z-loss (Chowdhery et al., 2022; Wortsman et al., 2023). However, only understanding the underlying causal mechanism and providing the exact width-dependent scaling exponents enables to design **principled interventions**. Our results precisely identify the mechanism responsible: practically relevant large-scale neural networks naturally operate at the boundary of the controlled divergence regime, allowing persistent hidden-layer feature learning despite logit divergence. Accurately attributing this phenomenon to the correct mechanism opens the door for principled and theoretically grounded interventions. Concretely, our results clarify that practical networks trained under SP lie in the controlled divergence regime and therefore inherently cause logit divergence, leading to numerical instability. This provides a concrete hypothesis to test: instability issues due to logit divergence could be systematically resolved with stable parameterizations like muP potentially eliminating the need for ad-hoc interventions like the z-loss. A thorough investigation of this is beyond the scope of the current paper.
>
> **Logit divergence induces overconfidence.** A similar argument applies to uncertainty calibration. Our theory also explains why we should expect *predictions to be increasingly overconfident with increasing model scale and suggests that this may be partially mitigated by considering stable parameterizations like muP.* However there maybe inherent trade-offs between miss-calibration and faster memorization due to logit divergence. Their effect on performance and designing the ideal intervention deserves a more thorough investigation beyond the scope of the current paper.
>
> 4. **Faithful evaluation of the accuracy of infinite-width theory.**
>
> As a first contribution, we provide a refined evaluation of whether the width-dependent exponents of important quantities like effective updates or the maximal-stable learning rate scaling predicted by infinite-width theory accurately hold over the course of training. As opposed to previous work (Everett et al., 2024), we do find that *these predictions hold remarkably accurately already at moderate width ≤512 and over the course of training* (see e.g. Figures 4 and 5, F.4 and F.5), when evaluated with sufficient attention to detail:
>
> While in principle the refined coordinate check (RCC) was previously known (Yang and Hu, 2021, Appendix H), we find that the *predicted update exponents are surprisingly accurate in this decomposition already at moderate width $≤512$ and over the course of training* (Figures 2 and 4), *even in parameterizations with width-dependent dynamics like SP*. To lower the hurdle for practitioners to incorporate the RCC in their workflow as a diagnostic tool, *we will make our fine-grained and easily adaptable implementation of the RCC using LitGPT publicly available upon acceptance.* Understanding and correcting miss-scaled update signals through each weight matrix has impactful consequences on performance and trainability at large scale.
>
> **References:**
>
> Chowdhery et al. “Palm: Scaling language modeling with pathways.” *JMLR* (2023).
>
> OLMo Team. “2 olmo 2 furious.” *arXiv:2501.00656* (2025).
>
> Wortsman et al. “Small-scale proxies for large-scale transformer training instabilities.” *ICLR* (2024).

---

> > ### Comment · Reviewer_r9tC · 2025-08-05
> >
> > I thank the authors for their rebuttal that substantially elaborates the unclear points in the original manuscript, specifically the practical utility of the presented findings. Therefore, I decide to keep my initial high score for this submission.

---

### Official Review · Reviewer_dfje · 2025-07-02

**Clarity:** 2
**Significance:** 2
**Originality:** 2
**Rating:** 4
**Confidence:** 2

**Summary:**

This paper investigates why standard parameterization (SP) with He initialization remains effective at large learning rates, despite infinite-width theory predicting instability. The authors demonstrate that cross-entropy loss fundamentally changes the stability landscape compared to MSE loss, enabling a regime where logits diverge but training remains stable. They provide theoretical analysis and validate their predictions empirically across small tasks on MLPs and Transformers.

**Questions:**

- How does numerical precision affect the controlled divergence regime in practice?
- What are the practical implications beyond explaining why SP works?
- What happens at very large widths or training times?

**Ethical Concerns:**

["NO or VERY MINOR ethics concerns only"]

**Final Justification:**

The authors addressed most of my critics and answered my questions. I'm still concerned about the weak empirical evaluation and the presentation ("golden thread").

**Limitations:**

The authors briefly mention numerical considerations but do not adequately address how floating point limitations affect the practical applicability of their theoretical results.

**Paper Formatting Concerns:**

No.

**Quality:**

3

**Strengths And Weaknesses:**

### Strengths

- The paper addresses a fundamental puzzle in deep learning theory - why standard parameterization works despite theoretical predictions of instability.
- The key insight about CE loss enabling a controlled divergence regime is novel and provides a principled explanation for empirical observations.
- Empirical validation across multiple architectures (MLPs, Transformers), optimizers (SGD, Adam), and datasets strengthens the theoretical claims.
- The connection between width-scaling theory and practical learning rate selection cloud be valuable for practitioners.

### Weaknesses

- I found the paper a bit hard to read and missed a clear "golden thread".
- Proposition 1 (line 208) is marked as "informal" but is central to the paper's claims. The formal statement in Appendix C.3 should be in the main text.
- Numerical precision issues when logits diverge are mentioned only briefly (line 354). This is a critical practical concern that undermines the applicability of the controlled divergence regime.
- Section 5.2 on SP-full-align feels disconnected from the main narrative and the claim about "breaking learning rate transfer on image datasets" (line 319) needs more thorough investigation.
- While the paper explains why SP works, it doesn't lead to better training methods or actionable insights for practitioners. The contribution is primarily explanatory rather than enabling new capabilities.
- In general, I found the scope of the paper a bit unclear and it lacks a strong empirical validation that the width-scaling considerations provide good predictions of the maximal stable learning rate and that the maximal stable learning rate often dominates the optimal learning rate.

---

> ### Author Rebuttal · Authors · 2025-07-30
>
> Thank you for your thoughtful review and constructive criticism. We are delighted that you agree our results **resolve a fundamental puzzle in deep learning theory by providing a principled explanation through the controlled divergence regime, and that you also recognize their potential practical impact, validated by our empirical evaluations, on practical learning rate selection.** Below, we address your main points of criticism and answer your questions.
>
> **Scope.** As you acknowledge, our results resolve a **fundamental and long-standing problem in deep learning theory**: we want to understand finite nets in SP, but previous work had only proposed proxy models that are easier to study but differ qualitatively in their feature learning ability. As we show, practical neural networks operate in a understudied controlled divergence regime. Our theory resolves this fundamental puzzle, closes the gap between infinite-width theory and width-scaling practice and hopefully enables the theory community to generate more practically relevant insights.
>
> *“It is unclear that maximal stable learning rate often dominates the optimal learning rate.”*
>
> We acknowledge that our discussion in the paper might have caused some confusion regarding the definitions of the maximal stable learning rate (**max-stable LR**). To address this, we clarify the definitions explicitly below, and will incorporate these clarifications into the revised manuscript.
>
> **Theoretical max-stable-LR:** Historically, the theoretical maximal stable LR of a neural network is defined as the boundary beyond which either the logits or at least one layer's activations become unstable with width.
>
> **Empirical max-stable-LR:** A natural empirical equivalent of a max-stable-LR of a network is the largest learning rate beyond which training diverges, precisely quantified by the learning rate threshold beyond which the network’s predictions degrade to random guessing (e.g., within ~1% of random guessing accuracy). **Under this definition, the max-stable LR of the network clearly provides a strict upper bound on the optimal learning rate.**
>
> We appreciate the reviewer’s feedback, which helps us refine and clearly convey these foundational concepts in our paper.
>
> *“Empirical evidence for learning rate scaling in SP.”*
>
> Thank you for encouraging us to provide a more accessible summary of our empirical results. For convenience, we have included tables of learning rate (LR) exponents below, and we will ensure they appear at least in the Appendix to provide an accessible overview, referenced clearly from the main paper.
> *Additionally, we now provide the optimal and maximal stable learning rate exponents on two further datasets: FashionMNIST and TinyImagenet. Notably, for every dataset considered, the maximal stable learning rate exponent consistently aligns closely with our theoretical predictions, namely -0.5 for CE loss and -1 for MSE loss.*
>
> The optimal LRs generally follow the same scaling, though they occasionally exhibit strong finite-width effects. In our response to Reviewer YurV, we summarize relevant existing literature that establishes the connection between optimal and maximal stable learning rates.
>
> | Dataset | Architecture | Optimal LR exponent | Maximal stable LR exponent |
> | --- | --- | --- | --- |
> | tinyimagenet | 8-layer MLP | -0.3347 | -0.3347 |
> | fashionmnist | 8-layer MLP | -0.55 | -0.7 |
> | mnist | 8-layer MLP | -0.46 | -0.54 |
> | cifar10 | 8-layer MLP | 0.07 | -0.37 |
> | dclm-baseline | GPT | -0.38 | -0.38 |
>
> Even when the optimal LR exponent does not closely track the maximal stable exponent - as observed with CIFAR-10 - the learning rate curves (such as Figure F.19, summarized in the table below and averaged over four random seeds) demonstrate that the **optimal LR exponent saturates toward the maximal stable exponent at width 16384.** This trend is expected to continue at larger widths, where our theory holds even more accurately. All corresponding learning rate curves will be included in the revised Appendix.
>
> **CIFAR-10, SGD, SP, CE loss**
>
> | Width | 2.05e-02 | 4.10e-02 | 8.19e-02 | 1.64e-01 | 3.28e-01 | 6.55e-01 |
> | --- | --- | --- | --- | --- | --- | --- |
> | **256** | **42.57** | 41.51 | 40.95 | 38.26 | 23.62 | 10.00 |
> | **1024** | 44.33 | **44.54** | 44.45 | 42.81 | 10.00 | 10.00 |
> | **4096** | 46.13 | 46.57 | **46.69** | 10.00 | 10.00 | — |
> | **16384** | 46.10 | **46.92** | 10.00 | 10.00 | — | — |
>
> *“How SP-full-align relates to the main narrative.”*
>
> SP-full-align connects to the main narrative because, like SP with CE loss, **this parameterization also operates in the *controlled divergence regime*:** training remains stable despite systematic logit divergence. The key distinction from standard SP is that, due to its layerwise learning rates, all layers exhibit effective feature learning similar to $\mu$P—a property often linked to improved performance (e.g., Bordelon et al., 2024; Kunin et al., 2024). Additionally, we note that SP-full-align was originally motivated by alignment considerations, which we disprove in Section 3.1. Instead, our work provides a corrected explanation for the empirical behavior observed in networks trained with SP-full-align.
>
> *“More evidence for the claim "breaking learning rate transfer on image datasets under SP-full-align" (line 319).”*
>
> Thank you for suggesting this valuable clarification. *To further strengthen our claim, we conducted additional experiments training 8-layer MLPs on FashionMNIST and TinyImagenet.* Consistent across all our SP-full-align experiments (summarized in the table below), we find that while the maximal stable learning rate remains remarkably width-independent - as predicted by our theory - optimal learning rates decay across all datasets considered. We previously reported a subset of these results in Appendix F.8, but agree that presenting them in an accessible tabular form significantly enhances clarity.
>
> | Dataset | Optimal LR exponent | Maximal stable LR exponent |
> | --- | --- | --- |
> | fashionmnist | -0.35 | 0.0 |
> | tinyimagenet | -1.45 | 0.0 |
> | mnist | -0.25 | 0.0 |
> | cifar10 | -0.33 | 0.0 |
>
> **Provide formal result in the main paper.** We would also have liked to present the full theoretical results in the main paper. But unfortunately this requires introducing all of the Tensor Program definitions and machinery, which would take too much space and would significantly harm readability without significant information gain. We instead opted for distilling the necessary knowledge into a more accessible and practitioner-oriented introduction of the main scaling arguments.
>
> **"Golden thread" of the paper.** Our intention was precisely to provide such a cohesive narrative by first clearly formulating the puzzle of large learning rate stability in SP, demonstrating that existing explanations fall short of addressing the observed discrepancies, and subsequently identifying the underlying mechanism that resolves this puzzle. Finally, we discuss the fundamental implications of stable training despite logit divergence. Given that other reviewers noted the paper is already well-written and accessible, we would particularly appreciate additional concrete suggestions on improving clarity further, as ensuring accessibility to a broad audience was one of our main concerns while writing this paper.
>
> **Larger width and longer training time.** We emphasize that our experiments already span a sufficiently large width to be practically meaningful: the hidden dimension of the LLAMA-3 model with 405B parameters is 16384, exactly matching the width used in our MLP experiments. Due to computational constraints in our academic environment, we are unable to provide experiments at even larger scales. However, we expect our theoretical predictions would only become more accurate at widths beyond this already substantial scale. Indeed, **it is particularly noteworthy that our theory already yields accurate exponent predictions at these practically relevant widths.** We also expect our experiments to hold robustly over longer training time in the online setting. The discussion in the multi-epoch setting is more nuanced as detailed in our response to reviewer YurV.
>
> *“Contribution is primarily explanatory rather than enabling new capabilities.”*
>
> While we strongly believe that the explanatory nature of our work is valuable in its own right, there are several concrete *practical implications of our work. Due to space constraints we have provided a detailed response to this point in our response to Reviewer r9tC*.
>
> *“How does numerical precision affect controlled divergence?”*
>
> Note that our results highlight that *neural nets in standard practice operate in the controlled divergence regime* so we should expect training instability due to finite precision  — which is precisely what has been established empirically (Chowdhery et al., 2022; Wortsman et al., 2023). Our results precisely identify the mechanism responsible: practically relevant large-scale neural networks naturally operate at the boundary of the controlled divergence regime, allowing persistent hidden-layer feature learning *despite logit divergence*. (also see our response to r9tC where we highlight practical implications of this attribution).
>
> We sincerely *thank the reviewer for stimulating several changes that will enhance clarity and provide more convincing evidence for our claims*. Overall, we see the revised paper as a necessary intermediate step with fundamental contributions that opens up several exciting avenues for future work. We hope that our changes and clarifications address your main concerns. In that case, *we kindly ask you to consider raising your score*. We are happy to elaborate further, if any questions remain unanswered or if there remain further fundamental concerns.

---

### Official Review · Reviewer_YurV · 2025-07-05

**Clarity:** 3
**Significance:** 3
**Originality:** 3
**Rating:** 5
**Confidence:** 2

**Summary:**

The authors try to answer why standard parameterization remains stable and effective at large learning rates, despite the theoretical predictions. While the catapult effect and edge of stability theory does not fully explain this phenomenon, the authors find a reason from the choice of cross-entropy (CE) loss as a loss function. The authors insist that CE loss allows stable training under large learning rates even when the output logits diverge.

**Questions:**

1.	Is inducing the maximal stable learning rate from an optimal learning rate a frequently exercised custom or wisdom in this area?
2.	Page 30 mentions that the authors train a single epoch to prevent confounding from multi-epoch overfitting effects. Could you explain more on why this is necessary for the experiments?
3.	One question for my sheer curiosity. It is known that a cross-entropy loss leads to a max-margin solution. What would be the relationship between the learning rate and a converged solution? Beyond a maximal stable learning rate, would the parameter not converge to a max-margin solution?

**Ethical Concerns:**

["NO or VERY MINOR ethics concerns only"]

**Final Justification:**

I am satisfied with their clarification in the rebuttal. So I raise the score.

**Limitations:**

Yes

**Paper Formatting Concerns:**

Not that I know of.

**Quality:**

3

**Strengths And Weaknesses:**

Strengths

1.	The paper is easy to follow even for those who are not familiar with the subject. The informal explanations of the propositions and theorems as well as the proof sketches highly enhance the readability and help the readers understand the intuition behind the theories.
2.	The paper has strong theoretical contributions. The focus on the cross-entropy loss seems very interesting to me.
3.	The experimental results seem to support the theoretical results as well. The experiments seem to be extensive and include examples on large models like GPT.

Weaknesses

1.	I am not very much fond of the idea of inducing the maximal stable learning rate from an optimal learning rate. The authors briefly explain that Proposition 2 suggests that the idea is true since it is the only setting under which feature learning is preserved at large width in all hidden layers. I only partially agree with the authors’ opinion, and I think authors need to provide a more rigorous explanation to confirm this. The result is practically useful, though.
2.	In Page 30, it says that the authors train a single epoch to prevent confounding from multi-epoch overfitting effects. I personally cannot find a particular reason why this is necessary, since feature learning often occurs after a certain period of training time, especially when the data is complex and grokking occurs.

---

> ### Author Rebuttal · Authors · 2025-07-30
>
> Thank you for your thoughtful review and constructive criticism. We are thrilled that you find our paper **easy to follow even for readers unfamiliar with the theory, with strong theoretical contributions elucidating the important effect that cross-entropy loss has on training stability and with extensive experiments also covering practical LLM settings**. We now address your main points of criticism and answer your questions.
>
> ### **Is optimal LR scaling induced by the maximal stable LR scaling?**
>
> First, let us clarify the definitions of maximal stable learning rates (**max-stable LR**):
>
> - **Theoretical definition:** Historically, the maximal stable LR for a neural network was defined as the boundary beyond which either the logits or at least one layer's activations become unstable with width. A layer-specific variant defines this as the LR threshold at which that particular layer’s activations diverge.
> - **Empirical definition:** Empirically, the maximal stable LR is naturally defined as the largest learning rate beyond which training diverges, precisely quantified by the learning rate threshold beyond which the network’s predictions degrade to random guessing (e.g., within ~1% of random guessing accuracy).
>
> Previous work (Yang et al., 2022; Dey et al., 2023; Everett et al., 2024) extensively demonstrates that *when all layers share the same theoretical maximal stable LR exponent, both empirical max-stable-LR and optimal LR scaling closely matches the theoretical max-stable-LR scaling***.** This phenomenon is robustly established through numerous experiments, particularly in the context of $\mu$P. However, under SP, this discussion becomes much more nuanced since 1) Different layers exhibit distinct theoretical max-stable LR exponents and it is not obvious how to define the theoretical max-stable LR at a network level. 2) It is also unclear what the scaling exponents of empirical maximal stable LR and optimal LR look like.
>
> **Key contributions of our work:**
>
> - **Breakdown of correspondence:** We reveal that under cross-entropy (CE) loss, the empirical maximal stable LR does **not** correspond with the traditional theoretical maximal stable LR. Instead, we find that empirical maximal stable LRs occur at the boundary to the **catastrophically unstable regime where both logits and activations of all layers diverge**, rather than merely at the edge of the stable regime (as previously believed).
> - **Theoretical redefinition:** This observation motivates us to formally redefine the theoretical maximal stable LR explicitly as the boundary to this catastrophically unstable regime.
> - **Optimal LR scaling alignment:** To our surprise, our experiments show that when network dynamics are dominated by a certain layer type (e.g., deep nonlinear MLPs dominated by hidden layers), the empirical optimal LR scaling often saturates towards our definition of theoretical maximal stable LR scaling with respect to that layer type.
>
> We intentionally **refrain** from making explicit claims about the exact scaling exponent for the optimal LR. Although we narrow its potential range and successfully predict optimal LR values in many scenarios, we acknowledge limitations and the difficulty of fully characterizing conditions under which this alignment breaks down. Properly accounting for finite-width effects remains essential yet highly nontrivial. We believe a rigorous characterization of the optimal LR exponent at finite widths remains challenging, as it likely requires strong assumptions about architecture or data distributions—analogous to assumptions required by neural scaling laws (Hoffmann et al., 2022; Bachmann et al., 2024). Hence, this paper intentionally emphasizes general insights on signal propagation and training stability, deferring deeper theoretical exploration of optimal LR exponents to future work (as explicitly detailed in the subsection *‘Understanding optimal learning rate exponents’* of our future work section).
>
> Empirically, however, we emphasize that *the optimal learning rate consistently saturates at the maximal stable learning rate at sufficient widths across all our SGD experiments in SP, including GPT-like transformers and deep ReLU MLPs* (summarized in the table in the Response to Reviewer dfje). While the empirical connection between optimal and max-stable learning rates is well-documented, the mechanisms remain poorly understood, independent of model scale (see related *edge-of-stability literature* by Cohen et al., 2021, 2022; Lewkowycz et al., 2020; Cai et al., 2024; Damian et al., 2023, among others). Our findings offer a novel perspective on the advantages of large learning rates for facilitating feature learning at large model scales. Previously suggested explanations include improved generalization through reduced sharpness (Andriushchenko et al., 2023a), a shift in the learning order of patterns (Li et al., 2019), enhanced SGD noise (Keskar et al., 2017), and implicit bias towards sparsity (Andriushchenko et al., 2023b).
>
> ### **Multi-epoch setting**
>
> Thanks for raising this important question, we will provide this clarification in the main paper. First, we would like to clarify that the notion of feature learning commonly used in the context of grokking - i.e., alignment with task-specific features - differs significantly from our definition of feature learning in the context of width scaling, which specifically refers to feature updates that neither diverge nor vanish as width scales. Under this latter definition, feature learning in SP within the controlled divergence regime occurs after only one step of SGD, without requiring prolonged training.
>
> To evaluate our theoretical claims, which currently address width dependence without exploring dependencies on training time or data size, we consider the online training setting to be most appropriate. Online training avoids introducing additional overfitting dynamics that have been theoretically shown to accumulate when data points are repeatedly presented (Bordelon et al., 2024). Furthermore, we focus explicitly on the optimization effects of  learning rate rather than its regularizing properties. Recent theoretical work indicates that SGD noise has negligible influence on learning-rate scaling in the online scenario (Paquette et al., 2024). Finally, we note that modern large language models are often trained online, underscoring the practical relevance of our analysis.
>
> But we do agree that the multi-epoch setting is an interesting and important setting to study in the future, and *will also mention it in the future work section*.
>
> ### **Relation to max-margin solutions**
>
> As opposed to our more architecture- and data-agnostic stability considerations, the convergence results to a max-margin solution are more architecture (often only 2-layer networks) and data-dependent (see e.g. Lyu et al. (2021) for positive and negative examples), and they often study gradient flow. When a neural net converges to a max-margin solution under gradient flow, we expect convergence to continue to hold for ‘small enough’ LRs. Beyond the max-stable LR, the parameters would indeed diverge, because large updates induce divergence in the activations, which leads to a cascading feedback loop between exploding gradients and activations. To provide a concrete example for CE loss, under favorable separability assumptions, consider Theorem 3.2 in Cai et al. (2025) which considers muP with width-independent LR. Here, increasing the LR while keeping training time fixed implies divergence with increasing model scale.
>
> We sincerely *thank the reviewer for stimulating several changes that will enhance clarity and provide more convincing evidence for our claims*. Overall, we see the revised paper as a necessary intermediate step with fundamental contributions that opens up several exciting avenues for future work. We hope that our changes and clarifications partially address your main concerns. In that case, *we kindly ask you to consider raising your score*. We are happy to elaborate further, if any questions remain unanswered or if there remain further fundamental concerns.
>
>
> ###  **Added References:**
>
> Andriushchenko et al. “A modern look at the relationship between sharpness and generalization.” *ICML* (2023a).
>
> Andriushchenko et al. “Sgd with large step sizes learns sparse features.” *ICML* (2023b).
>
> Bachmann et al. “Scaling mlps: A tale of inductive bias.” *NeurIPS* (2024).
>
> Bordelon et al. "A dynamical model of neural scaling laws." *ICML* (2024).
>
> Cai et al. "Implicit Bias of Gradient Descent for Non-Homogeneous Deep Networks." *arXiv:2502.16075* (2025).
>
> Cai et al. “Large stepsize gradient descent for non-homogeneous two-layer networks: Margin improvement and fast optimization.” *NeurIPS* (2024).
>
> Hoffmann et al. “Training compute-optimal large language models.” *arXiv:2203.15556 (2022).*
>
> Keskar et al. “On large-batch training for deep learning: Generalization gap and sharp minima.” *ICLR* (2017).
>
> Li et al. “Towards explaining the regularization effect of initial large learning rate in training neural networks.” *NeurIPS* (2019).
>
> Lyu et al. "Gradient descent on two-layer nets: Margin maximization and simplicity bias." *NeurIPS* (2021).

---

> > ### Comment · Reviewer_YurV · 2025-08-06
> >
> > Thank you for the detailed explanation and clarification. Especially, my inquiry about inducing the optimal LR from the maximal stable LR has been addressed clearly. Therefore, I will raise my score.

---

### Comment · Area_Chair_oMUe · 2025-08-04

Dear Reviewers,

As the discussion deadline approaches, please kindly review the authors’ responses and share your thoughts. Please also remember to submit the mandatory acknowledgement as well as your "Final Justification". If you have already done so, please ignore this message.

Thank you for your engagement and support!

Area Chair

---

### Decision · Program_Chairs · 2025-09-17

**Decision:**

Accept (spotlight)

**Comment:**

This paper tackles the puzzle of why standard parameterization remains effective at large learning rates despite infinite-width theory predicting instability. Among several potential explanations, the authors identify cross-entropy loss as a key factor, backed by a theoretical analysis based on Tensor Program. They also present a series of experiments across optimizers, architectures, and datasets to further validate the claim.

The reviewers considered the paper a strong contribution to the scaling theory for deep learning. Some concerns on the clarity, weak empirical evaluation and practical implications were raised, but the authors' rebuttal largely addressed these concerns. While some reviewers found the paper easy to follow, the presentation of the paper could be further strengthened for non-experts. Overall, I recommend acceptance.